# RealPDEBench: A Benchmark for Complex Physical Systems with Real-World Data

**Peiyan Hu**[*†1,3] **Haodong Feng**[*1] **Hongyuan Liu**[*1] **Tongtong Yan**[2] **Wenhao Deng**[1]
**Tianrun Gao**[†1,4] **Rong Zheng**[†1,5] **Haoren Zheng**[†1,2] **Chenglei Yu**[1] **Chuanrui Wang**[1]
**Kaiwen Li**[†1,2] **Zhi-Ming Ma**[3] **Dezhi Zhou**[2] **Xingcai Lu**[6] **Dixia Fan**[1] **Tailin Wu**[‡1]
`{hupeiyan, fenghaodong, liuhongyuan, wutailin}@westlake.edu.cn`
[1]School of Engineering, Westlake University; [2]Global College, Shanghai Jiao Tong University;
[3]Academy of Mathematics and Systems Science, Chinese Academy of Sciences;
[4]Department of Geotechnical Engineering, Tongji University; [5]School of Physics, Peking University;
[6]Key Laboratory for Power Machinery and Engineering of M. O. E., Shanghai Jiao Tong University

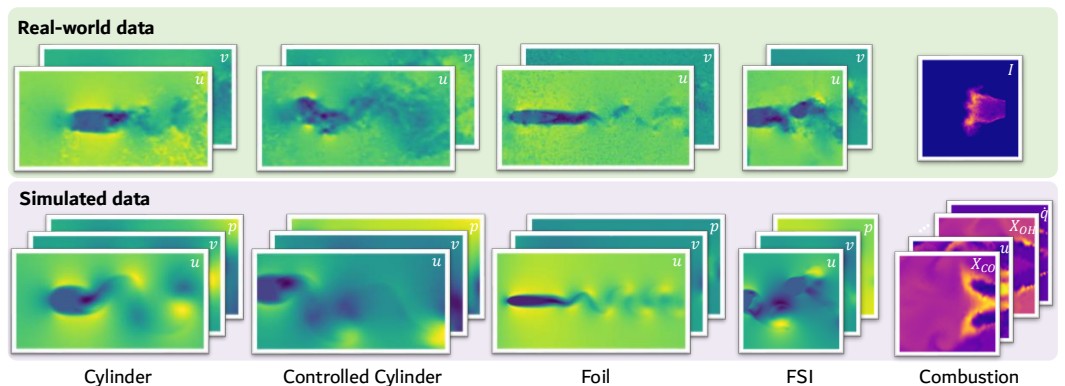

Figure 1: Five scenarios in RealPDEBench with the corresponding real-world and simulated data. It demonstrates the differences between real-world and simulated data, such as the different modalities, measurement noise, and numerical errors. These discrepancies motivate and call for our proposed benchmark, RealPDEBench, to systematically collect data and conduct experimental analysis.

## Abstract

Predicting the evolution of complex physical systems remains a central problem in science and engineering. Despite rapid progress in scientific Machine Learning (ML) models, a critical bottleneck is the lack of expensive real-world data, resulting in most current models being trained and validated on simulated data. Beyond limiting the development and evaluation of scientific ML, this gap also hinders research into essential tasks such as sim-to-real transfer. We introduce RealPDEBench, the first benchmark for scientific ML that integrates real-world measurements with paired numerical simulations. RealPDEBench consists of five datasets, three tasks, nine metrics, and ten baselines. We first present five real-world measured datasets with paired simulated datasets across different complex physical systems. We further define three tasks, which allow comparisons between real-world and simulated data, and facilitate the development of methods to bridge the two. Moreover, we design nine evaluation metrics, spanning data-oriented and physics-oriented metrics, and finally benchmark ten representative baselines, including state-of-the-art models, pretrained PDE foundation models, and a traditional method. Experiments reveal significant discrepancies between simulated and real-world data, while showing that pretraining with simulated data consistently improves both accuracy and convergence. In this work, we hope to provide insights from real-world data, advancing scientific ML toward bridging the sim-to-real gap and real-world deployment. Our benchmark, datasets, and instructions are available at `https://realpdebench.github.io/`.

---

[*]Equal contribution. [†]Work done as an intern at Westlake University. [‡]Corresponding author.

# 1 INTRODUCTION

Predicting the future evolution of complex physical systems is a central and enduring problem in both science and engineering. Such systems play fundamental roles in a wide range of domains, including fluid dynamics (Griebel et al., 1998; Ferziger & Peric, 2002), combustion (Poinsot & Veynante, 2005; Glarborg et al., 2018), and plasma physics (Chen, 2015). These systems are characterized by complex spatiotemporal dynamics. Their high-dimensional, strongly nonlinear, unsteady, and tightly coupled nature presents a formidable barrier to accurate prediction.

To more accurately predict complex physical systems, a series of traditional numerical methods have been developed (Neumaier, 1987; Bratanow, 1978; Dormand & Prince, 1980; Griffith & Patankar, 2020). These methods approximate complex physical phenomena by formulating them as symbolic Partial Differential Equations (PDEs), which are then discretized and solved numerically to obtain approximate solutions. More recently, numerous scientific Machine Learning (ML) models have emerged. These models train neural networks by optimizing loss functions defined either from data (Pfaff et al., 2020; Li et al., 2021) or from governing PDEs (Raissi et al., 2019; Cuomo et al., 2022), thereby learning the dynamics of complex physical systems.

Although scientific ML has gained widespread attention due to its efficiency, it still has some issues. One of the most significant ones is that most current models are only learned and validated on simulated data from traditional numerical solvers, lacking experiments in real-world scenarios (Thiyagalingam et al., 2022). However, due to factors such as numerical errors, measurement noise, and unmeasured modalities (variables of the physical system), there is a huge gap between real-world measured data and numerically simulated data (Roache, 1998; Oberkampf & Trucano, 2002; Kravchenko & Moin, 2000; Veynante & Vervisch, 2002; Hochgreb, 2019). As a result, we cannot truly evaluate how current scientific ML models perform in the real world against numerical methods. The absence of real-world datasets has fundamentally restricted both the development and evaluation of scientific ML. Moreover, this limitation further hinders progress on many important tasks, such as learning from noisy real-world data, sim-to-real transfer, and understanding how limitations of measurement techniques influence model performance. Consequently, real-world datasets are crucial for the advancement of scientific ML, yet they have remained largely scarce (Thiyagalingam et al., 2022). This scarcity primarily stems from the high cost of acquiring real-world data, as it requires the construction of experimental setups and the rich experience of measurement.

To bridge this gap, we propose the first scientific ML benchmark in real-world complex physical systems, RealPDEBench, which contains paired real-world data and simulated data. Our contributions are fourfold: data, tasks, metrics, and baselines. First, from the **data** perspective, we provide more than 700 trajectories, each exceeding 2000 frames under distinct operating conditions, that cover five scenarios in the domains of fluid dynamics and combustion. For each scenario, multiple system parameters are included, and for every parameter setting, both real-world data and simulated data of equal temporal duration are provided. In particular, the Combustion dataset highlights the importance of real-world data, as its inherent complexity, multi-physics, and multi-scale nature make accurate simulation challenging. Second, from the **task** perspective, we consider three settings, all evaluated on real-world data. Specifically, the models are (i) trained on simulated data, (ii) trained on real-world data, and (iii) pretrained on simulated data and then finetuned on real-world data. Through experiments across the three tasks, we are able to compare the respective strengths and limitations of real-world and simulated data. Our work provides a foundation for further exploring how to combine the advantages of both and achieve improved models for future work.

Third, we provide a comprehensive set of evaluation **metrics**, comprising nine in total. Specifically, these metrics can be categorized into data-oriented and physics-oriented metrics, along with a dedicated metric designed for investigating the impact of pretraining on simulated data. Fourth, for the **baselines**, we consider nine data-driven scientific ML models and one traditional method. They include not only the latest state-of-the-art (SOTA) architectures but also the pretrained foundation PDE model. We systematically compare their modeling capabilities on real-world data. All the aforementioned datasets, tasks, metrics, and baselines are integrated into a unified and highly modular code framework, which enables rapid adaptation of prediction and sim-to-real tasks to new datasets or models. Our experimental findings reveal a gap between real-world and simulated data. On the other hand, we also observe that reasonably leveraging simulated data can improve prediction performance on real-world data, which offers guidance for advancing Scientific ML models toward real-world applications.

## 2 RELATED WORK

Several benchmarks have been proposed (Chung et al., 2022; Gupta & Brandstetter, 2022; Toshev et al., 2023; Koehler et al., 2024; Liu et al., 2024b; Nathaniel et al., 2024) to evaluate the scientific ML models on PDEs (Li et al., 2021; Raissi et al., 2019; Lu et al., 2021; Li et al., 2024), fluid scenarios (Chen et al., 2025), and inverse problems (Wu et al., 2024b; Zheng et al., 2025). Each benchmark has its own focus: PDEBench (Takamoto et al., 2022) provides a vast amount of data and benchmarks across different PDEs, the Well (Ohana et al., 2024) focuses on the data quality including high resolution and large data volume, Luo et al. (2023) and Tali et al. (2024) provide a large amount of data in fluid mechanics which is governed by the complex Navier-Stokes equation. In the domain of science, some works provide real observational data (Kang et al., 2023; 2025; Lyu et al., 2024; Zhang et al., 2025) (Casey & Wintergerste, 2000), but they are not designed for machine learning, with limited data and sparse operating conditions. A concurrent work, REALM (Mao et al., 2025), proposes a benchmarking framework designed to test neural surrogates on multiphysics reactive flows in realistic regimes. In comparison, our proposed benchmark RealPDEBench contains more than 700 real-world physics experiments and numerical simulations in multiple physical systems, and we collect the paired real-world measurements and numerical data. These data are leveraged to evaluate the performance of scientific ML in real-world scenarios and their ability to transfer learning from simulation to reality.

Scientific ML has been extensively studied, and related works can be found in the following overviews (Lavin et al., 2021; Brunton & Kutz, 2022; Karniadakis et al., 2021; Cuomo et al., 2022; Wang et al., 2024a). As for the baselines considered in our work, we focus on approximating the evolution of physical system states using data-driven deep learning models. These works can be divided into two categories: small models trained on a single dataset to solve specific problems (Li et al., 2021; Wang et al., 2024b; Hu et al., 2024), and pre-trained or foundation models trained on largescale datasets (Hao et al., 2023; Ye et al., 2024; Herde et al., 2024; Feng et al., 2025b). Methods such as neural operators (Li et al., 2021) are initially used to solve specific problems in a single physical system. This type of method is widely used in various directions, such as fluids (Chen et al., 2025; Feng et al., 2025a), combustion (Weng et al., 2025), electromagnetism (Zhou et al., 2025), power systems (Huang & Wang, 2022), etc. With the increase of available data and the development of models, more and more works are dedicated to developing universal foundation models for multiple physical systems (Totounferoush et al., 2025). Our proposed RealPDEBench can be used by all the scientific ML models mentioned above, verifying their performance on real-world measurements, and providing a unique reference and data support for developing new models.

## 3 REALPDEBENCH: BRIDGING REAL-WORLD AND SIMULATED DATA

In this section, we present four key components of RealPDEBench. First, we define the tasks considered in RealPDEBench, and discuss the roles of real-world and simulated data. Next, we provide a brief description of each dataset in RealPDEBench, along with the data collection, data format, and extensibility. We then introduce the employed evaluation metrics, which can be categorized into data-oriented and physics-oriented types. Finally, we describe the baselines considered in the experiments.

### 3.1 TASK DEFINITION: ROLES OF REAL-WORLD AND SIMULATED DATA

**Prediction task.** All tasks considered in RealPDEBench fall into the category of prediction tasks, namely, predicting the future evolution of complex physical systems. More formally, we aim to learn a mapping between the input and output spaces, $F : \mathcal{A} \times \Gamma \rightarrow \mathcal{U}$, where the input is given by the Cartesian product of initial discretized states of system states $a \in \mathcal{A}$ and system parameters $\gamma \in \Gamma$, and the output $u$ corresponds to the subsequent discretized temporal evolution of the system.

**Roles of real-world and simulated data.** Real-world and simulated data each possess distinct advantages and limitations, which in turn motivate us to leverage the advantages of both. Realworld data avoid numerical errors and simplified physics but is costly, noisy, and often limited in observability. For example, the incoming flow cannot be strictly guaranteed to be uniform, and camera noise leads to measurement errors. Simulated data are relatively cheaper and offer broader modalities (variables of physical systems) with dense parameter coverage, yet suffer from numerical

errors often caused by modeling like Large Eddy Simulation (LES) and discretization like second-order convergence (Weymouth & Yue, 2011). Since the ultimate goal of scientific ML is to model real systems, *evaluation* is all conducted on real-world data, while the *training* paradigm admits multiple possible designs.

**Three categories of prediction tasks.** Based on the characteristics of the two types of data, we design three categories of tasks: real-world training, simulated training, and simulated pretraining with real-world finetuning (abbreviated as real-world finetuning in Sec. 4). For each dataset, there are $N$ real-world samples and $N$ simulated samples. $n$ real-world samples are used for training, $(N - n)/2$ for validation, and $(N - n)/2$ for testing. For all tasks, the *validation* and *test* set is fixed to be these $N - n$ real-world samples, ensuring a consistent evaluation protocol. For *training*, the three task settings are defined as follows. In *real-world training*, models are trained directly on the $n$ real-world samples. In *simulated training*, models are trained on all $N$ simulated samples. In *simulated pretraining with real-world finetuning*, models are first pretrained on the $N$ simulated samples and subsequently finetuned using the $n$ real-world samples. This setup reflects common practical scenarios, where real-world data are scarce, and simulated data are abundant.

## 3.2 OVERVIEW OF DATASETS AND PDEs

In order to comprehensively evaluate the ability of scientific ML in the real world, an ideal benchmark is expected to include a series of key physical challenges, such as transition to turbulence (Bloor, 1964), prediction of control response (He et al., 2000), nonlinear multi-physics coupling (Athani et al., 2025), three-dimensional effects (Spalart & Venkatakrishnan, 2016), and physical processes involving chemical reactions (Cathonnet, 2003). Following these principles, we carefully select and construct five representative scenarios (Huang et al., 2023; Kumar et al., 2016; Lin et al., 2024; Shukla et al., 2024; Liu et al., 2024a) for RealPDEBench. Specifically, the scenarios include the classical transition to turbulence at wake (Cylinder), controlled system (Controlled Cylinder), fluid-structure interaction (FSI), three-dimensional effects of fluid dynamics (Foil), and reactive flows (Combustion). There are a total of 736 trajectories of real-world and simulated data. The governing equations of the above systems range from the Navier-Stokes equations for basic cylinder flows to coupled FSI equations, and reactive Navier-Stokes equations with species transport for combustion systems. Developing surrogate models that approximate these real-world challenges with high fidelity across different parameter regimes remains a significant challenge (Conti et al., 2024). We argue that achieving this capability is a necessary precondition to applying such models for real-world problems. While these problems have been studied in prior work on simulated data (Takamoto et al., 2022; Ohana et al., 2024; Luo et al., 2023), a comprehensive real-world benchmark dataset spanning complex physical systems is, to the best of our knowledge, not available.

In the following parts, we provide a brief introduction and important features of each dataset. More details are provided in Appendix B.

**Cylinder** represents a fundamental benchmark problem in fluid dynamics, featuring wake dynamics behind circular cylinders across various Reynolds numbers (Re) from laminar to turbulent regimes. This dataset captures the classical Kármán vortex street formation and provides both simulation and real-world measurements for studying unsteady flow phenomena.

**Controlled Cylinder** extends the basic cylinder flow by introducing active control through external forcing, spanning different Reynolds numbers and control sequences (periodic sinusoidal control at different frequencies), which highlights the challenges of learning control-forced fluid dynamics.

**Fluid-structure Interaction (FSI)** captures phenomena where the circular cylinder undergoes structural vibrations due to fluid forces, representing critical coupling dynamics encountered in real-world applications such as bridges under wind loading and offshore platforms in ocean currents. Parameters include different Re, mass ratios, and damping coefficients, covering lock-in phenomena (Zhang et al., 2015) and galloping instabilities (Sun et al., 2020) across various configurations.

**Foil** contains cross-sectional data extracted from 3D simulations and experiments. The 3D effect introduces increased fluid complexity, generating enhanced small-scale vortex structures. Our dataset covers diverse angles-of-attack (aoa) and Reynolds numbers, which is particularly valuable for foil design and fluid dynamics optimization in marine engineering applications.

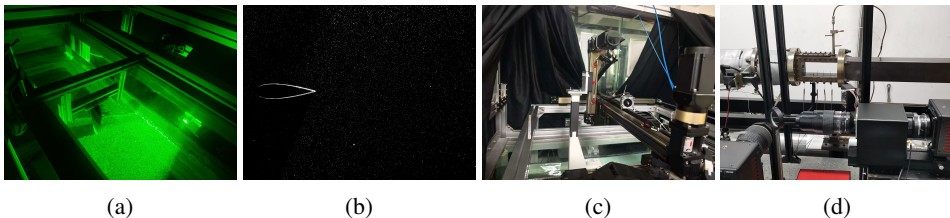

| (a) | (b) | (c) | (d) |

Figure 2: Photos of real-world data collection. (a) Water tunnel after laser irradiation. (b) Particle imaging photos taken by the camera. (c) Motion control equipment. (d) Swirl combustor equipment.

**Combustion** presents experimental and numerical cross-sectional data of 3D swirl-stabilized $NH_3$/$CH_4$/air flames. Combustion involves complex multi-field coupling problems, such as the pressure field, temperature field, sound field, velocity field, concentration field, etc. Accurately predicting the evolution of concentration, temperature, and other fields caused by combustion is of great significance to efficiently control the thermoacoustic instability for aircraft and space engines.

**Collection of real-world data.** Real-world data of fluid systems are measured in the circulating water tunnels using the Particle Image Velocimetry (PIV) technique (Abdulmouti, 2021). The circulating water tunnel is employed to generate an incoming flow that interacts with internal structures, thereby inducing complex fluid–structure interactions and associated physical phenomena. Meanwhile, in order to measure the changes in the flow field, we sprinkle fluorescent particles (hollow-glass microspheres, 10um) into the water and irradiate with a continuous laser (forming a 2mm thick laser layer on the shooting surface), thereby observing the velocity of the fluid through the velocities of the fluorescent particles, as shown in Figure 2a. These velocities are recorded using high-speed cameras (Figure 2b). Afterwards, we use PIVLab to process the photos and obtain the flow velocity states at each time step. In addition, the structure is installed on a rail driven by a stepper motor. Through the motor, we can control the movement of the structure, as shown in Figure 2c. Next, we measure the real-world data of combustion using the flame chemiluminescence (CL) imaging technique (Alviso et al., 2017). Specifically, we construct a swirl combustor and use mass flow controllers to control the injection ratio of air, $NH_3$, and $CH_4$ (Figure 2d). These components are ignited in the combustor, and then we measure the light intensity using an OH* CL camera. More details are provided in Appendix B.3.

**Collection of simulated data.** Furthermore, we use Computational Fluid Dynamics (CFD) to generate the simulated data, which have the same parameters as real-world experiments. Specifically, we generate simulated data for four fluid systems using the Finite Volume Method (FVM) and Immersed Boundary Method (IBM) (Weymouth & Yue, 2010). For 2D experiments, we apply Lilypad (Weymouth & Yue, 2011) as the solver, and for 3D simulations, we apply Waterlily (Weymouth & Font, 2025) on GPUs, which is an efficient 3D CFD solver. Next, for the Combustion dataset, we conduct a three-dimensional, implicit, unsteady LES to calculate thermoacoustic instabilities in a swirl-stabilized flame. The Eddy Dissipation Concept (EDC) (Lourier et al., 2017) is employed to model the turbulence-chemistry interaction.

**Data format and extensibility.** All datasets are stored in `HDF5` (Folk et al., 2011) format. Each file contains one trajectory sampled at equal time intervals on a uniform spatial grid, represented as a `NumPy` (van der Walt et al., 2011) array with $C$ modalities of shape $(T, X, Y)$. It also records the corresponding system parameters, such as Re, oscillation frequency, and equivalence ratio. Our codebase is implemented on the `PyTorch` (Paszke et al., 2019) platform. All datasets are wrapped under a common module `RealDataset`, and baselines are implemented under module `Model`, which enables straightforward extensibility to new datasets and baselines (Appendix C).

### 3.3 OVERVIEW OF METRICS

We propose a comprehensive set of nine evaluation metrics for the three tasks, which can be categorized into data-oriented and physics-oriented metrics. Below, we denote $\{y_k\}_{k=1}^K$ as ground truth samples, and $\{\hat{y}_k\}_{k=1}^K$ as model's predictions. We additionally denote temporal coordinates for each sample as $\{t\}_{t=1,...,T}$, and spatial coordinates as $\{x_i\}_{i=1,...,I}$.

### 3.3.1 DATA-ORIENTED METRICS

**RMSE, MAE, Rel $L_2$.** We first employ several standard metrics in ML, including Root Mean Squared Error (RMSE), Mean Absolute Error (MAE), and Relative $L_2$ Error (Rel $L_2$).

**Coefficient of determination ($R^2$).** In addition, we consider the coefficient of determination ($R^2$), which measures how well the observed outcomes are replicated by the model, defined by the proportion of variance in the ground truth explained by the predictions, as

$$R^2 = 1 - \frac{\sum_k (\boldsymbol{y}_k - \hat{\boldsymbol{y}}_k)^2}{\sum_k (\boldsymbol{y}_k - \bar{\boldsymbol{y}})^2},$$

where $\bar{\cdot}$ means the average and $\bar{\boldsymbol{y}} = \sum_k \boldsymbol{y}_k / K$.

**Update Ratio.** Finally, the Update Ratio measures the relative efficiency of simulated pretraining with real-world finetuning versus real-world training from scratch. Let $\text{RMSE}_0$ denote the best RMSE achieved with real-world training. Define $N_1$ and $N_2$ as the number of finetuning and training updates required to reach $\text{RMSE}_0$, respectively. The metric is then given by $N_1/N_2$.

### 3.3.2 PHYSICS-ORIENTED METRICS

From the physical perspective, we adopt three types of evaluation metrics.

**Fourier Space Error (fRMSE).** First, we consider the Fourier Space Error (fRMSE), which is a frequency-domain metric for evaluating prediction accuracy in spectral space (Takamoto et al., 2022). It is computed by applying a 3D Fast Fourier Transform (FFT) to both prediction and target fields, taking squared differences of Fourier coefficients, grouping them by frequency magnitude, and averaging within each group. To provide a more fine-grained analysis, we report the Fourier error at different frequency bands (low, middle, and high) by partitioning the spectrum.

**Frequency Error (FE).** Second, since lots of physical systems exhibit periodicity, accurately capturing their temporal cycles is of critical importance. To further evaluate temporal dynamics, we compute the Frequency Error (FE). This is achieved by first summing the predicted and ground-truth fields over spatial dimensions to obtain temporal signals, applying a 1D FFT, and then measuring the MAE between the two spectra (Feng et al., 2024):

$$\text{FE} = \frac{1}{KT} \sum_{k,t} \left| \mathcal{F}\left( \sum_i \boldsymbol{y}_k(t, x_i) \right) - \mathcal{F}\left( \sum_i \hat{\boldsymbol{y}}_k(t, x_i) \right) \right|.$$

**Kinetic Energy Error (KE).** The Kinetic Energy Error (KE) is a metric applied to the velocity field (Wang et al., 2020), which is measured as

$$\text{KE} = |e - \hat{e}|, \quad e = \frac{\overline{(\boldsymbol{u}')^2} + \overline{(\boldsymbol{v}')^2}}{2}, \quad \overline{(\boldsymbol{u}')^2} = \frac{1}{T} \sum_t (\boldsymbol{u}(t) - \bar{\boldsymbol{u}})^2,$$

where $\boldsymbol{u}$ and $\boldsymbol{v}$, two channels of $\boldsymbol{y}$, denote the velocity field in the $x$- and $y$-directions, respectively.

**Mean Velocity Profile Error (MVPE).** Finally, in many numerical studies of fluid literature (Kravchenko & Moin, 2000; Ma et al., 2000; Wissink & Rodi, 2008; Neunaber et al., 2025), the mean velocity profile (MVP) is a generally employed summary statistic to measure discrepancies between simulated data and real-world data. For example, there is a significant decrease of $u$ near the wake region behind the cylinder. We select different probes positioned at $\{(x_{\text{probe},j}, y_{\text{probe},j})\}_{j=1}^{N_{\text{probe}}}$ to calculate the differences between the time-average velocity field of real-world data, simulated data, and model predictions, forming the MVPE:

$$\text{MVPE} = \frac{1}{KN_{\text{probe}}} \sum_{k,j} |\bar{u}(x_{\text{probe},k,j}, y_{\text{probe},k,j}) - \bar{\hat{u}}(x_{\text{probe},k,j}, y_{\text{probe},k,j})|.$$

We note that this metric evaluates the long-term performance, so we adopt it for autoregressive evaluation (Sec. 4.5) and the comparison between simulated and real-world data (Appendix B.4).

### 3.4 BASELINE MODELS

**DMD** (Kutz et al., 2016) is a reduced order model that extracts spatiotemporal coherent structures from time-series flow field data through a data-driven matrix decomposition approach. **U-Net** (Ronneberger et al., 2015) is an auto-encoding architecture that propagates information efficiently at

different scales. **CNO** (Raonic et al., 2023) is a CNN modification designed to maintain structural consistency across continuous-discrete mappings and achieve artifact-free operator approximation. **DeepONet** (Lu et al., 2021) is a deep operator network using a branch-trunk architecture to approximate nonlinear operators by learning function-to-function mappings. **FNO** (Li et al., 2021) is a neural operator based on the Fourier transform that solves PDEs by learning operator mappings in the frequency domain, which has the advantages of resolution invariance. **WDNO** (Hu et al., 2025) applies diffusion-based generation within the wavelet space to model trajectories, effectively capturing abrupt changes. **MWT** (Gupta et al., 2021) compresses the associated operator's kernel using fine-grained wavelets and learns the projection of the kernel onto fixed multiwavelet polynomial bases through explicit embedding of inverse multiwavelet filters. **GK-Transformer** (Cao, 2021) is an attention-based neural operator that modifies the Transformer's self-attention mechanism by removing softmax normalization and incorporating Galerkin-type projections to improve approximation capacity and efficiency in learning PDEs solution operators. **Transolver** (Wu et al., 2024a) introduces attention-based learning of physical states, enabling the model to achieve intrinsic geometry-independent capacity while improving the modeling of physical correlations. **DPOT** (Hao et al., 2024) is an auto-regressive denoising operator transformer for large-scale PDE pre-training. We consider its pretrained small model (30M) and the pretrained large model (509M) in experiments, and finetune them on our datasets. Please refer to Appendix E for more details.

## 4 EXPERIMENTS

### 4.1 EXPERIMENT SETUP

In this section, we conduct experiments on the tasks and datasets introduced in Sec. 3.1 and Sec. 3.2. The train, validation, and test data are split at the parameter level. We report the RMSE, Relative $L_2$ Error, fRMSE, and Update Ratio of all datasets and baselines under the three training categories, as summarized in Table 1. The results for all other metrics are in Appendix A.1. To better accommodate the differing characteristics of real-world and simulated data, we tailor simulated training with two strategies, adding noise and randomly masking unmeasured modalities (see Sec. 4.2). During evaluation, to additionally assess the models' ability for long-term prediction, we include an option for autoregressive evaluation, with details in Sec. 4.5.

Please note that our results mainly compare baselines on real-world data, a setting that differs from the majority of their prior applications. Below, we will discuss some common findings from experiments. The code and data are available here.

### 4.2 GAP OF REAL-WORLD AND SIMULATION

We first aim to demonstrate the gap between simulated data and real-world data, which is also one of the core motivations for proposing RealPDEBench. We demonstrate these gaps from three aspects: different modalities, the errors of simulated training and real-world training, and the measurement and numerical errors. First, the modalities of simulated and real-world data are different. Due to the limitations of measurement techniques, simulated data usually contain more modalities than real-world data (as shown in Figure 1). To better leverage these additional modalities, we randomly mask unmeasured modalities with a certain probability when training with simulated data (details in Appendix D). Second, we train all baselines separately on simulated and real-world data, and test them on the same real-world test dataset (as mentioned in Sec. 3.1). In Table 1, there are significant differences between the models trained on the two types of data, and the real-world trainings have from 9.39% to 78.91% improvements on Rel $L_2$. Models trained on simulated data are difficult to generalize directly to real-world data, even if their physical parameters are consistent. Third, both real-world data and simulated data have errors. Real-world data generally have significant noise due to measurement technology limitations, as shown in Figure 1, while simulated data have numerical errors, which may be caused by simplified physical processes or ideal conditions. Therefore, to improve the generalization of models trained on simulated data, we add additional noise to the simulated data to approximate the distribution of real-world data. In addition, Figure 3a shows that simulated training yields much higher Frequency Errors than real-world training, highlighting that simulated data cannot perfectly capture periodicity in real-world systems.

Table 1: Results of RMSE, Rel $L_2$, fRMSE, and Update Ratio. Different datasets have different colors. Because DMD lacks the training process, we place its inference results in the last column and leave the rest blank. The smaller the error result, the darker the color. The **bolded** is the best.

| Dataset | Baseline | Params | Simulated Training | | | Real-world Training | | | Real-world Finetuning | | | |
|---|---|---|---|---|---|---|---|---|---|---|---|---|
| | | | RMSE | Rel $L_2$ | fRMSE | RMSE | Rel $L_2$ | fRMSE | RMSE | Rel $L_2$ | fRMSE | Ratio |
| Cylinder | U-Net | 23.0 M | 0.0758 | 0.2165 | 0.0122 | 0.0700 | **0.0701** | 0.0103 | 0.0632 | **0.0728** | 0.0097 | 0.3636 |
| | CNO | 8.0 M | 0.0729 | 0.1849 | 0.0113 | 0.0424 | 0.0897 | 0.0069 | 0.0403 | 0.1013 | 0.0066 | 1.0000 |
| | DeepONet | 3.6 M | 0.0863 | 0.3592 | 0.0132 | 0.0713 | 0.1528 | 0.0108 | 0.0661 | 0.1503 | 0.0107 | 0.5758 |
| | FNO | 50.4 M | 0.0739 | 0.2575 | 0.0115 | 0.0585 | 0.0855 | 0.0087 | 0.0545 | 0.0780 | 0.0087 | **0.1111** |
| | WDNO | 104.3 M | 0.0699 | 0.2053 | 0.0105 | 0.0689 | 0.1666 | 0.0091 | 0.0513 | 0.0969 | 0.0073 | 0.1250 |
| | MWT | 2.9 M | 0.0733 | 0.2240 | 0.0107 | 0.0540 | 0.0832 | 0.0081 | 0.0584 | 0.0850 | 0.0088 | 0.9253 |
| | GK-Transformer | 84.4 M | 0.0876 | 0.2941 | 0.0139 | 0.1196 | 0.0898 | 0.0166 | 0.0995 | 0.0908 | 0.0146 | 0.4400 |
| | Transolver | 4.3 M | 0.1121 | 0.3224 | 0.0174 | 0.1093 | 0.1887 | 0.0160 | 0.0965 | 0.1723 | 0.0144 | 1.0000 |
| | DPOT-S-FT | 30.8 M | 0.0513 | 0.1474 | 0.0076 | 0.0586 | 0.0983 | 0.0086 | 0.0440 | 0.0766 | 0.0067 | 0.1250 |
| | DPOT-L-FT | 649.8 M | **0.0486** | **0.1446** | **0.0070** | **0.0390** | 0.0812 | **0.0056** | **0.0394** | 0.0733 | **0.0059** | 1.0000 |
| | ML Average | - | 0.0752 | 0.2356 | 0.0115 | 0.0692 | 0.1106 | 0.0101 | 0.0613 | 0.0997 | 0.0093 | 0.5666 |
| | DMD | - | - | - | - | - | - | - | 0.0862 | 0.3590 | 0.0114 | - |
| Controlled Cylinder | U-Net | 23.0 M | 0.0195 | 0.1360 | 0.0029 | **0.0080** | **0.0555** | 0.0010 | **0.0079** | **0.0543** | **0.0009** | 0.7632 |
| | CNO | 8.0 M | **0.0167** | **0.1209** | **0.0022** | 0.0081 | 0.0583 | **0.0009** | 0.0080 | 0.0574 | **0.0009** | 0.8400 |
| | DeepONet | 3.6 M | 0.0540 | 0.4256 | 0.0134 | 0.0309 | 0.2399 | 0.0058 | 0.0291 | 0.2284 | 0.0052 | 0.5200 |
| | FNO | 50.4 M | 0.0285 | 0.2007 | 0.0059 | 0.0097 | 0.0723 | 0.0012 | 0.0094 | 0.0702 | 0.0011 | 0.5217 |
| | WDNO | 359.8 M | 0.0240 | 0.1829 | 0.0037 | 0.0115 | 0.0927 | 0.0014 | 0.0101 | 0.0752 | 0.0013 | **0.3617** |
| | MWT | 2.9 M | 0.0220 | 0.1594 | 0.0037 | 0.0102 | 0.0747 | 0.0011 | 0.0102 | 0.0746 | 0.0010 | 0.5487 |
| | GK-Transformer | 50.8 M | 0.0261 | 0.1816 | 0.0050 | 0.0103 | 0.0767 | 0.0012 | 0.0101 | 0.0748 | 0.0012 | 0.5200 |
| | Transolver | 4.3 M | 0.0294 | 0.1894 | 0.0069 | 0.0171 | 0.1200 | 0.0024 | 0.0168 | 0.1176 | 0.0022 | 0.4285 |
| | DPOT-S-FT | 30.8 M | 0.0233 | 0.1722 | 0.0039 | 0.0084 | 0.0598 | 0.0010 | 0.0085 | 0.0615 | 0.0010 | 1.0000 |
| | DPOT-L-FT | 649.8 M | 0.0248 | 0.1784 | 0.0043 | 0.0084 | 0.0603 | 0.0010 | 0.0085 | 0.0611 | 0.0010 | 1.0000 |
| | ML Average | - | 0.0268 | 0.1947 | 0.0052 | 0.0123 | 0.0910 | 0.0017 | 0.0119 | 0.0875 | 0.0016 | 0.6504 |
| | DMD | - | - | - | - | - | - | - | 0.0340 | 0.2233 | 0.0059 | - |
| FSI | U-Net | 23.0 M | **0.0223** | **0.1589** | **0.0025** | **0.0085** | **0.0583** | 0.0007 | **0.0084** | 0.0579 | 0.0007 | 0.6667 |
| | CNO | 8.0 M | 0.0241 | 0.1724 | 0.0029 | 0.0105 | 0.0741 | 0.0009 | 0.0096 | 0.0679 | 0.0008 | 0.5600 |
| | DeepONet | 3.4 M | 0.0637 | 0.4606 | 0.0097 | 0.0350 | 0.2502 | 0.0051 | 0.0332 | 0.2368 | 0.0048 | 0.3125 |
| | FNO | 268.5 M | 0.0426 | 0.3095 | 0.0059 | 0.0129 | 0.0892 | 0.0012 | 0.0127 | 0.0881 | 0.0012 | 0.5714 |
| | WDNO | 91.7 M | 0.0369 | 0.2700 | 0.0050 | 0.0117 | 0.0817 | 0.0011 | 0.0116 | 0.0824 | 0.0011 | 0.5116 |
| | MWT | 2.9 M | 0.0339 | 0.2487 | 0.0046 | 0.0128 | 0.0910 | 0.0010 | 0.0128 | 0.0912 | 0.0010 | 0.9992 |
| | GK-Transformer | 67.6 M | 0.0307 | 0.2241 | 0.0039 | 0.0127 | 0.0916 | 0.0011 | 0.0124 | 0.0898 | 0.0010 | 0.6200 |
| | Transolver | 4.3 M | 0.0305 | 0.2119 | 0.0043 | 0.0236 | 0.1567 | 0.0027 | 0.0223 | 0.1480 | 0.0025 | **0.1600** |
| | DPOT-S-FT | 41.3 M | 0.0260 | 0.1886 | 0.0031 | 0.0105 | 0.0746 | 0.0008 | 0.0099 | 0.0701 | **0.0007** | 0.2500 |
| | DPOT-L-FT | 673.5 M | 0.0262 | 0.1898 | 0.0030 | 0.0099 | 0.0687 | 0.0008 | 0.0097 | 0.0668 | 0.0008 | 0.3125 |
| | ML Average | - | 0.0337 | 0.2434 | 0.0045 | 0.0148 | 0.1036 | 0.0015 | 0.0143 | 0.0999 | 0.0015 | 0.4964 |
| | DMD | - | - | - | - | - | - | - | 0.0450 | 0.2955 | 0.0060 | - |
| Foil | U-Net | 23.0 M | 0.0272 | 0.0745 | 0.0039 | **0.0100** | **0.0159** | 0.0011 | **0.0094** | **0.0145** | **0.0008** | 0.5116 |
| | CNO | 8.0 M | 0.0227 | 0.0438 | 0.0028 | 0.0136 | 0.0253 | 0.0018 | 0.0114 | 0.0206 | 0.0013 | 0.4400 |
| | DeepONet | 3.6 M | 0.0339 | 0.0565 | 0.0051 | 0.0222 | 0.0363 | 0.0028 | 0.0226 | 0.0375 | 0.0029 | 0.5185 |
| | FNO | 50.4 M | 0.0274 | 0.0540 | 0.0037 | 0.0130 | 0.0228 | 0.0015 | 0.0120 | 0.0206 | 0.0012 | **0.3192** |
| | WDNO | 358.4 M | 0.0242 | 0.0490 | **0.0026** | 0.0162 | 0.0444 | 0.0018 | 0.0106 | 0.0181 | 0.0010 | 0.1035 |
| | MWT | 2.9 M | 0.0264 | 0.0492 | 0.0036 | 0.0133 | 0.0227 | 0.0015 | 0.0125 | 0.0210 | 0.0012 | 0.4894 |
| | GK-Transformer | 50.8 M | 0.0277 | 0.0512 | 0.0038 | 0.0142 | 0.0245 | 0.0018 | 0.0138 | 0.0237 | 0.0017 | 0.5918 |
| | Transolver | 4.3 M | 0.0345 | 0.0465 | 0.0050 | 0.0220 | 0.0370 | 0.0027 | 0.0138 | 0.0237 | 0.0017 | 0.5918 |
| | DPOT-S-FT | 41.3 M | **0.0221** | **0.0397** | 0.0027 | 0.0106 | 0.0166 | **0.0010** | 0.0109 | 0.0174 | 0.0012 | 1.0000 |
| | DPOT-L-FT | 673.5 M | 0.0229 | 0.0402 | 0.0029 | 0.0105 | **0.0159** | 0.0011 | 0.0109 | 0.0161 | 0.0012 | 1.0000 |
| | ML Average | - | 0.0269 | 0.0505 | 0.0036 | 0.0146 | 0.0261 | 0.0017 | 0.0128 | 0.0213 | 0.0014 | 0.5566 |
| | DMD | - | - | - | - | - | - | - | 0.0322 | 0.0520 | 0.0041 | - |
| Combustion | U-Net | 23.3 M | **0.0358** | **0.7290** | **0.0051** | 0.0216 | 0.5487 | 0.0026 | 0.0213 | 0.5403 | 0.0025 | 0.5682 |
| | CNO | 8.0 M | 0.0401 | 0.8274 | 0.0057 | 0.0248 | 0.6030 | 0.0032 | 0.0238 | 0.5877 | 0.0030 | 0.7083 |
| | DeepONet | 3.5 M | 0.0403 | 0.8565 | 0.0056 | 0.0229 | 0.5751 | 0.0028 | 0.0227 | 0.5723 | 0.0028 | 0.7800 |
| | FNO | 67.1 M | 0.0363 | 0.7606 | 0.0052 | 0.0226 | 0.5664 | 0.0027 | 0.0225 | 0.5680 | 0.0027 | 0.4286 |
| | WDNO | 122.7 M | 0.0439 | 1.1016 | 0.0061 | 0.0380 | 0.8605 | 0.0055 | 0.0314 | 0.7441 | 0.0044 | **0.2632** |
| | MWT | 2.9 M | 0.0367 | 0.7536 | 0.0052 | 0.0221 | 0.5560 | 0.0027 | 0.0220 | 0.5549 | 0.0026 | 0.9982 |
| | GK-Transformer | 67.6 M | 0.0400 | 0.9143 | 0.0056 | 0.0247 | 0.6083 | 0.0031 | 0.0258 | 0.6421 | 0.0034 | 1.0000 |
| | Transolver | 4.3 M | 0.0442 | 0.9056 | 0.0061 | 0.0375 | 0.7825 | 0.0050 | 0.0375 | 0.7836 | 0.0050 | 1.0000 |
| | DPOT-S-FT | 41.5 M | 0.0393 | 0.7842 | 0.0054 | 0.0209 | 0.5349 | **0.0024** | 0.0211 | 0.5378 | 0.0024 | 1.0000 |
| | DPOT-L-FT | 674.2 M | 0.0378 | 0.7750 | 0.0053 | **0.0208** | **0.5331** | **0.0024** | **0.0206** | **0.5318** | **0.0023** | 0.8125 |
| | ML Average | - | 0.0394 | 0.8408 | 0.0055 | 0.0256 | 0.6169 | 0.0032 | 0.0249 | 0.6063 | 0.0031 | 0.7559 |
| | DMD | - | - | - | - | - | - | - | 0.0914 | 1.3360 | 0.0110 | - |

## 4.3 IMPROVEMENT WITH SIMULATED DATA PRETRAINING

Although the previous section has shown that there exists a gap between real-world and simulated data, simulated data possess unique advantages, including a lower cost, access to a greater number of modalities, and being free from measurement-induced noise.

This raises a key question: can simulated data enhance model performance on real-world data? Our results provide a strong affirmative answer, demonstrating clear benefits from simulated pretraining across two key aspects. First, as shown in Table 1, the errors in the Real-world Finetuning column are lower than those of Real-world Training, revealing that simulated pretraining and then real-

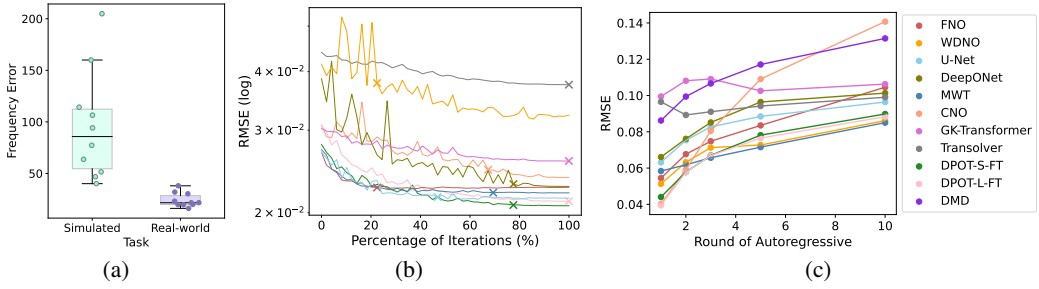

Figure 3: (a) Frequency errors of baselines (statistics of 10 values) on real-world vs. simulated data from Controlled Cylinder. (b) Validation RMSE curves of real-world finetuning on Combustion, with crosses marking the best RMSE of real-world training. The x-axis shows the percentage of update iterations. (c) RMSE under 1, 2, 3, 5, and 10 rounds of autoregressive evaluation on Cylinder.

world finetuning leads to better performance compared to training directly on the same amount of real-world data. On the one hand, this improvement can be attributed to the larger volume of simulated data, which exposes the model to trajectories under a wider range of system parameters. On the other hand, the simulated modalities that are unmeasured in real-world data have additional dynamic information, which is utilized through the mask-training strategy.

Second, the loss decreases faster during finetuning, which is evidenced by the Update Ratio metric. For most datasets and baselines, the value is less than 1, which means real-world finetuning requires fewer update iterations to reach the optimal performance of real-world training. Moreover, in Figure 3b, we plot the validation RMSE curves on the Combustion dataset. It can be clearly observed that real-world finetuning achieves a much faster decrease compared to real-world training, further confirming the benefit of simulated data in improving model performance.

## 4.4 COMPARISON OF BASELINES

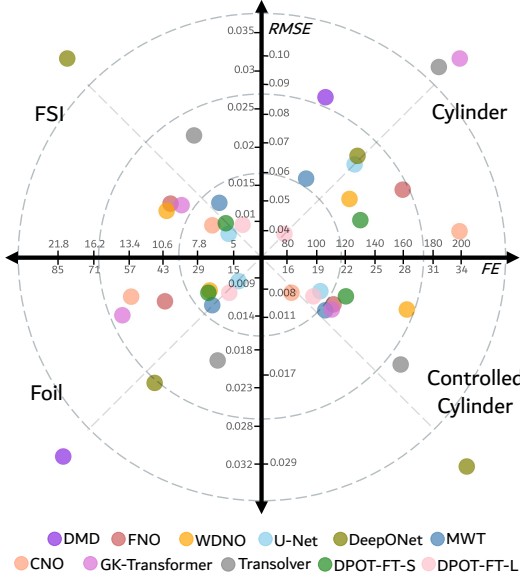

Figure 4: Trade-off of RMSE and Frequency Error. Distant points are linked with dashed lines. We omit the points where the DMD error is far from the center to make most of the points clearer.

Different scientific ML models have distinct advantages. In this subsection, we analyze and discuss the performance of baselines in terms of the trade-off between data-oriented metrics reflecting local pixel-level errors and physics-oriented metrics reflecting global features. Specifically, we select RMSE from the data-oriented metrics and Frequency Error from the physics-oriented metrics, which can reflect the model's ability to capture global periodicity. In order to compare the results on multiple datasets, we integrate them into Figure 4, where the x-axis is Frequency Error and the y-axis is RMSE. Each quadrant is a dataset. Proximity to the origin indicates better performance.

Figure 4 reveals three interesting phenomena. First, DPOT-L-FT, the large pretrained foundation model, achieves the best overall performance since it is closest to the origin. This reflects the benefits of large-scale PDE pretraining and a larger number of model parameters. Second, since most models are trained with data-oriented losses (*e.g.*, MSE), they excel at local features while their performance on physics-oriented metrics involving global features is weaker. Especially, convolution-based methods (U-Net, CNO) achieve lower RMSE, as the tasks resemble image processing, where convolution has proven to be highly effective (Albawi et al., 2017). Third, due to the use of the multiwavelet transform, in general, MWT exhibits advantages in learning periodicity. Therefore, it is

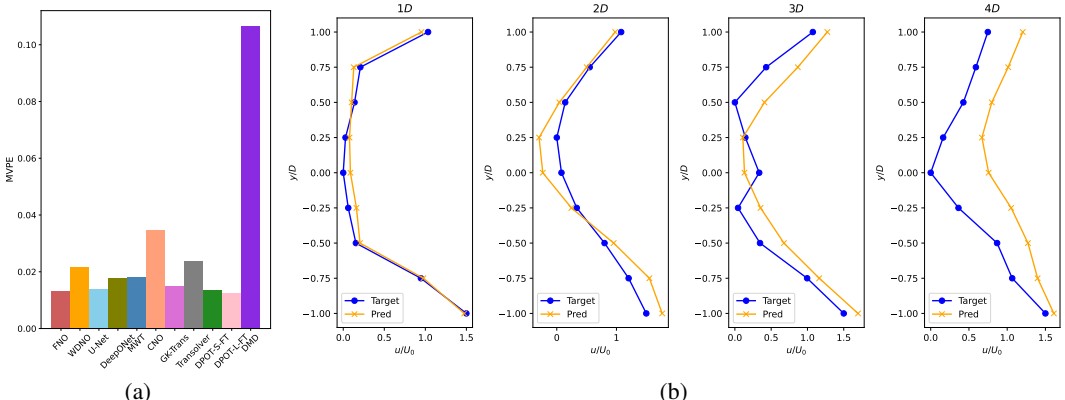

Figure 5: (a) MVPEs of real-world finetuning under 10-round autoregressive evaluation on Cylinder. (b) MVP of U-Net's 10-round autoregressive prediction on Cylinder.

important to select the appropriate network architecture and training strategy based on the different objectives of the specific tasks and data.

### 4.5 OTHER ANALYSIS

**Autoregressive evaluation.** To investigate the long-term performance, we conduct evaluations with 1, 2, 3, 5, and 10 rounds of autoregressive prediction. Given training inputs and outputs of $T$ time steps, autoregressive evaluation with $N$ rounds iteratively feeds each predicted $T$ steps back as input, yielding predictions over $NT$ steps. The prediction error is finally measured over the entire $NT$ steps. Figure 3c shows results on the Cylinder dataset. While CNO performs well in single-round prediction, its error grows at a faster rate than other methods as the number of autoregressive rounds increases. This suggests that it may suffer from substantial error accumulation, leading to inaccurate long-term predictions. In addition, on the 10-round evaluation, we calculate MVPEs of baselines and visualize the MVP of U-Net in Figure 5, which reveals the long-term summary statistic. From the results, we observe that DMD shows limitations under this metric, while the large-size DPOT model performs the best. More results on other datasets and metrics are provided in Appendix A.2.

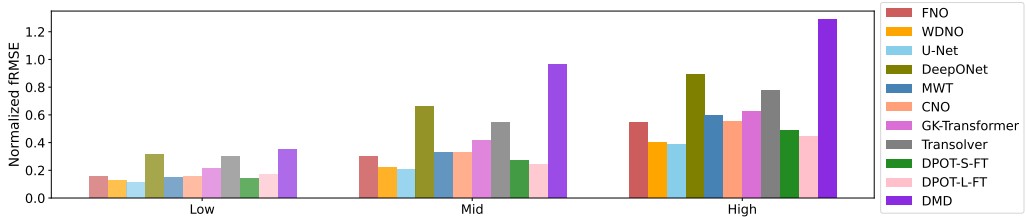

Figure 6: Normalized low-, mid- and high-frequency fRMSE on Foil.

**Low-, mid-, and high-frequency fRMSE.** We compute the normalized fRMSE within low-, mid-, and high-frequency bands, which reflects the model's ability to capture system dynamics at different frequency ranges. As shown in Figure 6, on the Foil dataset, CNO's relative performance compared to other models increases as the frequency gets higher. This phenomenon observed in CNO may be related to its design principle of eliminating aliasing errors. Full results are in Appendix A.3.

## 5 CONCLUSION

In this work, to address the critical issue of lacking real-world data, we introduce RealPDEBench, the first real-world scientific ML benchmark for complex physical system prediction. It incorporates paired real-world and simulated data, three categories of tasks, nine evaluation metrics, and ten baselines. Our work underscores the differences between real-world and simulated data and takes a significant step toward bridging the gap. For limitations, see Appendix F. We hope our benchmark and datasets will spur the development of algorithms that more effectively integrate real-world and simulated data, paving the way for scientific ML that is truly applicable in real-world settings.

## ACKNOWLEDGEMENT

We thank Jiashu Pan, Long Wei, Zhe Yuan, and Tianci Bu for providing feedback on our manuscript. We also thank Chunyu Liu for discussions. We gratefully acknowledge the support of Westlake University Research Center for Industries of the Future; Westlake University Center for High-performance Computing. The content is solely the responsibility of the authors and does not necessarily represent the official views of the funding entities.

## ETHICS STATEMENT

This work proposes a benchmark for scientific Machine Learning using real-world and simulated data. All datasets are collected from physical experiments and numerical solvers, without involving sensitive information. Caution that models trained on this benchmark should not be directly applied in safety-critical scenarios without further validation.

## REPRODUCIBILITY STATEMENT

The code, data, checkpoints, and log files are available at `https://github.com/AI4Science-WestlakeU/RealPDEBench`, with full documentation to ensure transparency and reproducibility. We provide a unified and modular code framework, together with scripts for reproducing all experiments. We also provide numerical scripts, calibration files, processing software, and part of the raw data for validation of the accuracy of data acquisition and processing.

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

## A  OTHER EXPERIMENT RESULTS

In this section, we provide the experiment results that are not in the main text due to the page limit.

### A.1  OTHER METRICS

In addition to the results reported in Table 1 of the main text, our main experiments are also evaluated under other metrics, including MAE, $R^2$, KE, and FE as mentioned in Sec. 3.3. The results for these additional metrics are summarized in Table 2. We note that the KE is not applied to the Combustion dataset, because the modality in this dataset is the light intensity. Besides, the velocity field is not a key variable in this setting. From the table, overall, we find that U-Net and DPOT-L-FT perform relatively well on these metrics, showing their strong capability of modeling complex physical systems. Besides, the Combustion dataset is more challenging. For instance, the $R^2$ metric on this dataset is significantly lower than on the others.

Table 2: **Results of other metrics.** The **bolded** values are the best.

| Dataset | Baseline | Simulated Training | | | | Real-world Training | | | | Real-world Finetuning | | | |
|---|---|---|---|---|---|---|---|---|---|---|---|---|---|
| | | MAE↓ | $R^2$↑ | KE↓ | FE↓ | MAE↓ | $R^2$↑ | KE↓ | FE↓ | MAE↓ | $R^2$↑ | KE↓ | FE↓ |
| Cylinder | U-Net | 0.03591 | 0.62955 | **0.00040** | 195.31436 | 0.01266 | 0.68363 | **0.00027** | 141.57477 | 0.01213 | 0.74257 | **0.00027** | 128.03439 |
| | CNO | 0.02909 | 0.71633 | 0.00050 | 124.15546 | 0.01208 | **0.90384** | 0.00062 | 144.89873 | 0.01369 | **0.91341** | 0.00051 | 201.88423 |
| | DeepONet | 0.04357 | 0.51881 | 0.00050 | 176.41255 | 0.02025 | 0.67225 | 0.00040 | 150.94789 | 0.01912 | 0.71811 | 0.00049 | 130.10420 |
| | FNO | 0.03040 | 0.64762 | 0.00048 | 244.28989 | 0.01277 | 0.77924 | 0.00046 | 124.86349 | 0.01105 | 0.80835 | 0.00063 | 161.91525 |
| | WDNO | 0.02762 | 0.68467 | 0.00079 | 191.33083 | 0.02190 | 0.69337 | 0.00133 | 150.92130 | 0.01396 | 0.83004 | 0.00054 | 124.44437 |
| | MWT | 0.02454 | 0.71324 | 0.00047 | **105.17693** | 0.01155 | 0.84425 | 0.00039 | **70.63197** | 0.01232 | 0.81807 | 0.00039 | 93.87933 |
| | GK-Transformer | 0.04164 | 0.59035 | 0.00056 | 247.20154 | 0.02099 | 0.23596 | 0.00032 | 220.35785 | 0.01887 | 0.47159 | 0.00038 | 202.11179 |
| | Transolver | 0.04706 | 0.32880 | 0.00050 | 504.92477 | 0.02861 | 0.36250 | 0.00047 | 236.56488 | 0.02556 | 0.50232 | 0.00045 | 187.44112 |
| | DPOT-S-FT | 0.01847 | 0.83044 | 0.00049 | 137.41856 | 0.01440 | 0.77873 | 0.00070 | 180.77551 | 0.01056 | 0.87481 | 0.00060 | 132.15703 |
| | DPOT-L-FT | **0.01749** | **0.84745** | 0.00070 | 107.45474 | **0.01010** | 0.90178 | 0.00036 | 90.55615 | **0.00936** | 0.89962 | 0.00044 | **78.74014** |
| | DMD | - | - | - | - | - | - | - | - | 0.03116 | 0.52011 | 0.00050 | 107.47345 |
| Controlled Cylinder | U-Net | 0.01050 | 0.92443 | **0.00005** | 46.81991 | **0.00385** | 0.98747 | 0.00003 | 23.11901 | 0.00372 | 0.98774 | 0.00003 | 18.91348 |
| | CNO | **0.00895** | **0.94480** | 0.00005 | 40.10512 | 0.00406 | 0.98693 | 0.00003 | **16.14166** | 0.00397 | 0.98722 | 0.00003 | **15.58394** |
| | DeepONet | 0.03755 | 0.42267 | 0.00007 | 159.99414 | 0.01967 | 0.81072 | 0.00006 | 38.02571 | 0.01850 | 0.83181 | 0.00006 | 35.81494 |
| | FNO | 0.01756 | 0.83962 | 0.00007 | 204.88274 | 0.00535 | 0.98136 | 0.00004 | 21.59613 | 0.00516 | 0.98238 | 0.00004 | 20.42206 |
| | WDNO | 0.01494 | 0.88544 | 0.00009 | 77.30387 | 0.00685 | 0.97388 | 0.00005 | 30.30858 | 0.00547 | 0.97991 | 0.00004 | 28.88806 |
| | MWT | 0.01304 | 0.90383 | **0.00005** | 63.71152 | 0.00558 | 0.97946 | 0.00004 | 19.60898 | 0.00558 | 0.97952 | 0.00004 | 19.47143 |
| | GK-Transformer | 0.01536 | 0.86483 | 0.00006 | 114.27576 | 0.00565 | 0.97879 | 0.00004 | 21.63828 | 0.00549 | 0.97988 | 0.00004 | 20.21486 |
| | Transolver | 0.01796 | 0.82884 | 0.00006 | 106.50475 | 0.00951 | 0.94187 | 0.00005 | 31.97471 | 0.00925 | 0.94420 | 0.00005 | 28.16750 |
| | DPOT-S-FT | 0.01355 | 0.89218 | **0.00005** | 94.21255 | 0.00425 | 0.98588 | **0.00003** | 19.98549 | 0.00440 | 0.98578 | **0.00003** | 21.86629 |
| | DPOT-L-FT | 0.01418 | 0.87813 | 0.00006 | 51.53077 | 0.00427 | 0.98594 | **0.00003** | 19.87323 | 0.00430 | 0.98573 | **0.00003** | 18.11246 |
| | DMD | - | - | - | - | - | - | - | - | 0.01794 | 0.77102 | 0.00007 | 44.73851 |
| FSI | U-Net | **0.01279** | **0.91558** | 0.00008 | **11.32213** | **0.00454** | 0.98779 | 0.00004 | 7.39539 | 0.00449 | 0.98798 | 0.00004 | 6.18526 |
| | CNO | 0.01408 | 0.90154 | 0.00008 | 24.27576 | 0.00626 | 0.98135 | 0.00005 | 9.73771 | 0.00565 | 0.98433 | 0.00005 | 7.76694 |
| | DeepONet | 0.04329 | 0.30952 | 0.00010 | 55.68024 | 0.02138 | 0.79215 | 0.00009 | 23.34537 | 0.02029 | 0.81301 | 0.00009 | 21.75345 |
| | FNO | 0.02784 | 0.69079 | 0.00010 | 37.22523 | 0.00758 | 0.97150 | 0.00006 | 12.70306 | 0.00744 | 0.97272 | 0.00006 | 11.78857 |
| | WDNO | 0.02435 | 0.76781 | 0.00019 | 45.80850 | 0.00677 | 0.97688 | 0.00006 | 9.92727 | 0.00697 | 0.97700 | 0.00006 | 12.17233 |
| | MWT | 0.02071 | 0.79785 | **0.00008** | 19.96735 | 0.00759 | 0.97140 | 0.00006 | 7.53941 | 0.00758 | 0.97136 | 0.00006 | 7.09237 |
| | GK-Transformer | 0.01878 | 0.83438 | 0.00013 | 24.86648 | 0.00759 | 0.97175 | 0.00006 | 14.14439 | 0.00736 | 0.97281 | 0.00006 | 10.72866 |
| | Transolver | 0.01806 | 0.83691 | 0.00011 | 22.74508 | 0.01278 | 0.90214 | 0.00009 | 13.07140 | 0.01204 | 0.91273 | 0.00009 | 9.52701 |
| | DPOT-S-FT | 0.01573 | 0.88539 | 0.00009 | 70.25787 | 0.00615 | 0.98129 | 0.00005 | 7.11340 | 0.00575 | 0.98332 | 0.00005 | 6.48055 |
| | DPOT-L-FT | 0.01543 | 0.88336 | **0.00008** | 31.36328 | 0.00555 | 0.98334 | 0.00005 | **5.01643** | 0.00540 | 0.98412 | 0.00005 | **4.86275** |
| | DMD | - | - | - | - | - | - | - | - | 0.02423 | 0.65550 | 0.00010 | 30.34143 |
| Foil | U-Net | 0.01334 | 0.93131 | 0.00016 | 205.27599 | 0.00321 | **0.99073** | 0.00007 | 22.60979 | **0.00277** | 0.99186 | 0.00007 | **14.63533** |
| | CNO | 0.00829 | 0.95236 | **0.00015** | 94.85928 | 0.00552 | 0.98276 | 0.00011 | 76.37526 | 0.00426 | 0.98784 | 0.00008 | 57.35152 |
| | DeepONet | 0.01305 | 0.89360 | 0.00018 | 146.62540 | 0.00680 | 0.95437 | 0.00019 | 43.25816 | 0.00704 | 0.95272 | 0.00019 | 47.95818 |
| | FNO | 0.01108 | 0.93037 | 0.00017 | 140.33791 | 0.00483 | 0.98428 | 0.00009 | 52.49263 | 0.00404 | 0.98664 | 0.00009 | 43.82851 |
| | WDNO | 0.00999 | 0.94567 | 0.00023 | **43.29331** | 0.00842 | 0.97564 | 0.00012 | 94.06609 | 0.00344 | 0.98951 | 0.00008 | 26.05156 |
| | MWT | 0.01009 | 0.93540 | **0.00015** | 136.97717 | 0.00449 | 0.98368 | 0.00009 | 30.65537 | 0.01245 | 0.98560 | 0.00009 | 25.09887 |
| | GK-Transformer | 0.01069 | 0.92888 | 0.00016 | 186.29233 | 0.00489 | 0.98127 | 0.00010 | 58.32386 | 0.00473 | 0.98244 | 0.00009 | 60.83815 |
| | Transolver | 0.01020 | 0.88917 | **0.00015** | 63.22823 | 0.00684 | 0.95497 | 0.00016 | 45.88408 | 0.00567 | 0.96413 | 0.00015 | 22.92277 |
| | DPOT-S-FT | **0.00769** | **0.95464** | 0.00016 | 92.92261 | 0.00325 | 0.98964 | 0.00008 | 15.31028 | 0.00351 | 0.98890 | 0.00008 | 26.61732 |
| | DPOT-L-FT | 0.00777 | 0.95150 | 0.00016 | 54.12466 | **0.00310** | 0.98978 | 0.00008 | **15.25477** | 0.00324 | 0.98898 | 0.00008 | 18.24177 |
| | DMD | - | - | - | - | - | - | - | - | 0.00937 | 0.90388 | 0.00019 | 84.34200 |
| Combustion | U-Net | **0.00934** | **0.29106** | - | **84.45078** | 0.00575 | 0.74085 | - | 26.02853 | 0.00542 | 0.74736 | - | 25.05470 |
| | CNO | 0.01107 | 0.11123 | - | 107.27361 | 0.00682 | 0.65906 | - | 38.39980 | 0.00654 | 0.68804 | - | 34.42250 |
| | DeepONet | 0.01171 | 0.10024 | - | 105.85174 | 0.00617 | 0.70817 | - | 29.43881 | 0.00602 | 0.71397 | - | 29.82702 |
| | FNO | 0.00994 | 0.26832 | - | 87.34087 | 0.00586 | 0.71771 | - | 30.63720 | 0.00597 | 0.72008 | - | 30.53438 |
| | WDNO | 0.01707 | -0.06953 | - | 134.92741 | 0.01087 | 0.19841 | - | 90.55799 | 0.00888 | 0.45328 | - | 57.61599 |
| | MWT | 0.00994 | 0.26942 | - | 85.93394 | 0.00562 | 0.73053 | - | 28.44920 | 0.00566 | 0.73148 | - | 27.92226 |
| | GK-Transformer | 0.01319 | 0.11814 | - | 112.22723 | 0.00699 | 0.66380 | - | 38.43433 | 0.00747 | 0.63269 | - | 43.30125 |
| | Transolver | 0.01387 | -0.07878 | - | 120.68479 | 0.01172 | 0.22220 | - | 76.67628 | 0.01170 | 0.22252 | - | 75.68771 |
| | DPOT-S-FT | 0.01096 | 0.14192 | - | 104.63753 | 0.00583 | 0.75845 | - | **23.07150** | 0.00584 | 0.75402 | - | 24.17408 |
| | DPOT-L-FT | 0.01030 | 0.20883 | - | 92.19399 | **0.00537** | **0.76120** | - | 23.25353 | **0.00535** | **0.76467** | - | **22.73871** |
| | DMD | - | - | - | - | - | - | - | - | 0.02081 | -3.63692 | - | 120.96538 |

### A.2  AUTOREGRESSIVE EVALUATION

Below, we provide the full result of autoregressive evaluation in Table 3 (2 rounds of autoregressive evaluation), Table 4 (3 rounds of autoregressive evaluation), and Table 5 (1, 2, and 3 rounds of

autoregressive evaluation under RMSE, relative $L_2$ error, and fRMSE). From the results, several observations can be made. First, the Controlled Cylinder and Foil datasets exhibit relatively stable performance, with models such as U-Net maintaining high accuracy, suggesting that their underlying dynamics are more favorable for stable long-horizon prediction. In contrast, the Combustion dataset remains the most challenging: all methods achieve significantly lower $R^2$ values, and errors increase more sharply. Second, U-Net variants consistently achieve superior results across most metrics and datasets, highlighting their robustness in capturing spatiotemporal dynamics. Transformer-based approaches (GK-Transformer, Transolver) show competitive short-term performance but degrade faster with increasing horizons. Finally, Frequency Errors increase markedly from two to three autoregressive steps, indicating that the error of capturing temporal dynamics grows rapidly over time.

Table 3: **2 rounds of autoregressive evaluation.** The **bolded** values are the best.

| Dataset | Baseline | RMSE ↓ | MAE ↓ | Rel $L_2$ ↓ | $R^2$ ↑ | KE ↓ | fRMSE ↓ | FE ↓ |
|---|---|---|---|---|---|---|---|---|
| Cylinder | U-Net | 0.07535 | 0.01478 | **0.09430** | 0.63360 | **0.00070** | 0.00645 | 198.26685 |
| | CNO | 0.05911 | 0.01962 | 0.15177 | 0.77450 | 0.00145 | 0.00597 | 479.91360 |
| | DeepONet | 0.07612 | 0.02123 | 0.16087 | 0.62603 | 0.00115 | 0.00685 | 214.22360 |
| | FNO | 0.06765 | 0.01362 | 0.09777 | 0.70467 | 0.00134 | 0.00632 | 281.23196 |
| | WDNO | 0.06319 | 0.01768 | 0.12782 | 0.74232 | 0.00091 | 0.00566 | 337.25354 |
| | MWT | 0.06178 | 0.01341 | 0.10118 | 0.75365 | 0.00079 | 0.00541 | 139.07365 |
| | GK-Transformer | 0.10814 | 0.02063 | 0.11349 | 0.24523 | 0.00071 | 0.00821 | 241.95125 |
| | Transolver | 0.08928 | 0.02582 | 0.19561 | 0.48559 | 0.00096 | 0.00753 | 204.22385 |
| | DPOT-S-FT | 0.05778 | 0.01342 | 0.09745 | 0.78450 | 0.00096 | 0.00570 | 190.29182 |
| | DPOT-L-FT | **0.05774** | **0.01264** | 0.09439 | **0.78480** | 0.00092 | **0.00533** | **137.29283** |
| | DMD | 0.09943 | 0.03575 | 0.40840 | 0.36195 | 0.00131 | 0.00799 | 176.32237 |
| Controlled Cylinder | U-Net | **0.01083** | **0.00523** | **0.07585** | **0.97676** | **0.00008** | **0.00130** | 34.18222 |
| | CNO | 0.01126 | 0.00565 | 0.08116 | 0.97485 | **0.00008** | 0.00135 | **28.88530** |
| | DeepONet | 0.03063 | 0.01974 | 0.23983 | 0.81402 | 0.00017 | 0.00477 | 103.55875 |
| | FNO | 0.01221 | 0.00656 | 0.08960 | 0.97047 | 0.00009 | 0.00146 | 34.40803 |
| | WDNO | 0.01331 | 0.00720 | 0.09932 | 0.96490 | 0.00009 | 0.00168 | 53.22416 |
| | MWT | 0.01328 | 0.00726 | 0.09763 | 0.96504 | 0.00009 | 0.00156 | 36.93626 |
| | GK-Transformer | 0.01291 | 0.00698 | 0.09513 | 0.96695 | 0.00010 | 0.00155 | 32.69817 |
| | Transolver | 0.01974 | 0.01105 | 0.14127 | 0.92278 | 0.00013 | 0.00258 | 47.72284 |
| | DPOT-S-FT | 0.01140 | 0.00590 | 0.08268 | 0.97422 | **0.00008** | 0.00138 | 38.86863 |
| | DPOT-L-FT | 0.01132 | 0.00573 | 0.08163 | 0.97461 | **0.00008** | 0.00137 | 34.77075 |
| | DMD | 0.04111 | 0.02186 | 0.27235 | 0.66506 | 0.00021 | 0.00619 | 75.94245 |
| FSI | U-Net | **0.01379** | **0.00685** | 0.09313 | **0.96760** | 0.00014 | **0.00119** | 12.71709 |
| | CNO | 0.01527 | 0.00829 | 0.10587 | 0.96032 | 0.00014 | 0.00131 | 21.32656 |
| | DeepONet | 0.03472 | 0.02122 | 0.24746 | 0.79477 | 0.00025 | 0.00345 | 49.31825 |
| | FNO | 0.01714 | 0.00944 | 0.11648 | 0.94998 | 0.00016 | 0.00155 | 23.81794 |
| | WDNO | 0.01756 | 0.00972 | 0.12205 | 0.94749 | 0.00016 | 0.00157 | 28.00188 |
| | MWT | 0.01817 | 0.00999 | 0.12837 | 0.94375 | 0.00016 | 0.00151 | 15.51482 |
| | GK-Transformer | 0.01683 | 0.00930 | 0.11695 | 0.95178 | 0.00016 | 0.00148 | 21.17584 |
| | Transolver | 0.02358 | 0.01278 | 0.15668 | 0.90214 | **0.00009** | 0.00267 | 13.07140 |
| | DPOT-S-FT | 0.01476 | 0.00781 | 0.10131 | 0.96289 | 0.00015 | 0.00125 | 13.45711 |
| | DPOT-L-FT | 0.01440 | 0.00736 | 0.09714 | 0.96467 | 0.00014 | 0.00123 | **10.97794** |
| | DMD | 0.05420 | 0.02875 | 0.35644 | 0.49863 | 0.00030 | 0.00524 | 55.02239 |
| Foil | U-Net | **0.01228** | **0.00333** | **0.01905** | **0.98601** | **0.00012** | **0.00080** | **23.86791** |
| | CNO | 0.01503 | 0.00535 | 0.02667 | 0.97904 | 0.00016 | 0.00122 | 107.73207 |
| | DeepONet | 0.05055 | 0.01461 | 0.06368 | 0.76275 | 0.00095 | 0.00401 | 523.23944 |
| | FNO | 0.01582 | 0.00493 | 0.02709 | 0.97677 | 0.00016 | 0.00109 | 77.52514 |
| | WDNO | 0.01409 | 0.00414 | 0.02321 | 0.98156 | 0.00013 | 0.00089 | 44.82506 |
| | MWT | 0.01529 | 0.00454 | 0.02528 | 0.97829 | 0.00015 | 0.00104 | 38.69154 |
| | GK-Transformer | 0.01663 | 0.00555 | 0.02851 | 0.97432 | 0.00015 | 0.00135 | 104.44638 |
| | Transolver | 0.02074 | 0.00616 | 0.03521 | 0.96008 | 0.00025 | 0.00144 | 46.71509 |
| | DPOT-S-FT | 0.01420 | 0.00420 | 0.02182 | 0.98127 | 0.00013 | 0.00108 | 48.70568 |
| | DPOT-L-FT | 0.01449 | 0.00397 | 0.02094 | 0.98049 | **0.00012** | 0.00112 | 40.78880 |
| | DMD | 0.03335 | 0.01051 | 0.05529 | 0.89673 | 0.00029 | 0.00282 | 156.66721 |
| Combustion | U-Net | 0.02279 | **0.00570** | 0.55826 | 0.70763 | - | 0.00204 | 44.70588 |
| | CNO | 0.02728 | 0.00733 | 0.62687 | 0.58102 | - | 0.00253 | 67.08688 |
| | DeepONet | 0.02930 | 0.00738 | 0.67170 | 0.51662 | - | 0.00278 | 88.01767 |
| | FNO | 0.02422 | 0.00633 | 0.59011 | 0.66980 | - | 0.00221 | 52.72492 |
| | WDNO | 0.03360 | 0.00936 | 0.76863 | 0.36430 | - | 0.00324 | 95.65475 |
| | MWT | 0.02346 | 0.00594 | 0.57131 | 0.69001 | - | 0.00212 | 48.42057 |
| | GK-Transformer | 0.02755 | 0.00770 | 0.65741 | 0.57253 | - | 0.00257 | 72.15970 |
| | Transolver | 0.04066 | 0.01277 | 0.82321 | 0.06895 | - | 0.00373 | 129.05060 |
| | DPOT-S-FT | 0.02262 | 0.00614 | **0.55686** | 0.71180 | - | 0.00203 | 44.36895 |
| | DPOT-L-FT | **0.02255** | 0.00577 | 0.56212 | **0.71376** | - | **0.00202** | **43.57514** |
| | DMD | 0.09502 | 0.02172 | 1.38468 | -4.08454 | - | 0.00932 | 162.85385 |

Table 4: **3 rounds of autoregressive evaluation.** The **bolded** values are the best.

| Dataset | Baseline | RMSE ↓ | MAE ↓ | Rel $L_2$ ↓ | $R^2$ ↑ | KE ↓ | fRMSE ↓ | FE ↓ |
|---|---|---|---|---|---|---|---|---|
| Cylinder | U-Net | 0.08270 | 0.01679 | 0.11133 | 0.55877 | **0.00107** | 0.00473 | 268.25345 |
| | CNO | 0.08047 | 0.02730 | 0.21099 | 0.58218 | 0.00269 | 0.00560 | 887.97833 |
| | DeepONet | 0.08515 | 0.02318 | 0.17226 | 0.53221 | 0.00165 | 0.00509 | 277.13663 |
| | FNO | 0.07479 | 0.01537 | 0.11204 | 0.63911 | 0.00181 | 0.00500 | 379.31039 |
| | WDNO | 0.07129 | 0.02093 | 0.15456 | 0.67208 | 0.00127 | 0.00450 | 558.40649 |
| | MWT | **0.06570** | 0.01511 | 0.11673 | **0.72153** | 0.00111 | **0.00404** | **196.52660** |
| | GK-Transformer | 0.10912 | 0.02197 | 0.12845 | 0.23177 | 0.00112 | 0.00566 | 284.11075 |
| | Transolver | 0.09106 | 0.02808 | 0.21982 | 0.46500 | 0.00156 | 0.00528 | 236.66180 |
| | DPOT-S-FT | 0.06700 | 0.01550 | 0.11305 | 0.71036 | 0.00128 | 0.00454 | 252.41632 |
| | DPOT-L-FT | 0.06689 | **0.01480** | **0.10939** | 0.71132 | 0.00120 | 0.00425 | 199.98730 |
| | DMD | 0.10664 | 0.03774 | 0.42629 | 0.26768 | 0.00210 | 0.00607 | 239.53703 |
| Controlled Cylinder | U-Net | **0.01289** | **0.00636** | **0.09091** | **0.96705** | 0.00013 | **0.00134** | 49.11861 |
| | CNO | 0.01380 | 0.00711 | 0.10075 | 0.96221 | 0.00014 | 0.00144 | **45.14964** |
| | DeepONet | 0.03241 | 0.02116 | 0.25293 | 0.79177 | 0.00029 | 0.00383 | 193.48720 |
| | FNO | 0.01424 | 0.00769 | 0.10474 | 0.95981 | 0.00014 | 0.00148 | 47.96572 |
| | WDNO | 0.01532 | 0.00833 | 0.11428 | 0.95348 | 0.00014 | 0.00164 | 73.24361 |
| | MWT | 0.01595 | 0.00885 | 0.11881 | 0.94953 | 0.00015 | 0.00166 | 58.24154 |
| | GK-Transformer | 0.01505 | 0.00824 | 0.11178 | 0.95507 | 0.00015 | 0.00157 | 45.33165 |
| | Transolver | 0.02236 | 0.01267 | 0.16144 | 0.90090 | 0.00021 | 0.00243 | 70.27728 |
| | DPOT-S-FT | 0.01344 | 0.00704 | 0.09799 | 0.96416 | **0.00013** | 0.00141 | 55.72421 |
| | DPOT-L-FT | 0.01326 | 0.00681 | 0.09643 | 0.96514 | **0.00013** | 0.00140 | 50.39636 |
| | DMD | 0.04714 | 0.02534 | 0.31516 | 0.55938 | 0.00036 | 0.00545 | 109.54674 |
| FSI | U-Net | **0.01767** | **0.00863** | **0.11893** | **0.94680** | 0.00023 | **0.00123** | 19.91612 |
| | CNO | 0.02237 | 0.01188 | 0.15529 | 0.91469 | 0.00027 | 0.00151 | 51.93961 |
| | DeepONet | 0.03638 | 0.02221 | 0.25920 | 0.77434 | 0.00042 | 0.00261 | 76.82671 |
| | FNO | 0.02078 | 0.01117 | 0.14110 | 0.92638 | 0.00026 | 0.00149 | 35.43581 |
| | WDNO | 0.02150 | 0.01164 | 0.14855 | 0.92120 | 0.00026 | 0.00153 | 44.13584 |
| | MWT | 0.02907 | 0.01424 | 0.21189 | 0.85591 | 0.00045 | 0.00178 | 40.14235 |
| | GK-Transformer | 0.02064 | 0.01115 | 0.14217 | 0.92740 | 0.00027 | 0.00145 | 32.12624 |
| | Transolver | 0.02358 | 0.01278 | 0.15668 | 0.90214 | **0.00009** | 0.00267 | **13.07140** |
| | DPOT-S-FT | 0.01869 | 0.00954 | 0.12678 | 0.94046 | 0.00026 | 0.00130 | 21.02419 |
| | DPOT-L-FT | 0.01795 | 0.00891 | 0.12045 | 0.94505 | 0.00023 | 0.00125 | 17.38784 |
| | DMD | 0.06251 | 0.03289 | 0.41228 | 0.33119 | 0.00052 | 0.00458 | 82.20554 |
| Foil | U-Net | **0.01458** | **0.00377** | **0.02269** | **0.98026** | **0.00014** | **0.00068** | **31.53317** |
| | CNO | 0.01816 | 0.00641 | 0.03241 | 0.96939 | 0.00021 | 0.00108 | 169.92812 |
| | DeepONet | 0.07452 | 0.02080 | 0.08767 | 0.48453 | 0.00171 | 0.00412 | 896.86194 |
| | FNO | 0.01867 | 0.00560 | 0.03144 | 0.96765 | 0.00018 | 0.00089 | 111.04512 |
| | WDNO | 0.01674 | 0.00463 | 0.02691 | 0.97399 | **0.00014** | 0.00075 | 59.70494 |
| | MWT | 0.01812 | 0.00504 | 0.02961 | 0.96952 | 0.00017 | 0.00085 | 52.03396 |
| | GK-Transformer | 0.01874 | 0.00617 | 0.03166 | 0.96739 | 0.00019 | 0.00110 | 143.23317 |
| | Transolver | 0.02180 | 0.00665 | 0.03694 | 0.95588 | 0.00030 | 0.00114 | 69.33317 |
| | DPOT-S-FT | 0.01690 | 0.00477 | 0.02530 | 0.97348 | 0.00016 | 0.00093 | 69.95742 |
| | DPOT-L-FT | 0.01740 | 0.00455 | 0.02452 | 0.97189 | 0.00015 | 0.00096 | 61.23933 |
| | DMD | 0.03469 | 0.01169 | 0.05916 | 0.88825 | 0.00033 | 0.00226 | 238.98497 |
| Combustion | U-Net | **0.02418** | **0.00599** | **0.58134** | **0.67051** | - | **0.00184** | **62.51865** |
| | CNO | 0.03039 | 0.00830 | 0.67417 | 0.47944 | - | 0.00232 | 95.54838 |
| | DeepONet | 0.03434 | 0.00910 | 0.97732 | 0.33515 | - | 0.00262 | 146.21243 |
| | FNO | 0.02616 | 0.00675 | 0.62085 | 0.61428 | - | 0.00200 | 73.07064 |
| | WDNO | 0.03559 | 0.00982 | 0.79687 | 0.28603 | - | 0.00277 | 122.88408 |
| | MWT | 0.02521 | 0.00631 | 0.59812 | 0.64181 | - | 0.00192 | 67.06616 |
| | GK-Transformer | 0.02935 | 0.00804 | 0.68410 | 0.51450 | - | 0.00223 | 94.56139 |
| | Transolver | 0.04187 | 0.01332 | 0.84313 | 0.01181 | - | 0.00307 | 148.69182 |
| | DPOT-S-FT | 0.02421 | 0.00648 | 0.58350 | 0.66966 | - | 0.00185 | 63.20166 |
| | DPOT-L-FT | 0.02454 | 0.00622 | 0.59435 | 0.66066 | - | 0.00187 | 64.32231 |
| | DMD | 0.09817 | 0.02263 | 1.42943 | -4.43148 | - | 0.00871 | 172.48363 |

Table 5: **1, 2, and 3 rounds of autoregressive evaluation under RMSE, relative $L_2$ error, and fRMSE.** The **bolded** values are the best.

| Dataset | Baseline | 1 Round | | | 2 Round | | | 3 Round | | |
|---|---|---|---|---|---|---|---|---|---|---|
| | | RMSE ↓ | Rel $L_2$ ↓ | fRMSE ↓ | RMSE ↓ | Rel $L_2$ ↓ | fRMSE ↓ | RMSE ↓ | Rel $L_2$ ↓ | fRMSE ↓ |
| Cylinder | U-Net | 0.0632 | **0.0728** | 0.0097 | 0.0754 | **0.0943** | 0.0065 | 0.0827 | 0.1113 | 0.0047 |
| | CNO | 0.0403 | 0.1013 | 0.0066 | 0.0591 | 0.1518 | 0.0060 | 0.0805 | 0.2110 | 0.0056 |
| | DeepONet | 0.0661 | 0.1503 | 0.0107 | 0.0761 | 0.1609 | 0.0069 | 0.0852 | 0.1723 | 0.0051 |
| | FNO | 0.0545 | 0.0780 | 0.0087 | 0.0677 | 0.0978 | 0.0063 | 0.0748 | 0.1120 | 0.0050 |
| | WDNO | 0.0513 | 0.0969 | 0.0073 | 0.0632 | 0.1278 | 0.0057 | 0.0713 | 0.1546 | 0.0045 |
| | MWT | 0.0584 | 0.0850 | 0.0088 | 0.0618 | 0.1012 | 0.0054 | **0.0657** | 0.1167 | **0.0040** |
| | GK-Transformer | 0.0995 | 0.0908 | 0.0146 | 0.1081 | 0.1135 | 0.0082 | 0.1091 | 0.1285 | 0.0057 |
| | Transolver | 0.0965 | 0.1723 | 0.0144 | 0.0893 | 0.1956 | 0.0075 | 0.0911 | 0.2198 | 0.0053 |
| | DPOT-S-FT | 0.0440 | 0.0766 | 0.0067 | 0.0578 | 0.0975 | 0.0057 | 0.0670 | 0.2198 | 0.0053 |
| | DPOT-L-FT | **0.0394** | 0.0733 | **0.0059** | **0.0577** | 0.0944 | **0.0053** | 0.0669 | **0.1094** | 0.0043 |
| | DMD | 0.0862 | 0.3590 | 0.0114 | 0.0994 | 0.4084 | 0.0080 | 0.1066 | 0.4263 | 0.0061 |
| Controlled Cylinder | U-Net | **0.0079** | **0.0543** | **0.0009** | **0.0108** | **0.0759** | 0.0013 | **0.0129** | **0.0909** | **0.0013** |
| | CNO | 0.0080 | 0.0574 | **0.0009** | 0.0113 | 0.0812 | 0.0014 | 0.0138 | 0.1008 | 0.0014 |
| | DeepONet | 0.0291 | 0.2284 | 0.0052 | 0.0306 | 0.2398 | 0.0048 | 0.0324 | 0.2529 | 0.0038 |
| | FNO | 0.0094 | 0.0702 | 0.0011 | 0.0122 | 0.0896 | 0.0015 | 0.0142 | 0.1047 | 0.0015 |
| | WDNO | 0.0101 | 0.0752 | 0.0013 | 0.0133 | 0.0993 | 0.0017 | 0.0153 | 0.1143 | 0.0016 |
| | MWT | 0.0101 | 0.0752 | 0.0013 | 0.0133 | 0.0976 | 0.0016 | 0.0160 | 0.1188 | 0.0017 |
| | GK-Transformer | 0.0101 | 0.0748 | 0.0012 | 0.0129 | 0.0951 | 0.0016 | 0.0151 | 0.1118 | 0.0016 |
| | Transolver | 0.0168 | 0.1176 | 0.0022 | 0.0197 | 0.1413 | 0.0026 | 0.0224 | 0.1614 | 0.0024 |
| | DPOT-S-FT | 0.0085 | 0.0615 | 0.0010 | 0.0114 | 0.0827 | 0.0014 | 0.0134 | 0.0980 | 0.0014 |
| | DPOT-L-FT | 0.0085 | 0.0611 | 0.0010 | 0.0113 | 0.0816 | 0.0014 | 0.0133 | 0.0964 | 0.0014 |
| | DMD | 0.0340 | 0.2233 | 0.0059 | 0.0411 | 0.2724 | 0.0062 | 0.0471 | 0.3152 | 0.0055 |
| FSI | U-Net | **0.0084** | **0.0579** | **0.0007** | **0.0138** | **0.0931** | 0.0012 | **0.0177** | **0.1189** | 0.0012 |
| | CNO | 0.0096 | 0.0679 | 0.0008 | 0.0153 | 0.1059 | 0.0013 | 0.0224 | 0.1553 | 0.0015 |
| | DeepONet | 0.0332 | 0.2368 | 0.0048 | 0.0347 | 0.2475 | 0.0035 | 0.0364 | 0.2592 | 0.0026 |
| | FNO | 0.0127 | 0.0881 | 0.0012 | 0.0171 | 0.1165 | 0.0016 | 0.0208 | 0.1411 | 0.0015 |
| | WDNO | 0.0116 | 0.0824 | 0.0011 | 0.0176 | 0.1221 | 0.0016 | 0.0215 | 0.1486 | 0.0015 |
| | MWT | 0.0128 | 0.0912 | 0.0010 | 0.0182 | 0.1284 | 0.0015 | 0.0291 | 0.2119 | 0.0018 |
| | GK-Transformer | 0.0124 | 0.0898 | 0.0010 | 0.0168 | 0.1170 | 0.0015 | 0.0206 | 0.1422 | 0.0015 |
| | Transolver | 0.0223 | 0.1480 | 0.0025 | 0.0236 | 0.1567 | 0.0027 | 0.0236 | 0.1567 | 0.0027 |
| | DPOT-S-FT | 0.0099 | 0.0701 | **0.0007** | 0.0148 | 0.1013 | 0.0013 | 0.0187 | 0.1268 | 0.0013 |
| | DPOT-L-FT | 0.0097 | 0.0668 | 0.0008 | 0.0144 | 0.0971 | **0.0012** | 0.0180 | 0.1205 | 0.0013 |
| | DMD | 0.0450 | 0.2955 | 0.0060 | 0.0542 | 0.3564 | 0.0052 | 0.0625 | 0.4123 | 0.0046 |
| Foil | U-Net | **0.0094** | **0.0145** | **0.0008** | **0.0123** | **0.0191** | **0.0008** | **0.0146** | **0.0227** | **0.0007** |
| | CNO | 0.0114 | 0.0206 | 0.0013 | 0.0150 | 0.0267 | 0.0012 | 0.0182 | 0.0324 | 0.0011 |
| | DeepONet | 0.0226 | 0.0375 | 0.0029 | 0.0506 | 0.0637 | 0.0040 | 0.0745 | 0.0877 | 0.0041 |
| | FNO | 0.0120 | 0.0206 | 0.0012 | 0.0158 | 0.0271 | 0.0011 | 0.0187 | 0.0314 | 0.0009 |
| | WDNO | 0.0106 | 0.0181 | 0.0010 | 0.0141 | 0.0232 | 0.0009 | 0.0167 | 0.0269 | 0.0008 |
| | MWT | 0.0125 | 0.0210 | 0.0012 | 0.0153 | 0.0253 | 0.0010 | 0.0181 | 0.0296 | 0.0009 |
| | GK-Transformer | 0.0138 | 0.0237 | 0.0017 | 0.0166 | 0.0285 | 0.0014 | 0.0187 | 0.0317 | 0.0011 |
| | Transolver | 0.0138 | 0.0237 | 0.0017 | 0.0207 | 0.0352 | 0.0014 | 0.0218 | 0.0369 | 0.0011 |
| | DPOT-S-FT | 0.0109 | 0.0174 | 0.0012 | 0.0142 | 0.0218 | 0.0011 | 0.0169 | 0.0253 | 0.0009 |
| | DPOT-L-FT | 0.0109 | 0.0161 | 0.0012 | 0.0145 | 0.0209 | 0.0011 | 0.0174 | 0.0245 | 0.0010 |
| | DMD | 0.0322 | 0.0520 | 0.0041 | 0.0334 | 0.0553 | 0.0028 | 0.0347 | 0.0592 | 0.0023 |
| Combustion | U-Net | 0.0213 | 0.5403 | 0.0025 | 0.0228 | 0.5583 | **0.0020** | 0.0242 | **0.5813** | **0.0018** |
| | CNO | 0.0238 | 0.5877 | 0.0030 | 0.0273 | 0.6269 | 0.0025 | 0.0304 | 0.6742 | 0.0023 |
| | DeepONet | 0.0227 | 0.5723 | 0.0028 | 0.0293 | 0.6717 | 0.0028 | 0.0343 | 0.9773 | 0.0026 |
| | FNO | 0.0225 | 0.5680 | 0.0027 | 0.0242 | 0.5901 | 0.0022 | 0.0262 | 0.6209 | 0.0020 |
| | WDNO | 0.0314 | 0.7441 | 0.0044 | 0.0336 | 0.7686 | 0.0032 | 0.0356 | 0.7969 | 0.0028 |
| | MWT | 0.0220 | 0.5549 | 0.0026 | 0.0235 | 0.5713 | 0.0021 | 0.0252 | 0.5981 | 0.0019 |
| | GK-Transformer | 0.0258 | 0.6421 | 0.0034 | 0.0276 | 0.6574 | 0.0026 | 0.0294 | 0.6841 | 0.0022 |
| | Transolver | 0.0375 | 0.7836 | 0.0050 | 0.0407 | 0.8232 | 0.0037 | 0.0419 | 0.8431 | 0.0031 |
| | DPOT-S-FT | 0.0211 | 0.5378 | 0.0024 | 0.0226 | **0.5569** | **0.0020** | 0.0242 | 0.5835 | 0.0019 |
| | DPOT-L-FT | **0.0206** | **0.5318** | **0.0023** | **0.0226** | 0.5621 | **0.0020** | 0.0245 | 0.5944 | 0.0019 |
| | DMD | 0.0914 | 1.3360 | 0.0110 | 0.0950 | 1.3847 | 0.0093 | 0.0982 | 1.4294 | 0.0087 |

In addition, we apply MVPE on the 10-round evaluation. The visualizations of all baselines are provided in Figure 7 and 8. From the visualizations, we observe that the velocity field $u$ is easier to capture, while the predictions of velocity $v$ are harder.

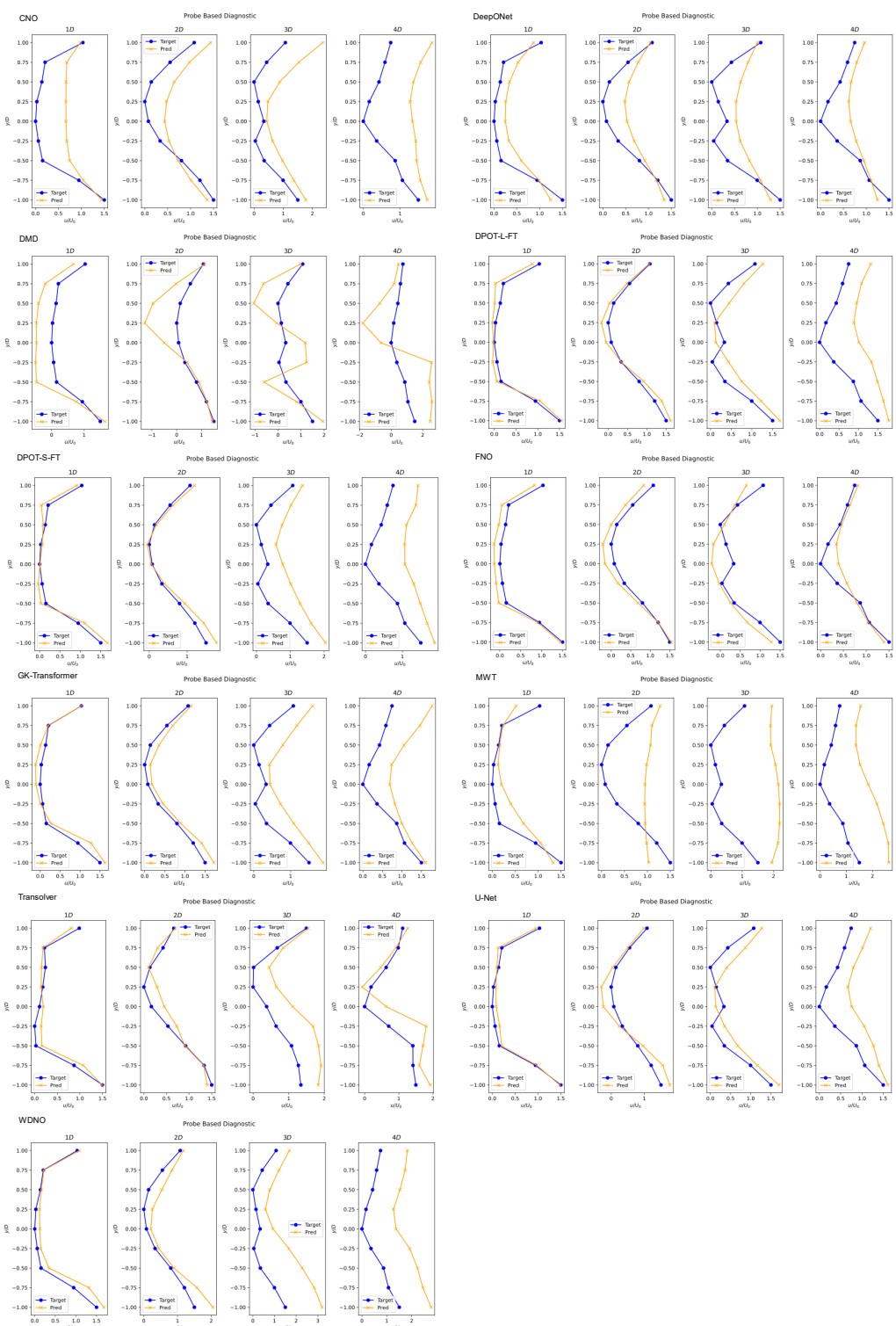

Figure 7: Mean velocity profile of $u$ predicted by baselines' 10-round autoregressive evaluation on Cylinder.

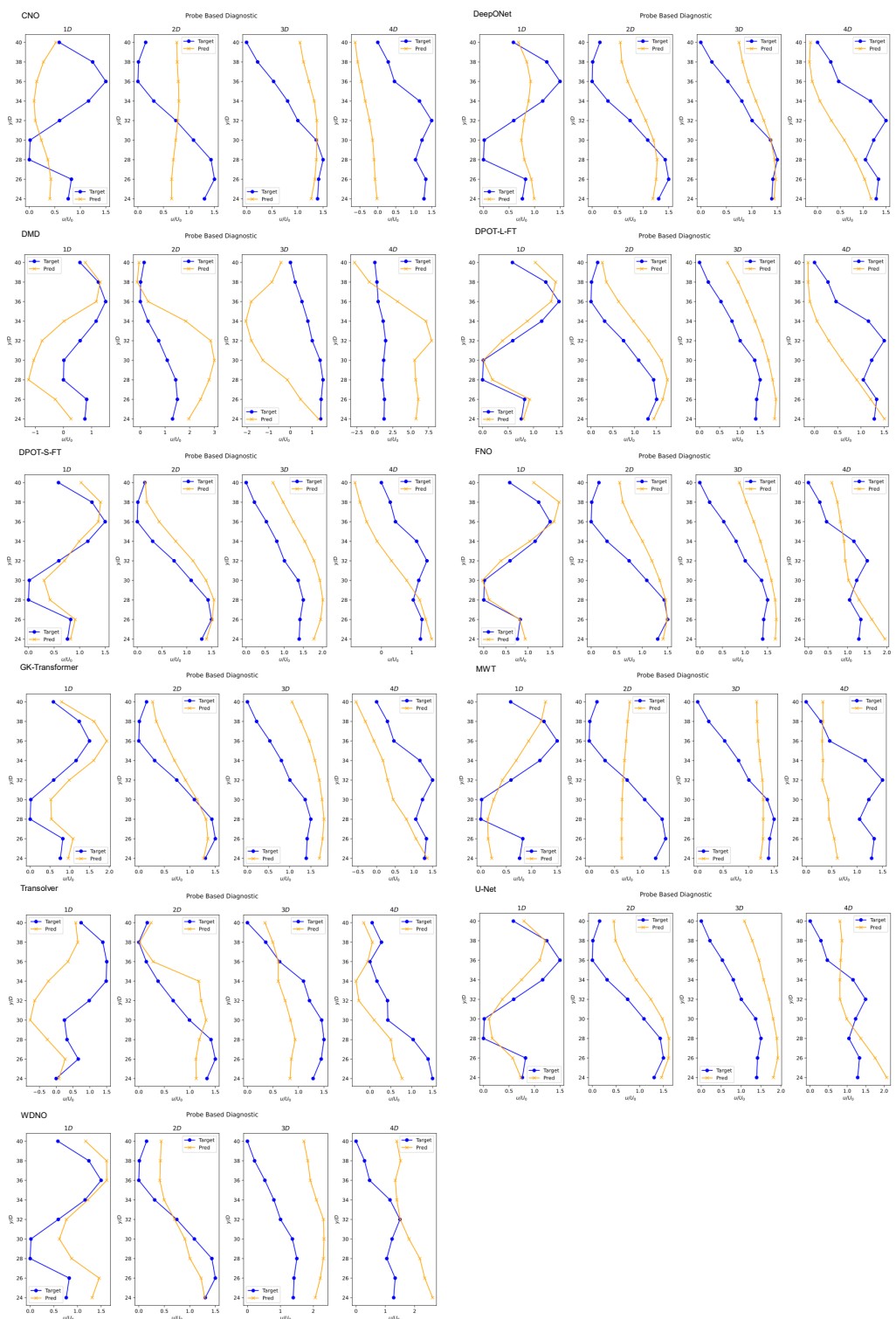

Figure 8: Mean velocity profile of $v$ predicted by baselines' 10-round autoregressive evaluation on Cylinder.

## A.3 Low-, Mid- and High-frequency fRMSE

We provide the full results of low-, mid-, and high-frequency fRMSE in Table 6, including results of all baselines on all datasets. From the results, we find that on the Controlled Cylinder dataset and the FSI dataset, although the U-Net does not perform the best under low-frequency fRSME, it outperforms under Mid- and High-frequency fRMSE. This displays that the U-Net excels at capturing the information with a higher frequency.

Table 6: **Low, mid, high frequency error.** The **bolded** values are the best.

| Dataset | Baseline | Low-fRMSE ↓ | Mid-fRMSE ↓ | High-fRMSE ↓ |
|---|---|---|---|---|
| Cylinder | U-Net | 0.01835 | 0.00774 | 0.00375 |
| | CNO | **0.00916** | 0.00626 | 0.00449 |
| | DeepONet | 0.01887 | 0.00887 | 0.00496 |
| | FNO | 0.01241 | 0.00905 | 0.00439 |
| | WDNO | 0.01203 | 0.00655 | 0.00368 |
| | MWT | 0.01145 | 0.00947 | 0.00519 |
| | GK-Transformer | 0.02855 | 0.01130 | 0.00504 |
| | Transolver | 0.02746 | 0.01146 | 0.00516 |
| | DPOT-S-FT | 0.01081 | 0.00562 | 0.00411 |
| | DPOT-L-FT | 0.00962 | **0.00509** | **0.00316** |
| | DMD | 0.01182 | 0.01455 | 0.00688 |
| Controlled Cylinder | U-Net | 0.00079 | **0.00102** | **0.00101** |
| | CNO | **0.00073** | 0.00103 | 0.00102 |
| | DeepONet | 0.00350 | 0.00642 | 0.00626 |
| | FNO | 0.00094 | 0.00120 | 0.00115 |
| | WDNO | 0.00129 | 0.00131 | 0.00128 |
| | MWT | 0.00085 | 0.00117 | 0.00116 |
| | GK-Transformer | 0.00097 | 0.00129 | 0.00126 |
| | Transolver | 0.00203 | 0.00238 | 0.00227 |
| | DPOT-S-FT | 0.00087 | 0.00108 | 0.00105 |
| | DPOT-L-FT | 0.00089 | 0.00107 | 0.00104 |
| | DMD | 0.00370 | 0.00837 | 0.00685 |
| FSI | U-Net | 0.00057 | **0.00076** | **0.00080** |
| | CNO | 0.00057 | 0.00084 | 0.00091 |
| | DeepONet | 0.00476 | 0.00540 | 0.00391 |
| | FNO | 0.00119 | 0.00117 | 0.00112 |
| | WDNO | 0.00091 | 0.00108 | 0.00115 |
| | MWT | 0.00066 | 0.00104 | 0.00129 |
| | GK-Transformer | 0.00096 | 0.00102 | 0.00104 |
| | Transolver | 0.00210 | 0.00277 | 0.00239 |
| | DPOT-S-FT | **0.00056** | 0.00079 | 0.00085 |
| | DPOT-L-FT | 0.00058 | 0.00080 | 0.00086 |
| | DMD | 0.00698 | 0.00617 | 0.00468 |
| Foil | U-Net | **0.00085** | **0.00085** | **0.00077** |
| | CNO | 0.00184 | 0.00124 | 0.00099 |
| | DeepONet | 0.00373 | 0.00299 | 0.00193 |
| | FNO | 0.00142 | 0.00122 | 0.00109 |
| | WDNO | 0.00099 | 0.00098 | 0.00086 |
| | MWT | 0.00117 | 0.00131 | 0.00119 |
| | GK-Transformer | 0.00243 | 0.00156 | 0.00123 |
| | Transolver | 0.00243 | 0.00247 | 0.00176 |
| | DPOT-S-FT | 0.00159 | 0.00106 | 0.00093 |
| | DPOT-L-FT | 0.00182 | 0.00098 | 0.00089 |
| | DMD | 0.00411 | 0.00478 | 0.00316 |
| Combustion | U-Net | 0.00257 | 0.00279 | 0.00201 |
| | CNO | 0.00344 | 0.00319 | 0.00217 |
| | DeepONet | 0.00289 | 0.00308 | 0.00216 |
| | FNO | 0.00307 | 0.00295 | 0.00209 |
| | WDNO | 0.00615 | 0.00443 | 0.00257 |
| | MWT | 0.00287 | 0.00290 | 0.00206 |
| | GK-Transformer | 0.00423 | 0.00349 | 0.00229 |
| | Transolver | 0.00716 | 0.00495 | 0.00275 |
| | DPOT-S-FT | 0.00230 | 0.00273 | 0.00208 |
| | DPOT-L-FT | **0.00229** | **0.00260** | **0.00197** |
| | DMD | 0.01526 | 0.00986 | 0.00832 |

# B    DATASET DETAILS

## B.1    DATA FORMAT

Details of individual datasets are given in Table 7. In the table, we report the number of trajectories, the number of frames, the resolution, the memory of the data, and modalities, where $u, v, w$ is the velocity field, $p$ is pressure, and $I$ is the luminescence intensity. The modalities of simulated combustion data include absolute pressure, chemistry heat release rate, mole fraction of $CH_4$, CO, $CO_2$, $H_2O$, $NH_2$, $NH_3$, and OH, temperature, $u, v, w, p$, and velocity magnitude.

Table 7: **Overview of datasets.** `n_traj` is the number of trajectories. `n_frame` is the number of frames.

| Dataset | n_traj | n_frame | $\Delta t$ (s) | Resolution Simulated | Resolution Real-world | Memory (GB) | Modalities Simulated | Modalities Real-world |
|---|---|---|---|---|---|---|---|---|
| Cylinder | $92 \times 2$ | 3990 | $2.5 \times 10^{-3}$ | $64 \times 128$ | $128 \times 256$ | 190.50 | $u, v, p$ | $u, v$ |
| Controlled Cylinder | $96 \times 2$ | 3990 | $2.5 \times 10^{-3}$ | $64 \times 128$ | $128 \times 256$ | 187.08 | $u, v, p$ | $u, v$ |
| FSI | $51 \times 2$ | 2173 | $2.0 \times 10^{-3}$ | $128 \times 128$ | $128 \times 128$ | 94.73 | $u, v, p$ | $u, v$ |
| Foil | $99 \times 2$ | 3990 | $2.5 \times 10^{-3}$ | $128 \times 256$ | $128 \times 256$ | 335.64 | $u, v, p$ | $u, v$ |
| Combustion | $30 \times 2$ | 2001 | $2.5 \times 10^{-4}$ | $128 \times 128$ | $128 \times 128$ | 110.12 | see text | $I$ |

## B.2    DATA COLLECTION

When collecting real-world data through hardware experiments, we consider multiple different physical parameters, and each dataset is set with parameters based on hardware capabilities, adjustable parameters, and physically meaningful ranges. We correspond all the physical parameters of the simulated data with the real-world data one by one, and ensure the convergence and approximation of the numerical simulation to the real-world experiments by adjusting the spatial calculation domain and time step. The boundary conditions of fluid systems are the no-slip boundary condition on the solid/fluid interface, a uniform flow inlet condition, and a zero gradient exit condition. The configuration of combustion includes two separate inlets for air and the fuel mixture. Both inlets are defined with mass flow inlet boundary conditions, while the exit is a pressure outlet. The no-slip condition is applied to all wall surfaces. These conditions correspond to our experimental measurement environment. Please note that all data, including real-world and simulation data, are collected after stabilization to prevent inaccuracies caused by physical parameter conversion. The hardware settings and physical parameters for each of our datasets are in the following paragraphs.

As for the fluid systems, we calculate the simulated data using CFD (Lilypad [0] and Waterlily [1]) based on the parameters of real-world data. For the combustion system, a three-dimensional, implicit, unsteady Large Eddy Simulation (LES) is employed to simulate thermoacoustic instabilities in a swirl-stabilized combustor. The simulations are performed using the CFD software STAR-CCM+ 2022.1 (Siemens Digital Industries Software, Siemens 2022). The computational framework utilizes a pressure-based, segregated solver for the coupled solution of the governing equations. Near-wall flow physics are captured using an all y+ wall treatment. For modeling turbulent combustion, the Eddy Dissipation Concept (EDC) is applied to account for finite-rate chemistry effects within the turbulent flame structure (Lourier et al., 2017). The combustion chemistry is described by a reduced chemical mechanism for ammonia-methane co-firing, comprising 38 species and 184 elementary reactions (Sun et al., 2022). The stiff chemical kinetics are efficiently integrated using the CVODE solver.

The computational domain features two separate inlets, defined as mass flow inlets for the air and the fuel mixture, respectively, and an outlet specified as a pressure outlet. No-slip conditions are applied to all walls. A series of operating conditions is simulated to systematically examine the effects of fuel composition and equivalence ratio. The investigated matrix includes pure $CH_4$ and $NH_3/CH_4$ blends with ammonia fractions ranging from 20% to 80% by volume. For each fuel composition, the equivalence ratio is varied, typically from fuel-lean to fuel-rich conditions (e.g., $\phi$= 0.75–1.3),

---

[0]https://github.com/weymouth/lily-pad.
[1]https://github.com/WaterLily-jl/WaterLily.jl.

under a constant backpressure of 1 bar. These conditions are designed to correspond directly to the experimental measurement environment, facilitating a thorough validation of the numerical model.

**Cylinder.** The sampling frequency is 400Hz, which means it can take 400 photos per second. The duration is 20 seconds. In order to increase the gap between two neighboring time steps and demonstrate the predictive ability of models, we down-sample these data from 8000 time steps to 4000 time steps. In addition, to reduce noise in the data, we apply time-averaged filtering (Meinhart et al., 2000), with a sliding window size of 10 time steps, and take the average within this sliding window. Regarding physical parameters, the diameter of the cylinder is 30mm. We change different Reynolds numbers because the fixed cylinder only has this physical quantity that can be changed. Specifically, we change the Reynolds number by altering the incoming flow velocity, with a range of $1800 - 12000$. They are the minimum and maximum flow velocities that the equipment can achieve. Through PIV technique, we measure the velocity (in x- and y- directions) of the flow field (Abdulmouti, 2021). We conduct experimental data collection approximately every 100 intervals within this range. However, due to data loss and poor data quality during the experimental process, we selected 92 trajectories from them.

**Controlled Cylinder.** Both the sampling frequency and duration are the same as those of the above dataset. Since we introduce an external control variable, we set the Reynolds number to 1781, 2625, 3562, 4406, 5343, 6281, 7125, 8062, 9000, 9843. Please note, we change the Reynold number by changing the incoming flow velocity, and the Reynold number is calculated by $LU/\nu$, where $L$ is the characteristic Length, $U$ is the incoming flow velocity, and $\nu$ is the kinematic viscosity. For the control variable, we use the sinusoid function for forced vibration control, which uses different control frequencies, including 0.5Hz, 0.6Hz, 0.7Hz, 0.8Hz, 0.9Hz, 1.0Hz, 1.1Hz, 1.2Hz, 1.3Hz, and 1.4Hz. In the end, we removed low-quality data from 100 measurements and obtained 96 trajectories.

**FSI.** Due to FSI's need to capture higher frequency fluid structure coupling phenomena, we conduct the experiments with a sampling time of 500Hz and a duration of 8s. We connect two cylinders in series, with the front cylinder fixed to generate the Karman vortex street, and the rear cylinder installed on a smooth rail that can move in the y-direction randomly under the fluid force it receives. We change the mass ratio of the rear cylinder (18.2 and 20.8), which is achieved by 3D printing the new cylinder. In addition, we also change the Reynolds number within 3272, 3545, 3955, 4091, 4636, 5045, 5318, 6682, and 9068. For each experiment, we measure three times, starting from different times to change the initial conditions. One more thing, our damping ratio is fixed at 0.8. Based on the above parameters, we finally collect 51 trajectories.

**Foil.** In this experiment, our sampling frequency and duration are the same as those of the Cylinder. Differently, our foil is a 3D structure. Before, we approximate the 2D fluid field by making the cylinder have a universal diameter above and below, as this can be seen as an infinitely long cylinder. But for the foil, the cross-section we 3D-print is different from the top and bottom, and we use the NACA0025 foil (Bullivant, 1941; Xu et al., 2023), where the top chord length is 100mm and the minimum bottom chord length is 20mm. We choose to shoot the cross-section at 50mm. The cross-section captured in this way has a 3D effect. Then, we consider the angle of attack (aoa) of the foil, which directly affects the changes in the fluid field (He et al., 2021). The values are 0, 5, 10, 15, and 20. Similarly, the Reynolds numbers are 2968, 3750, 4531, 5312, 6093, 6875, 7656, 8437, 9218, 10000, 10781, 11562, 12343, 13125, 13906, 14687, 15468, 16250, and 17031.

**Combustion.** We use another set of equipment in the combustion experiment. We measure the luminescence intensity of combustion through flame CL imaging technique (Alviso et al., 2017). The sampling frequency is 4000Hz and the duration is 1s. The setting is the combustion of $NH_3$ and $CH_4$ mixed with air in different proportions. The ratios of $CH_4$ are 100%, 80%, 60%, 40%, and 20%, while the ratios of $NH_3$ are corresponding 0, 20%, 40%, 60%, and 80%. The equivalence ratios are 0.75, 0.85, 0.9, 1, 1.05, 1.1, 1.2, 1.25, and 1.3.

### B.3 DETAILS OF DATA MEASUREMENT

**Circulating Water Tunnel.** All fluid experiments were performed in a closed-loop Circulating Water Tunnel (CWT), as shown in Figure 9. The facility, with overall dimensions of 4.85 m (L) × 1.65 m (W) × 1.72 m (H), comprises the primary flow sections, a circulating pump coupled with a variable speed drive, and interconnecting piping. The main test section is constructed from high-

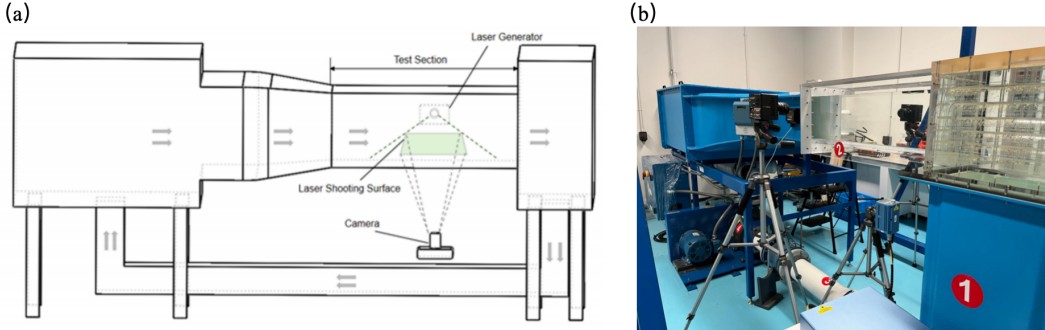

Figure 9: Schematic illustration of the Circulating Water Tunnel platform and the PIV measurement system installation. (a) A schematic of the water tunnel. (b) Experiment equipment.

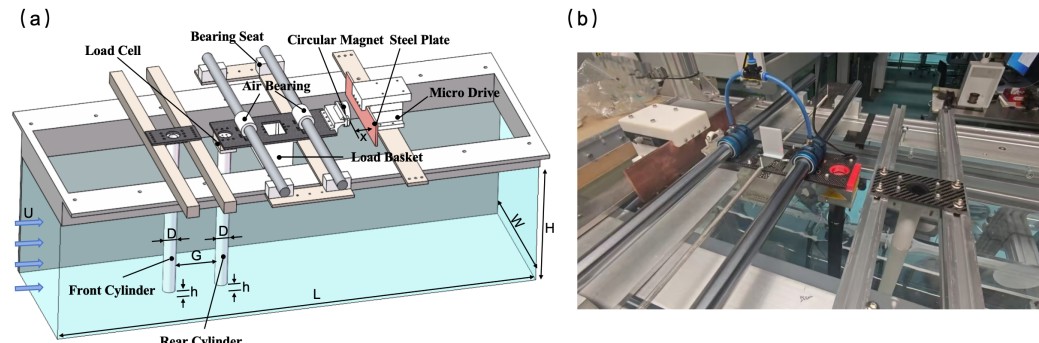

Figure 10: (a) A schematic of the FSI experimental set-up. (b) The experimental setup with free-vibration response of a cylinder in an oscillating cylinder wake.

transparency, highly abrasion-resistant acrylic, providing superior optical access for flow visualization and laser-based diagnostics. This section features internal working dimensions of 1.00 m (L) × 0.30 m (W) × 0.30 m (H). A key characteristic of the facility is its exceptional flow quality; within the operational velocity range of 0.01 m/s to 0.5 m/s, the free-stream turbulence intensity in the test section is maintained below 1%. This highly quiescent and stable flow environment is requisite for conducting high-precision measurements utilizing high-speed camera systems, laser diagnostics (*e.g.*, PIV), and sensitive force or pressure transducers.

**Cylinder.** For the static cylinder experiments, a cylinder with an outer diameter of $D = 30$ mm was rigidly fixed along the central axis of the CWT test section. The structure was fabricated from high-rigidity, lightweight photopolymer resin via 3D printing. To suppress three-dimensional end effects and approximate 2D flow conditions, the gap between the cylinder's lower end and the glass floor of the test section was precisely controlled at $h = 2$ mm, resulting in an immersion depth of $H = 298$ mm. The primary experimental parameter was the Reynolds number (Re). By continuously adjusting the inflow velocity, a Re range of 1800–12000 was investigated, corresponding to the minimum and maximum stable flow velocities of the facility. For PIV measurements, the horizontal laser sheet was positioned at a height of $z = 150$ mm from the test section floor. This location corresponds to the mid-span plane of both the test section and the high-aspect-ratio cylinder, capturing the most representative two-dimensional flow structures.

**Controlled Cylinder.** For the controlled cylinder experiments, the structure was integrated into a three-axis (XYZ) servo platform for active control(see Figure 2c. This platform was configured to execute single-degree-of-freedom (1-DOF) transverse (Y-axis) motion, while the X-axis (streamwise) and Z-axis (vertical) remained rigidly fixed. The cylinder ($D = 30$ mm, $H = 298$ mm, $h = 2$ mm) was precisely driven to perform forced harmonic oscillations following the trajectory $y(t) = A\sin(2\pi f t)$, with the amplitude $A$ fixed at 15 mm (A/D=0.5). This active control system, featuring a positioning accuracy of 0.1 mm and a control frequency of 100 Hz, ensured high-fidelity

reproduction of the motion profile. Under this configuration, the wake of the oscillating cylinder was systematically investigated at ten discrete Reynolds numbers (Re=1781, 2625, 3562, 4406, 5343, 6281, 7125, 8062, 9000, and 9843). These Re values were achieved by varying the inflow velocity while keeping the non-dimensional amplitude (A/D) constant. PIV measurements were likewise conducted at the mid-span plane ($z = 150$ mm).

**FSI.** The FSI experiments were conducted utilizing a tandem-cylinder arrangement. Both the upstream and downstream cylinders were fabricated from high-rigidity, lightweight photopolymer resin via 3D printing, each with an outer diameter of $D = 30$ mm. To suppress three-dimensional end effects, the gap between the lower end of the cylinders and the glass floor was held constant at $h = 2$ mm, ensuring an immersion depth of $H = 298$ mm. The central axes of both cylinders were aligned with the longitudinal centerline of the test section, and the streamwise (center-to-center) spacing was fixed at $x = 4D$. In this configuration, the upstream cylinder was rigidly fixed to serve as a static disturbance body. The downstream cylinder was designed as a high-precision single-degree-of-freedom (1-DOF) vibrating system. Its upper end was connected via a support plate to an air bearing, powered by a 0.5 MPa air supply. This structure strictly constrained the cylinder's motion to the transverse (cross-flow) direction while providing near-frictionless support. The dynamic characteristics (mass, stiffness, and damping) of the vibrating system were precisely controllable: (1) *Stiffness:* The restoring force was provided by two linear springs mounted symmetrically. The stiffness coefficients were pre-calibrated to ensure the system's natural frequency in air remained constant at 0.6 Hz, irrespective of the mass configuration. (2) *Mass:* The total oscillating mass ($m$) was controlled using a loading device (by adding or removing lead beads). The displaced fluid mass was $m_d = 210.6$ g, defining the non-dimensional mass ratio as $m_* = m/m_d$. (3) *Damping:* Structural damping was managed using a non-contact eddy current damping mechanism. This apparatus (see Figure 10b) utilizes a micro-actuator stage (resolution 0.01 mm) to adjust the gap between permanent magnets and a copper plate, thereby enabling precise tuning of the damping level. The actual damping ratio for each configuration was calibrated via free-decay tests conducted in air.

**Foil.** For the foil experiments, a three-dimensional tapered hydrofoil model based on the NACA0025 profile was utilized. The model was fabricated from high-rigidity, lightweight photopolymer resin via 3D printing. It featured a span of 298 mm, a root chord of 100 mm (at the top), and a tip chord of 20 mm (at the bottom). The model was rigidly fixed in the center of the CWT test section (see Figure 2a). The primary experimental parameters were the Angle of Attack ($\alpha$) and the Reynolds number (Re). The angle of attack was systematically varied across five conditions: $\alpha = 0°, 5°, 10°, 15°$, and $20°$. For each angle of attack, 19 discrete Reynolds numbers were investigated by adjusting the inflow velocity, covering a range of Re=2968–17031.PIV measurements were likewise conducted at the plane of $z = 50$ mm.

**Particle Image Velocimetry (PIV) System.** Flow visualization and quantitative measurements were performed using a Time-Resolved Particle Image Velocimetry (TR-PIV) system. The flow was seeded with hollow glass spheres (MV-H0520, China) with a nominal diameter of $10\,\mu$m and a density of $1.05$ g/cm$^3$, closely matching the density of water to ensure high flow-following fidelity. The illumination unit was a 10W continuous-wave (CW) laser (SM-SEMI-532nm-10W, China), which emitted a horizontal laser sheet (approx. 2 mm thick) into the measurement area from the side of the CWT after being shaped by an optical system. The image acquisition unit was a high-speed CMOS camera (Photron, FASTCAM Mini UX50, Japan) positioned beneath the tunnel floor, capturing images at a resolution of $1280 \times 1024$ pixels. The sampling frequency was set to 500 fps for the FSI experiment and 400 fps for the Cylinder, Controlled Cylinder, and Foil experiments. For the FSI experiment, acquired image sequences were processed using the commercial software MicroVec 3.6.5 [2]. Velocity fields were calculated using a multi-pass iterative cross-correlation algorithm, with an initial interrogation area (IA) of $32 \times 32$ pixels and a final IA of $16 \times 16$ pixels, both with a 50% overlap. Image sequences from the Cylinder, Controlled Cylinder, and Foil data were processed using the open-source software PIVlab 3.10 [3]. Velocity fields were computed using a Multipass FFT window deformation algorithm. This iterative procedure involved an initial pass (Pass 1) with a $32 \times 32$ pixel IA (50% overlap), followed by a final pass (Pass 2) with a $24 \times 24$ pixel IA (50% overlap). A $2 \times 3$-point Gaussian fitting algorithm was employed for sub-pixel displacement estimation. Following computation, raw vector fields underwent a rigorous post-processing

---

[2]https://piv.com.sg/.

[3]https://github.com/Shrediquette/PIVlab.

routine to reject spurious vectors. This process included a global rectangular velocity limit determined via histogram analysis, followed sequentially by a standard deviation filter (threshold=8) and a local median filter (threshold=3). Finally, any voids created by vector removal were filled using linear interpolation. Given the critical importance of velocity calibration in PIV post-processing, we performed a verification step beyond standard spatial calibration. Specifically, we compared the PIV-derived velocity in the free-stream region (areas undisturbed by the cylinder or foil) with the pre-calibrated bulk velocity of the circulating water tunnel to further confirm the accuracy of the flow measurements.

To facilitate validation and reproducibility, we have released a subset of the raw particle image sequences and calibration files in the database, along with the specific parameters for the Multipass FFT window deformation and multi-pass iterative cross-correlation algorithms. Furthermore, the raw velocity vector field data derived from the software processing has also been made available.

**Combustion.** The combustion dynamics are characterized primarily through high-speed OH* chemiluminescence imaging, which serves as the principal diagnostic technique for capturing flame structure and heat release distribution. A high-speed camera (Photron Mini AX 200) coupled with an image intensifier (EyeiTS-D-HQB-F) records the OH* chemiluminescence signals through a UV lens (Nikon PF10545MF-UV) and a $310 \pm 10$ nm bandpass filter, with the system capturing images at 2000 frames per second across a $70 \times 90$ mm$^2$ field of view. The instantaneous OH* intensity provides a direct indicator of local heat release rate, enabling detailed analysis of flame stabilization mechanisms and transient flame motions under varying operating conditions.

### B.4    COMPARISON BETWEEN SIMULATED AND REAL-WORLD DATA

In this section, we quantify the discrepancy between simulated and real-world data. For the fluid datasets, we use MVPE to measure the gap and visualize MVPs. Specifically, we evaluate MVPE over the full temporal window of each trajectory, with the results summarized in Table 8. The corresponding MVP visualizations are shown in Figure 11. The results reveal that the time-average velocity is close to the real-world data.

As for the Cylinder dataset, we also calculate the 200-step MVPE of simulated data and predictions from the baselines. Specifically, we compute MVPE over a 200-step temporal window. Model predictions are obtained by running ten rounds of autoregression to predict 200 steps. For the simulated and real data, we separately compute the norm of the velocity field, select for each the time point at which the norm attains its minimum as the starting point, and then extract the subsequent 200 time steps as the time window. The resulting MVPE for the simulated data is 0.05650, and Table B.4 reports the MVPE of each model under two training settings: Simulated training and Real-world finetuning. We observe that the MVPE of models pretrained on simulated data can be either larger or smaller than the MVPE of the simulated data itself. Smaller MVPE of models may be because the models input the same initial state as real-world data, while the alignment of simulated data and real-world data is not as strict as the models'. However, after finetuning on real-world data, all models achieve lower MVPE than the simulated data, highlighting both the importance of real-world measurements and the potential of deep learning models.

Table 8: **MVPE of simulated data (full trajectory).**

| Cylinder | Controlled Cylinder | FSI | Foil |
|:---:|:---:|:---:|:---:|
| 0.08718 | 0.06985 | 0.11440 | 0.08653 |

Table 9: **MVPE of baselines (200 steps).**

| | FNO | WDNO | UNet | DeepONet | MWT | CNO | GK-Transformer | Transolver | DPOT-SFT | DPOT-L-FT | DMD |
|---|---|---|---|---|---|---|---|---|---|---|---|
| Simulated training | 0.07418 | 0.05639 | 0.03137 | 0.05295 | 0.03547 | 0.06409 | 0.05431 | 0.08456 | 0.03378 | 0.03394 | - |
| Real-world finetuning | 0.01317 | 0.02181 | 0.01405 | 0.01787 | 0.01821 | 0.03488 | 0.01485 | 0.02374 | 0.01363 | 0.01250 | 0.10668 |

As for the combustion data, we implement frequency-domain analysis, focusing on OH radical mole fraction dynamics. The experimental instantaneous OH chemiluminescence signal is provided by

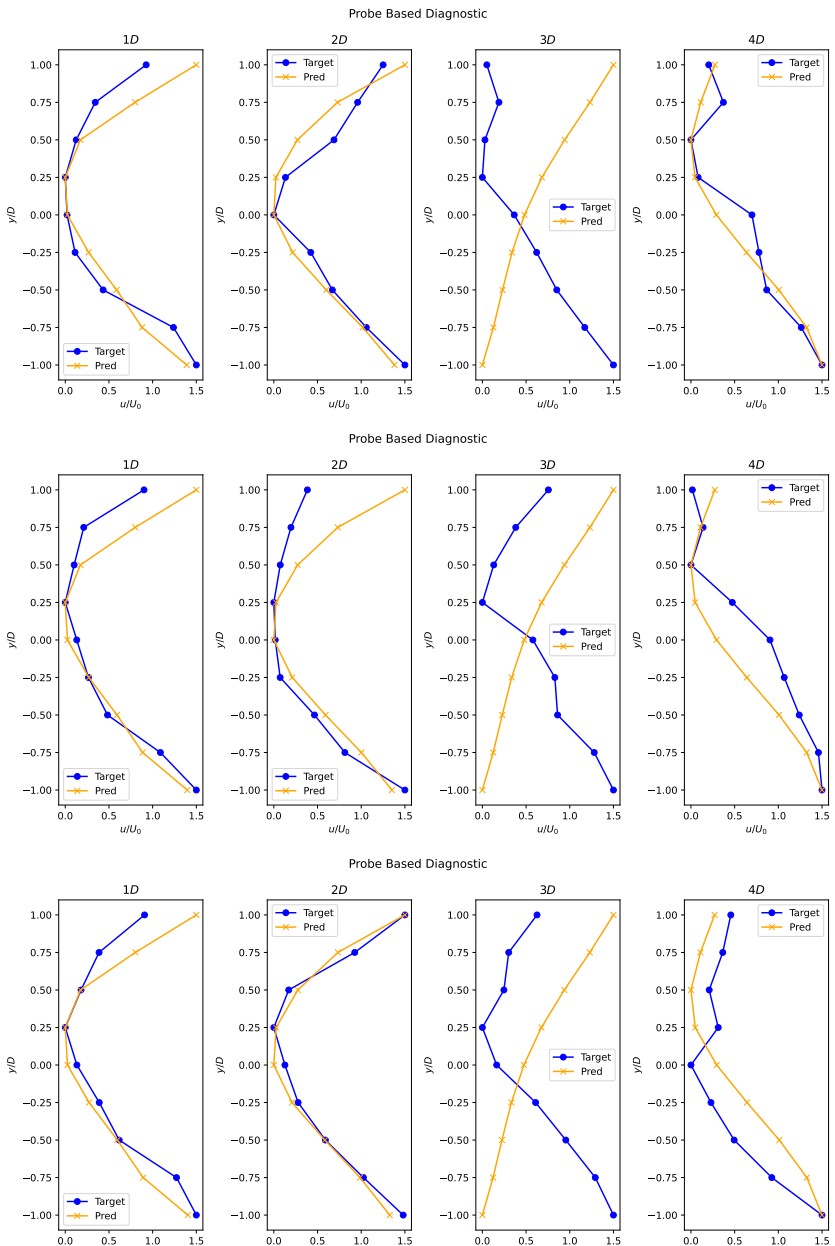

Figure 11: Mean velocity profile of simulated $u$ on Cylinder.

measurement, while the equivalent transient signal is extracted from the LES results by performing a spatial integration of the instantaneous OH concentration field. Both time series are detrended to remove the DC component, and transformed into the frequency domain using the Fast Fourier Transform (FFT), showing the dominant frequencies of the combustion instability. The visualization result is provided in Figure 12.

## C   CODE FRAMEWORK

### C.1   SCRIPTS

The training script's form is as follows:

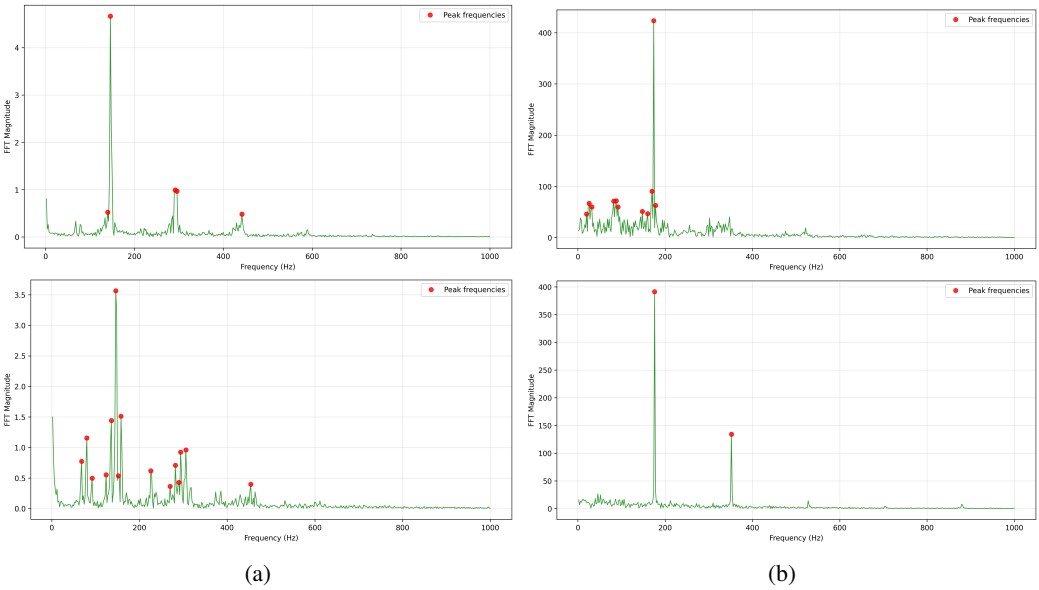

Figure 12: Frequency analysis of the Combustion dataset. (a) Real-world data. (b) Simulated data.

```
python train.py \
    --config CONFIG_PATH \
    --train_data_type TRAIN_DATA_TYPE \
    --is_finetune IS_FINETUNE
```

where CONFIG_PATH is the path of the baseline's config file, TRAIN_DATA_TYPE is real
or numerical, and IS_FINETUNE is True or False. TRAIN_DATA_TYPE=numerical
and IS_FINETUNE=False corresponds to simulated training, TRAIN_DATA_TYPE=real and
IS_FINETUNE=False corresponds to real-world training, and TRAIN_DATA_TYPE=real and
IS_FINETUNE=True corresponds to real-world finetuning.

The evaluation script is

```
python eval.py \
    --config CONFIG_PATH \
    --checkpoint_path CHECKPOINT_PATH
```

where CONFIG_PATH is the path of the baseline's config file, and CHECKPOINT_PATH is the path
of the checkpoint.

## C.2  DATA PRE-, POST-PROCESSING, OPTIMIZER AND SCHEDULER

Our code framework supports three types of data pre- and post-processing: none (no normaliza-
tion), Gaussian (standardization to zero mean and unit variance), and range (normalization by the
maximum absolute value).

Besides, our code framework adopts the Adam optimizer (Kingma & Ba, 2014), and the learning
rate scheduler supports two strategies: the step decay scheduler and cosine annealing scheduler
(Loshchilov & Hutter, 2016).

## C.3  DATASET MODULE

All datasets are loaded using a unified module RealDataset, whose parameters are detailed be-
low. To support evaluations under both in-domain and out-of-domain (OOD) settings, we provide
a flexible 'test_mode' option in the 'RealDataset' module. Users can specify 'test_mode' as 'all',
'in_dist', 'out_dist', 'seen', or 'unseen' to control which parameter regimes are included in the test
set. Here, 'in_dist' refers to parameters in the distribution of the training set, while 'out_dist' refers

to the out-of-distribution parameters. The 'seen' mode evaluates model performance on parameter regimes observed during training, while the 'unseen' mode enables OOD testing by selecting data from parameter regimes that do not appear in the training set. This design allows users to conveniently assess the model's generalization capability.

```python
class RealDataset(Dataset):
    def __init__(
        self,
        dataset_name,
        dataset_root,
        dataset_type,
        mode,
        test_mode,
        mask_prob,
        in_step,
        out_step,
        N_autoregressive,
        interval,
        train_ratio,
        split_numerical,
        trunk_length,
        noise_scale,
        n_sim_in_distribution,
        n_sim_out_distribution,
        n_sim_frame,
        sub_s_real=1,
        sub_s_numerical=1,
        noise_type='gaussian',
        optical_kernel_size=4,
        optical_sigma=1.0
    ):
        super().__init__()
        '''
        dataset_name: name of the dataset
        dataset_root: root path of the dataset
        dataset_type: real | numerical
        mode: train | val | test
        test_mode: all | in_dist | out_dist | seen | unseen
        mask_prob: probability of masking the unmeasured modalities, only
            for numerical
                datasets
        in_step: number of steps for input
        out_step: number of steps for output
        N_autoregressive: number of autoregressive times
        interval: interval of the sliding window
        train_ratio: ratio of training data
        split_numerical: split numerical data into training, validation
            and test data or not
        trunk_length: length of the trunk for splitting on simulation
            into training and
                test data
        noise_scale: scale of the noise added to numerical data
        n_sim_in_distribution: number of simulations for in-distribution
            test
        n_sim_out_distribution: number of simulations for out-
            distribution test
        n_sim_frame: number of frames in each simulation
        sub_s_real: spatial sub-sampling factor for real data
        sub_s_numerical: spatial sub-sampling factor for numerical data
        noise_type: type of noise, gaussian | poisson | optical
        optical_kernel_size: size of the kernel for optical noise
        optical_sigma: standard deviation of the kernel for optical noise
        '''
```

## C.4 Model Module

We define a unified `Model` class, from which all baselines are derived. This class provides the following functions.

```
class Model(nn.Module):
    def __init__(self, ...):
        # Initialize the model with given parameters

    def forward(self, x):
        # Forward pass of the model

    def train_loss(self, input, target):
        # Compute the training loss for a batch

    def load_checkpoint(self, checkpoint_path, device):
        # Load checkpoints from the save path
```

# D    Experiment Details

## D.1    Mask-training Strategy

Because the simulated data contain more modalities than the real-world data, we design specific strategies to utilize the additional modalities. Specifically, we employ the mask-training strategy. During simulated training, we randomly mask the additional modalities with a fixed probability. Consequently, the trained model can be directly applied to real-world test data by simply concatenating zero vectors in place of the unobserved modalities. With training on the additional modalities, the model can learn more information about the dynamics.

## D.2    Combustion Surrogate Model Training

For the Combustion dataset, a particular challenge is that the observed modalities in real-world data are not directly available in the simulated data. To address this, we train an additional surrogate model whose input is the simulated modalities and output is the corresponding real-world modalities at each time step. The surrogate model is trained on paired simulated and real-world data from the same periods. We choose the U-Net as the surrogate model due to its outstanding performance. During simulated training on this dataset, the simulated data are concatenated with the output of the surrogate model.

# E    Baseline Models

## E.1    DMD

The Dynamic Mode Decomposition (DMD) is a traditional method that extracts spatiotemporal coherent structures from time-series flow field data through a data-driven matrix decomposition approach. DMD processes volumetric flow data by reshaping the spatiotemporal field into snapshot matrices $\mathbf{X}_1$ (first $n-1$ snapshots) and $\mathbf{X}_2$ (last $n-1$ snapshots), where each column represents a spatial state at a given time step. It performs singular value decomposition (SVD) on $\mathbf{X}_1$ to obtain $\mathbf{X}_1 = \mathbf{U}\mathbf{\Sigma}\mathbf{V}^T$, with optional rank truncation $r$ for dimensionality reduction. The low-dimensional evolution operator is computed as $\tilde{\mathbf{A}} = \mathbf{U}^T\mathbf{X}_2\mathbf{V}\mathbf{\Sigma}^{-1}$, followed by eigenvalue decomposition to extract eigenvalues $\lambda_i$ and eigenvectors $\mathbf{W}$. DMD modes are constructed via $\mathbf{\Psi} = \mathbf{X}_2\mathbf{V}\mathbf{\Sigma}^{-1}\mathbf{W}$, where each mode $\psi_i$ represents a spatial coherent structure. Modal amplitudes $\mathbf{b}$ are determined by least-squares fitting of the initial condition $\mathbf{x}_0$ onto the DMD modes. Modes are sorted in descending order by amplitude magnitude $|\mathbf{b}|$, and the top $n_{\text{modes}} = 10$ modes with the largest amplitudes are selected and retained for prediction. Future states are predicted using the linear superposition formula $\mathbf{x}(t) = \sum_i b_i \psi_i \exp(\lambda_i \cdot t)$.

### E.2    U-NET

The 3D U-Net processes volumetric data through an encoder-decoder structure with skip connections. Its encoder module uses successive blocks of two 3D convolutional layers (kernel $K$, padding $P$, ReLU activation) followed by max pooling (stride/kernel $\mathbf{S}$), progressively halving spatiotemporal resolution while doubling channel counts. A bottleneck of two convolutional layers bridges to the decoder, which employs transposed convolutions (stride/kernel $\mathbf{S}$) for upsampling, feature concatenation via skip connections, and two convolutional layers per block. The output layer uses $1 \times 1 \times 1$ convolution to map features to target fluid state variables, enabling effective spatiotemporal feature extraction for medical imaging and fluid dynamics applications. In our configuration, the tunable parameters include the number of encoder–decoder layers, the batch size, the learning rate, and the total number of update iterations.

### E.3    CNO

The Convolutional Neural Operator (CNO) is a deep learning framework designed to learn mappings between function spaces through hierarchical convolutional architectures. It extends operator learning by incorporating spectrally motivated filters, which ensure stable frequency representations across scales. The network follows an encoder–decoder structure with skip connections and a bottleneck composed of residual blocks. The lifting and projection layers map input fields into a higher-dimensional latent space and back to the output space. Each CNO block consists of a 3D convolution, optional batch normalization, and a filtered activation, where upsampling and downsampling are handled inside the activation functions. In our configuration, the tunable parameters of the CNO model include the number of encoder–decoder layers, the number of residual blocks per level and in the bottleneck, the channel multiplier that controls feature width, as well as filter-related hyperparameters.

### E.4    DEEPONET

Deep Operator Network (DeepONet) is a deep learning network designed to approximate a nonlinear continuous operator, which models complex relationships between functions via two sub-networks. The branch net processes the input function to extract the input features, and the trunk net encodes the location coordinates for the output space. The outputs of the two sub-networks are combined by dot product to generate the final results. In our baseline, the code framework of DeepONet uses a 3D Convolutional Neural Network (CNN) in the branch net and Fully Connected Layers in the trunk net. In our configuration, the tunable parameters of the DeepONet model include the dropout rate and the feature dimension (p) of the outputs from both the trunk net and branch net.

### E.5    FNO

The Fourier Neural Operator (FNO) is a deep learning framework designed to learn mappings between infinite-dimensional function spaces. It parameterizes the integral kernel in the Fourier domain, where inputs are processed by successive Fourier layers that apply linear operations to enable efficient convolution. This design naturally supports zero-shot super-resolution. After several Fourier layers, there are two linear layers. In our configuration, the tunable parameters of the FNO model include the truncation level of frequencies (modes1, modes2, modes3), the number of Fourier layers, and the width of the linear layer.

### E.6    WDNO

Wavelet Diffusion Neural Operators (WDNOs) aim to model physical systems with generative models. The diffusion model is a generative framework that learns to transform random noise into data samples by reversing a gradual noising process (Ho et al., 2020). WDNOs combine the wavelet transform with diffusion models to capture the abrupt changes in complex physical systems, while leveraging the diffusion models' strong ability to model high-dimensional distributions. In our code framework, we take the U-Net as the denoising network. In the code framework of WDNO, the tunable parameters include the U-Net's base feature dimension, the U-Net's channel multipliers for each stage, the schedule of diffusion models, the type of wavelet basis, and the padding mode of wavelet transform.

### E.7 MWT

The Multiresolution Wavelet Transform Neural Operator (Gupta et al., 2021) learns mappings between function spaces by combining wavelet-based multiscale decomposition with sparse Fourier transforms. At each resolution level, MWT applies wavelet filters to extract coarse and fine-scale features, processes them via frequency- and space-domain kernels, and reconstructs the output using inverse transforms. This design enables efficient modeling of both global structures and localized variations. In our setup, we use a wavelet filter size of $k = 3$ and a spectral width $\alpha = 5$ to balance resolution and frequency modeling. The channel multiplier is set to $c = 4$ to increase latent capacity, and we stack $n = 4$ multiwavelet operator blocks. The coarsest level is $L = 0$, allowing full-resolution output. We use Legendre polynomials to construct the wavelet basis due to their favorable numerical properties.

### E.8 GK-TRANSFORMER

The Galerkin Transformer is an attention-based operator learning framework that removes the softmax in self-attention and admits a projection-based interpretation. In our baseline, we follow the original Galerkin Transformer architecture, while modifying its output regressor: instead of a simple Fourier transform, we employ a Fourier Neural Operator (FNO) layer. This replacement enhances the model's ability to capture volumetric features and recover fine-scale structures, while the rest of the GT design remains consistent with the original implementation. The tunable parameters of this model include the number of layers, the number of attention heads, the hidden size, and the parameters of the FNO regressor.

### E.9 TRANSOLVER

The Transformer-based Solver (Transolver) is a neural operator framework designed to solve partial differential equations (PDEs) by leveraging the global receptive field of transformer architectures. Instead of relying on localized convolutional kernels or spectral truncations, Transolver employs multi-head self-attention to capture long-range dependencies in the solution space, making it particularly effective for modeling nonlocal interactions. The architecture typically consists of stacked transformer encoder blocks, each containing self-attention, feed-forward layers, and residual connections, combined with positional encodings to handle spatial coordinates. To enhance numerical stability, normalization layers and gating mechanisms are integrated across layers. In our configuration, the tunable parameters of the Transolver model include the number of transformer encoder layers, the number of attention heads, the hidden dimension of each layer, the choice of positional encoding scheme, and the dropout rate used for regularization.

### E.10 DPOT

The Denoising Pre-training Operator Transformer (DPOT) is a scalable neural operator framework for learning a foundation model for spatiotemporal PDE mappings. It combines an auto-regressive denoising objective with a Fourier-attention Transformer backbone for robust, transferable representations across diverse PDE systems. During pre-training, DPOT predicts the next timestep from the previous $T$ frames with injected Gaussian noise to mitigate train-test mismatch and enhance long-horizon stability. The architecture includes a patch embedding layer with positional encoding, a temporal aggregation layer using Fourier features to compress $T$ input frames, and stacked multi-head Fourier attention blocks that apply learnable nonlinear transformations in the frequency domain for efficient global mixing. In our benchmark, we use the official small (30M) and large (509M) pretrained DPOT models. To adapt to varying dataset channels while reusing pretrained input/output projections (designed for 4 channels), we pad inputs/outputs with all-ones for datasets with fewer than 4 channels, following the paper; for more than 4 channels, we reinitialize the respective projection layer. Original training is at $128 \times 128$ resolution, so we preprocess all data to $128 \times 128$ via Fourier-domain zero-padding/truncation resize (as in the paper) before input, and postprocess predictions back to original resolution identically during training and inference. In our configuration, we load the pretrained DPOT models and leave model structural hyperparameters such as the number of Fourier attention layers, attention dimension, MLP dimension, or number of heads as they are and only tune the optimization hyperparameters for finetuning, such as learning rate and total learning steps, to preserve the integrity of the pretrained model.

## F    LIMITATIONS AND FUTURE WORKS

While RealPDEBench provides the benchmark with paired real-world and simulated datasets, its scope is currently limited. Extending the benchmark to other domains such as electromagnetics, structural mechanics, or aerodynamics would broaden its applicability. In addition, although we provide multiple physics-oriented metrics, no dedicated metrics have been introduced specifically for the Combustion system, which could further strengthen physics fidelity assessments. Another limitation is that the current benchmark does not systematically explore strong out-of-distribution regimes. Future work will expand RealPDEBench to cover additional physical scenarios, introduce other metrics, and design out-of-distribution tasks. We also maintain a long-term development view towards more data and models in the future, thereby enabling a more comprehensive evaluation of scientific ML in real-world contexts.

## G    VISUALIZATION

In this section, we visualize one sample's results of all baselines on all datasets (real-world data finetuning setting), including ground truth and predictions.

## H    THE USE OF LARGE LANGUAGE MODELS (LLMS)

In this paper, we employ LLMs as general-purpose assist tools for text refinement and language polishing. All core research ideas, datasets, and scientific conclusions presented in this paper are our own original contributions.

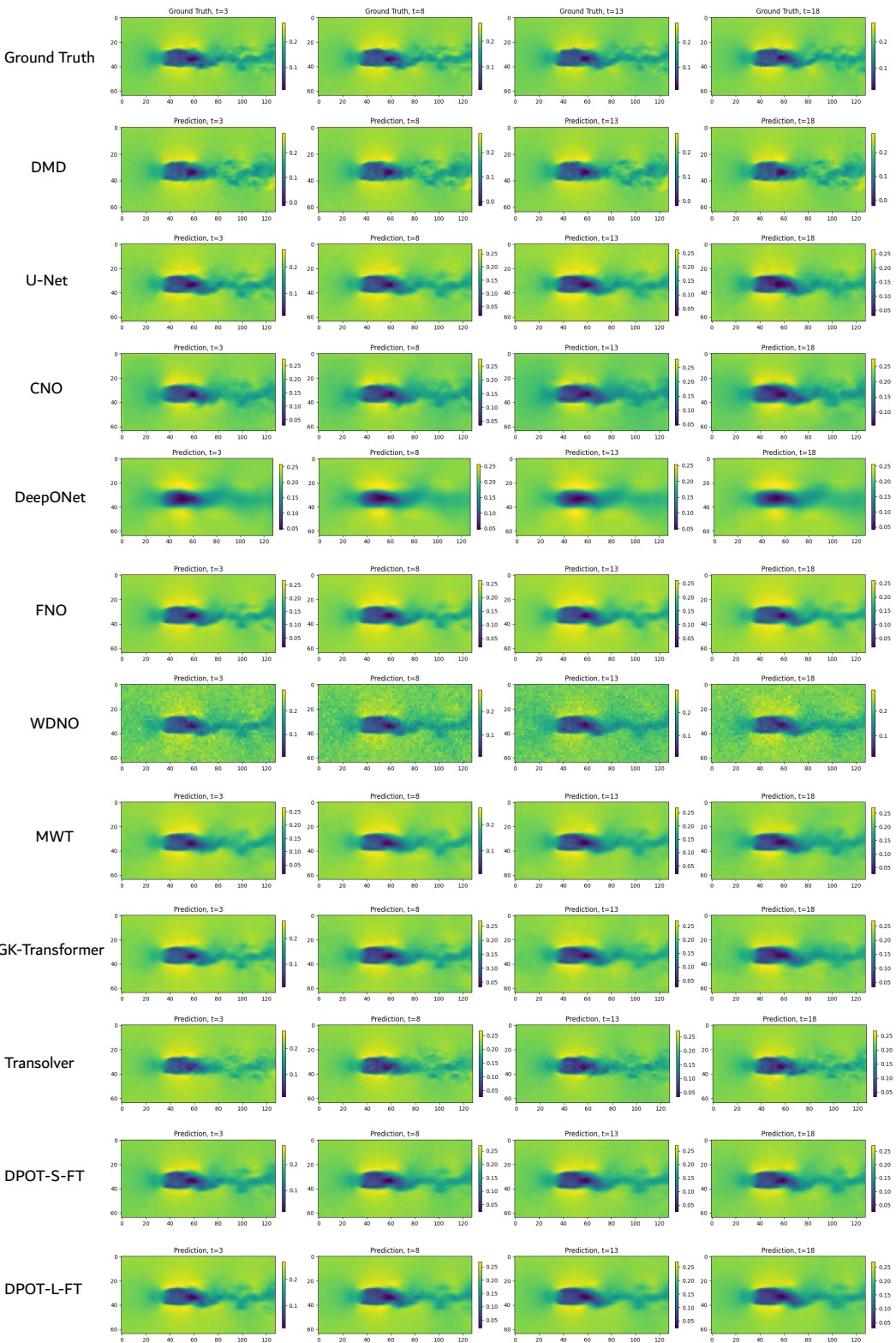

Figure 13: Visualization of results ($u$) on Cylinder dataset.

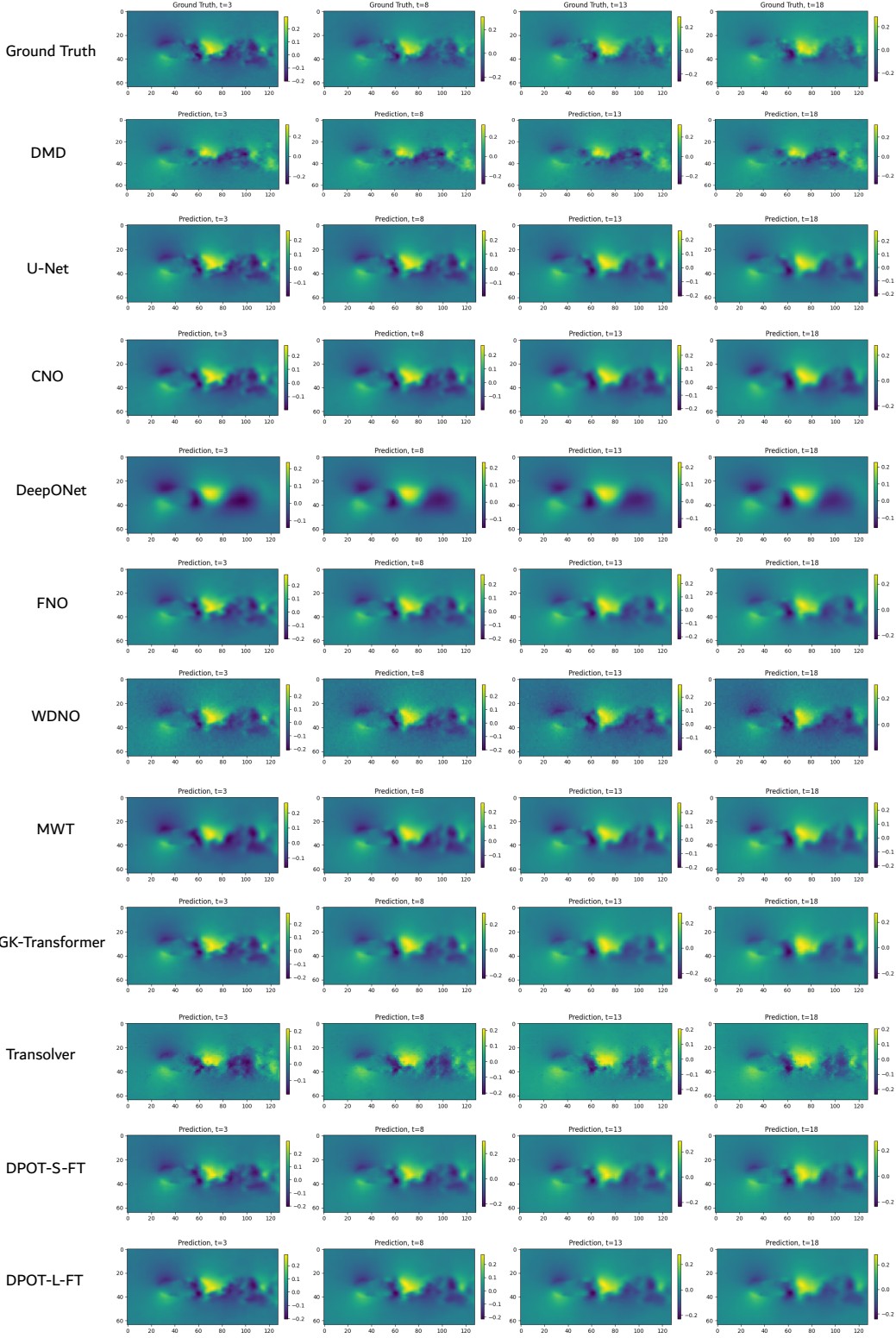

Figure 14: Visualization of results ($v$) on Cylinder dataset.

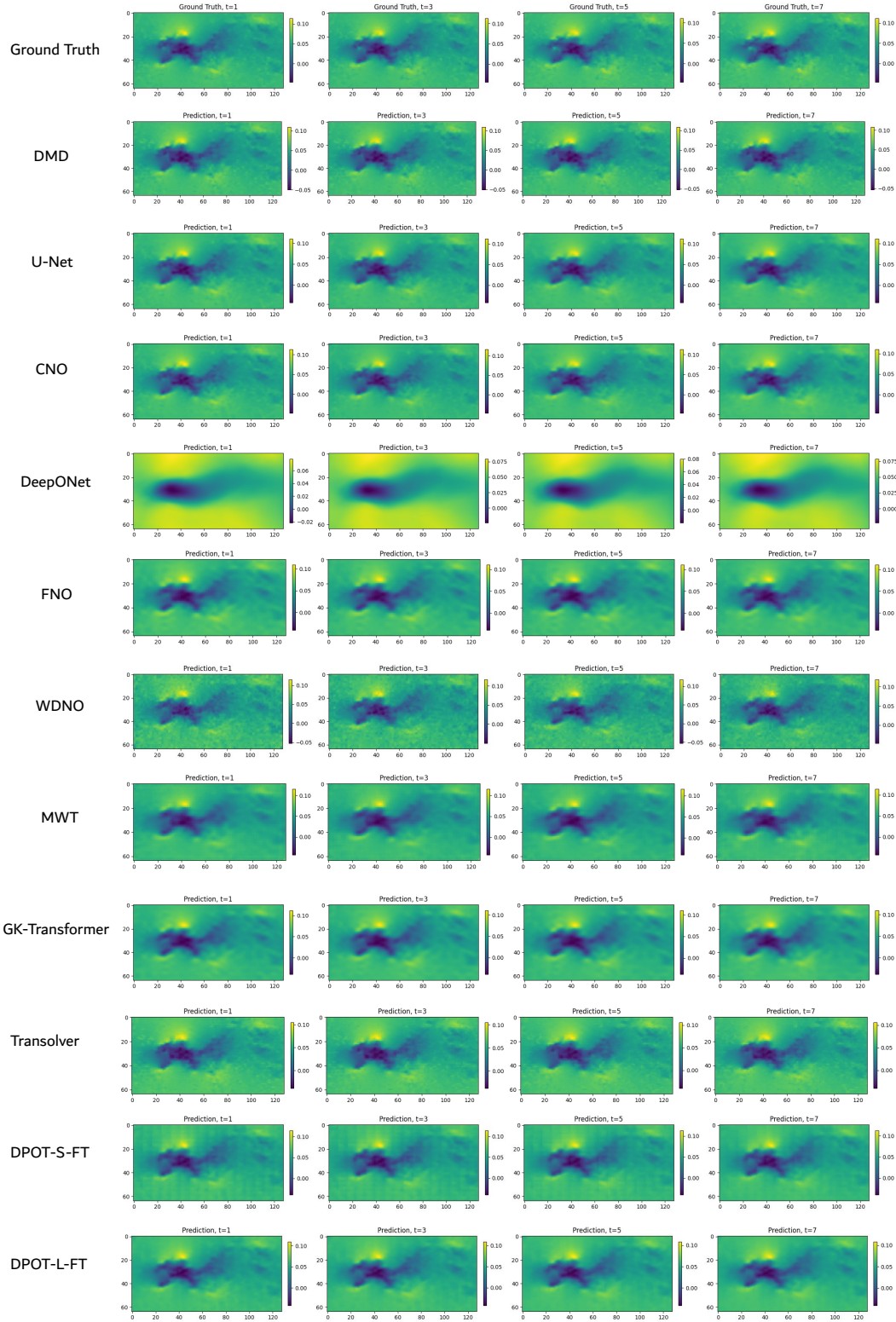

Figure 15: Visualization of results ($u$) on Controlled Cylinder dataset.

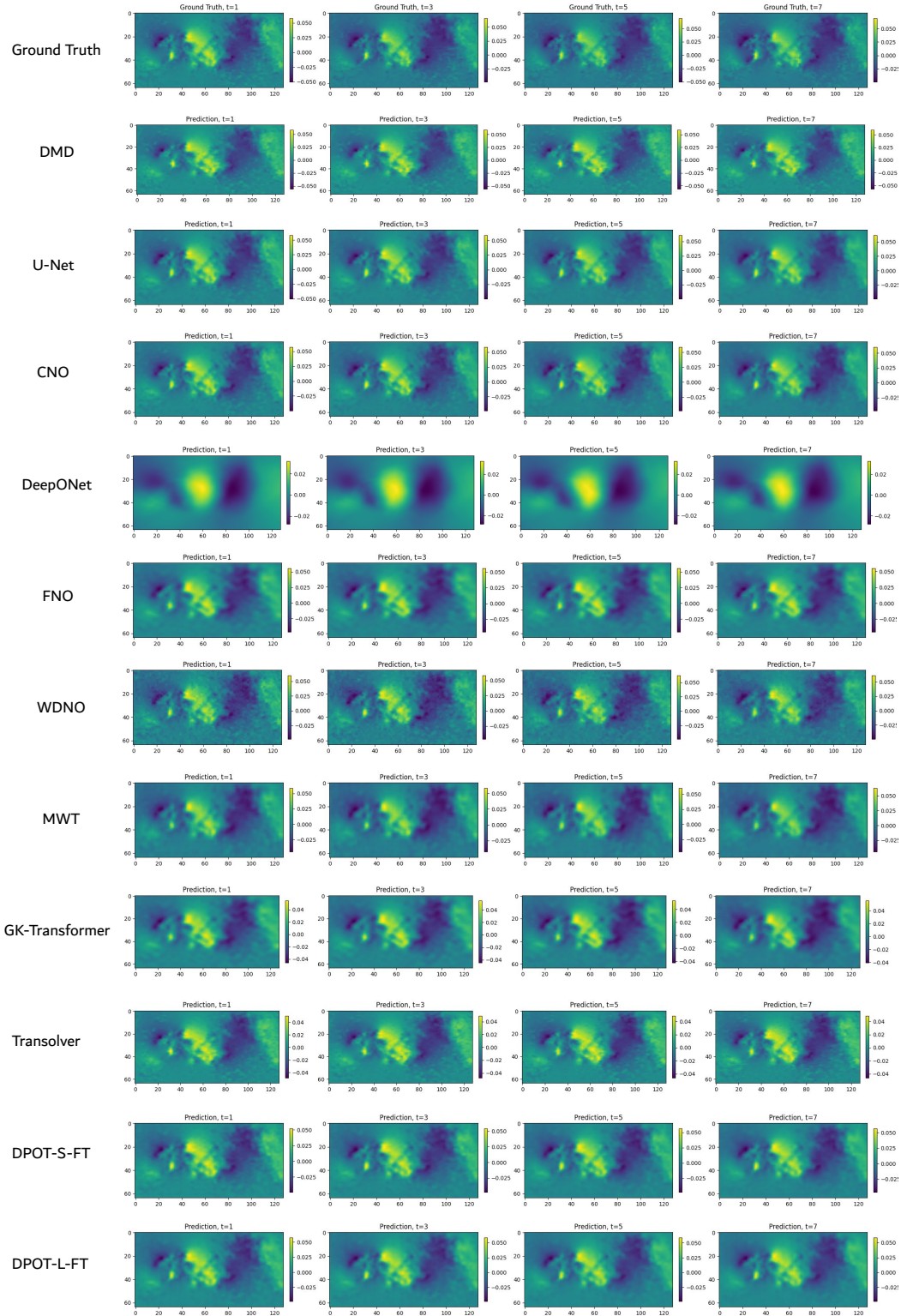

Figure 16: Visualization of results ($v$) on Controlled Cylinder dataset.

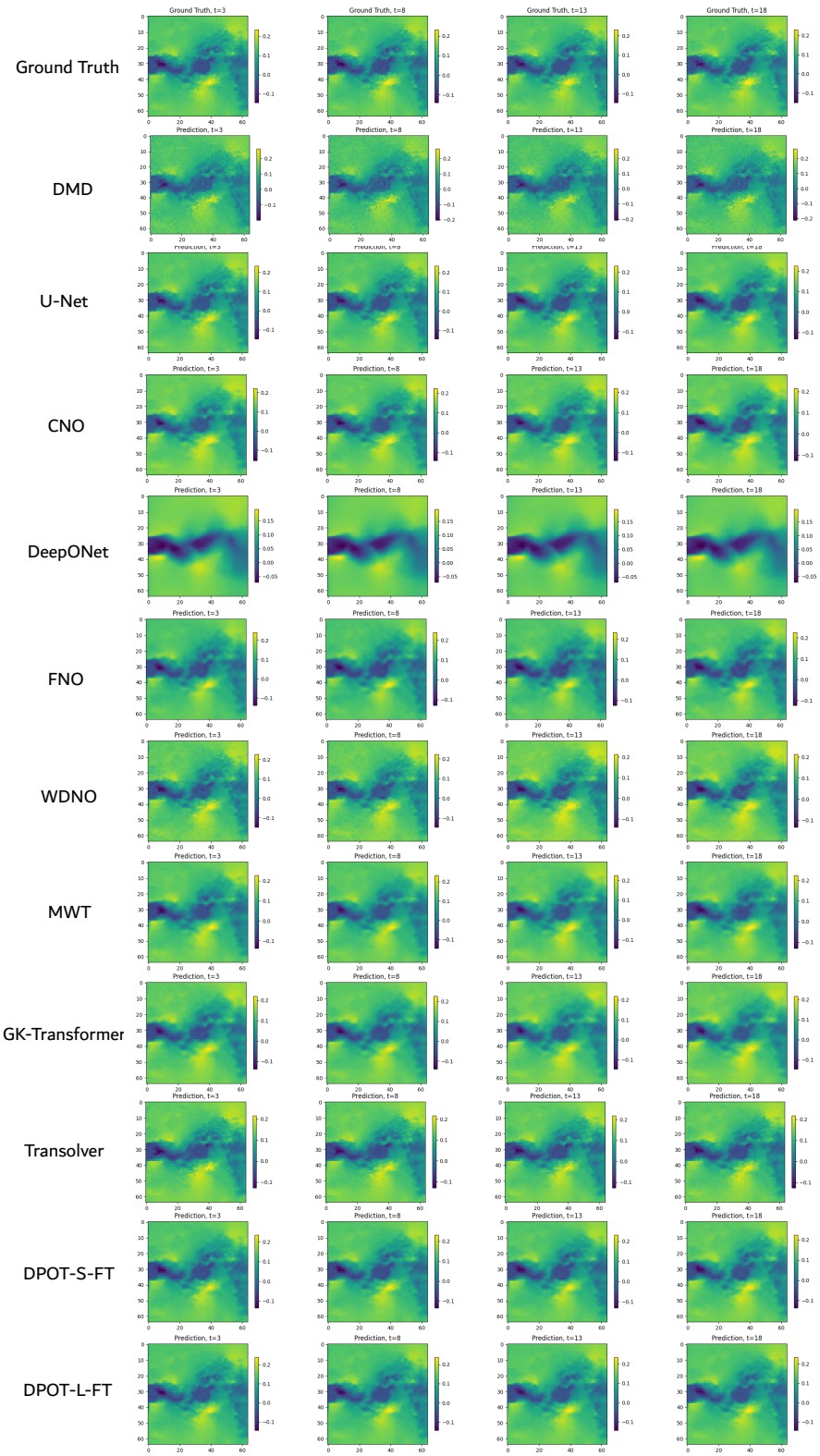

Figure 17: Visualization of results ($u$) on FSI dataset.

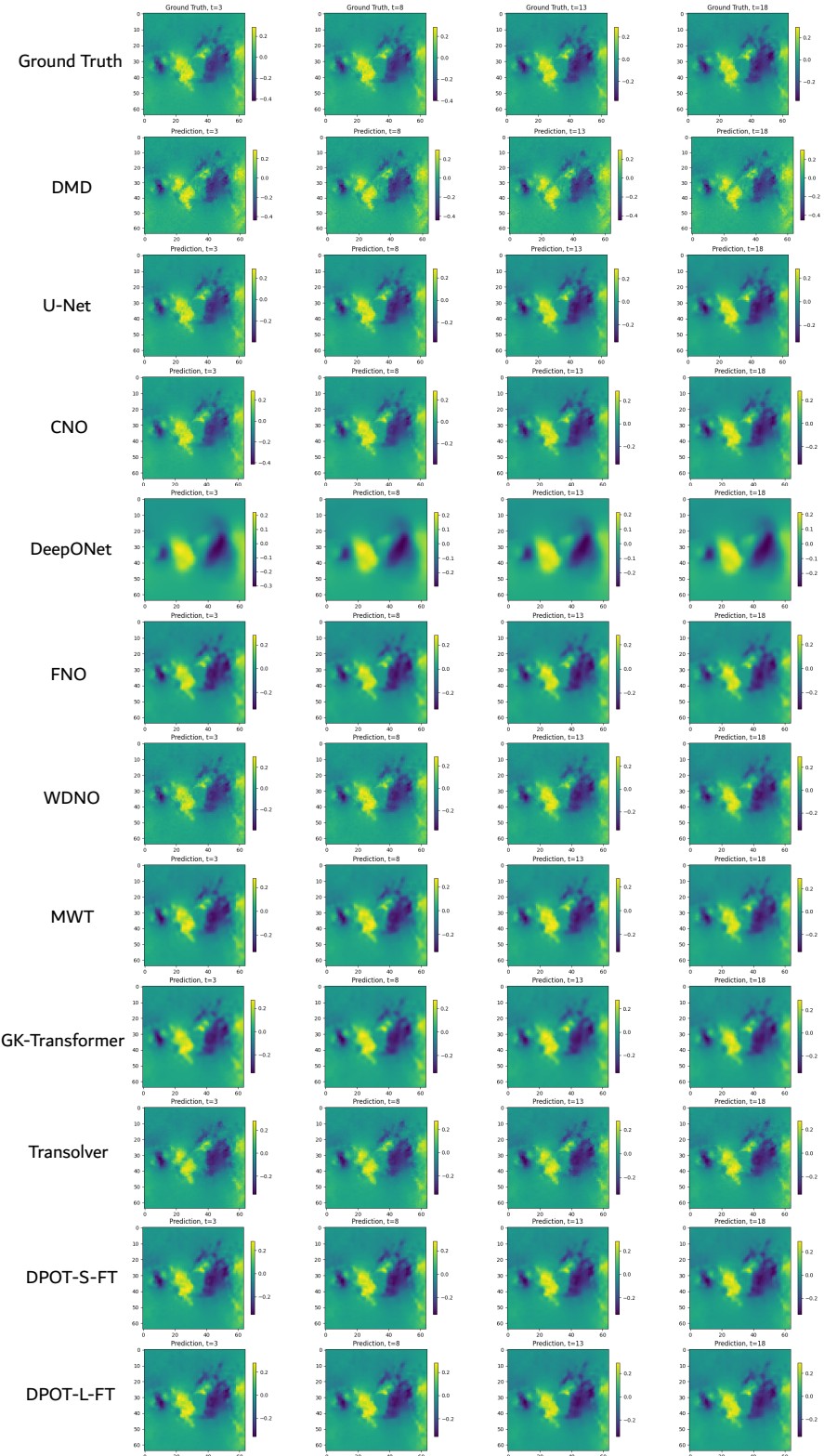

Figure 18: Visualization of results ($v$) on FSI dataset.

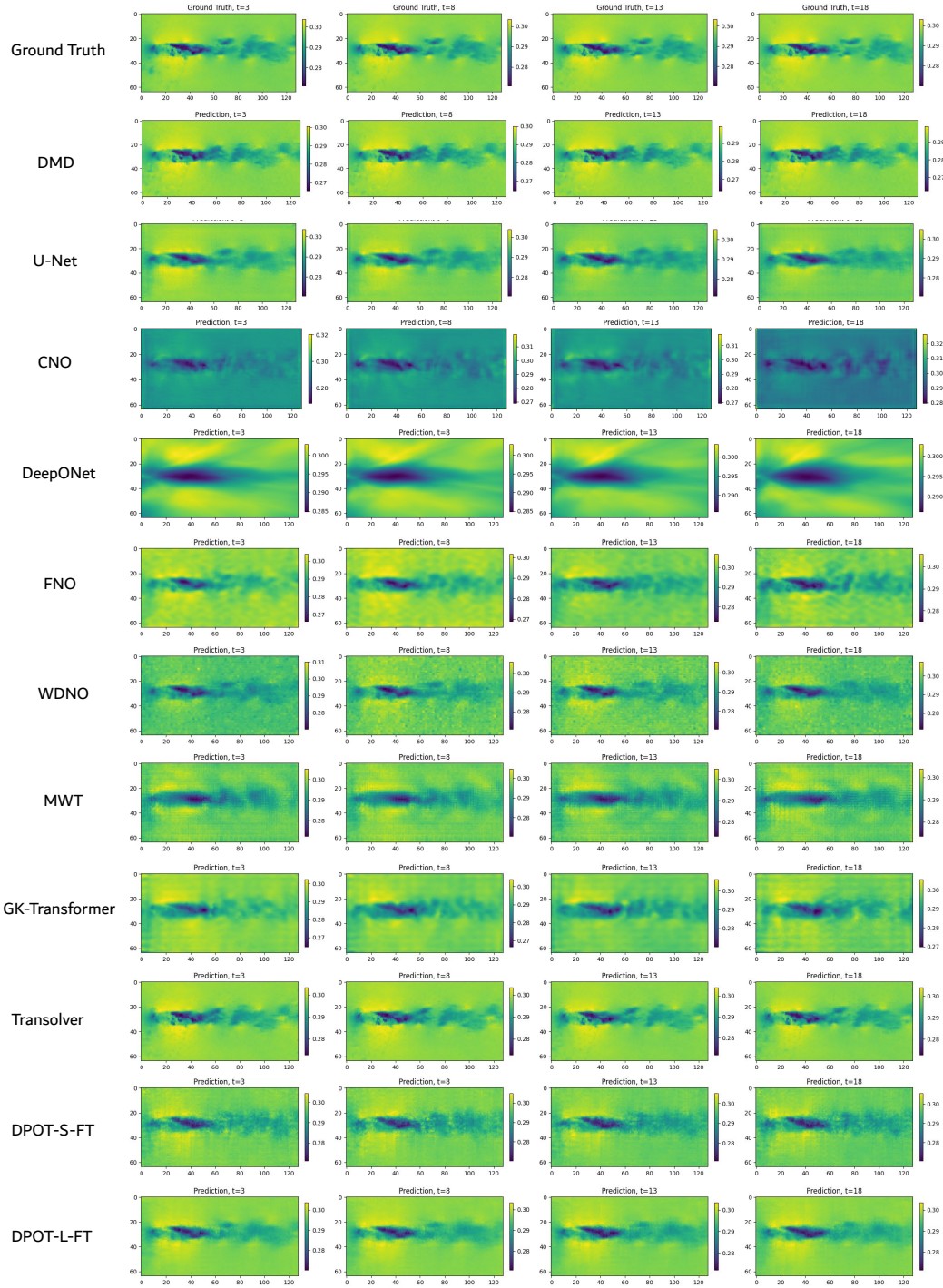

Figure 19: Visualization of results ($u$) on Foil dataset.

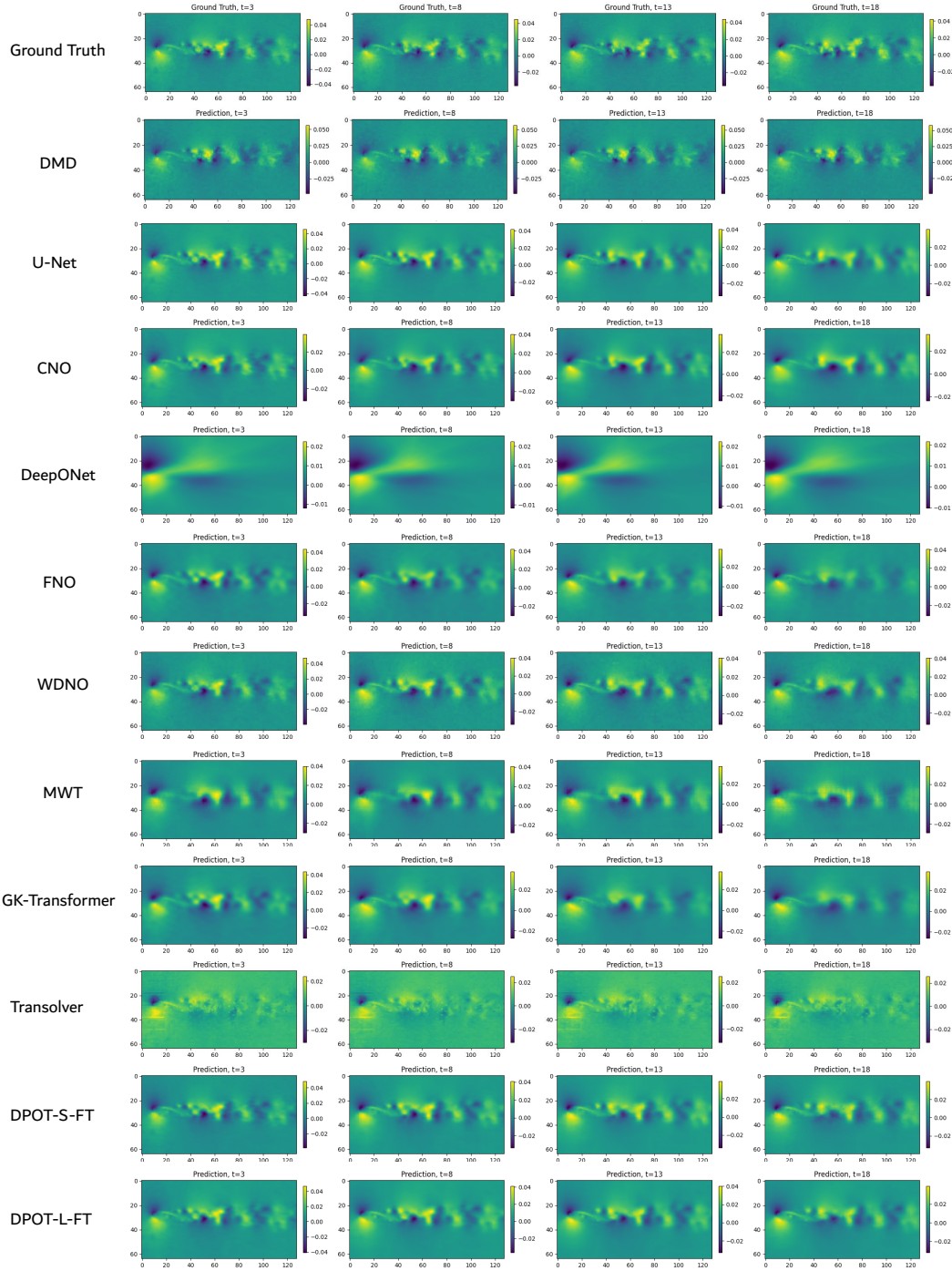

Figure 20: Visualization of results ($v$) on Foil dataset.

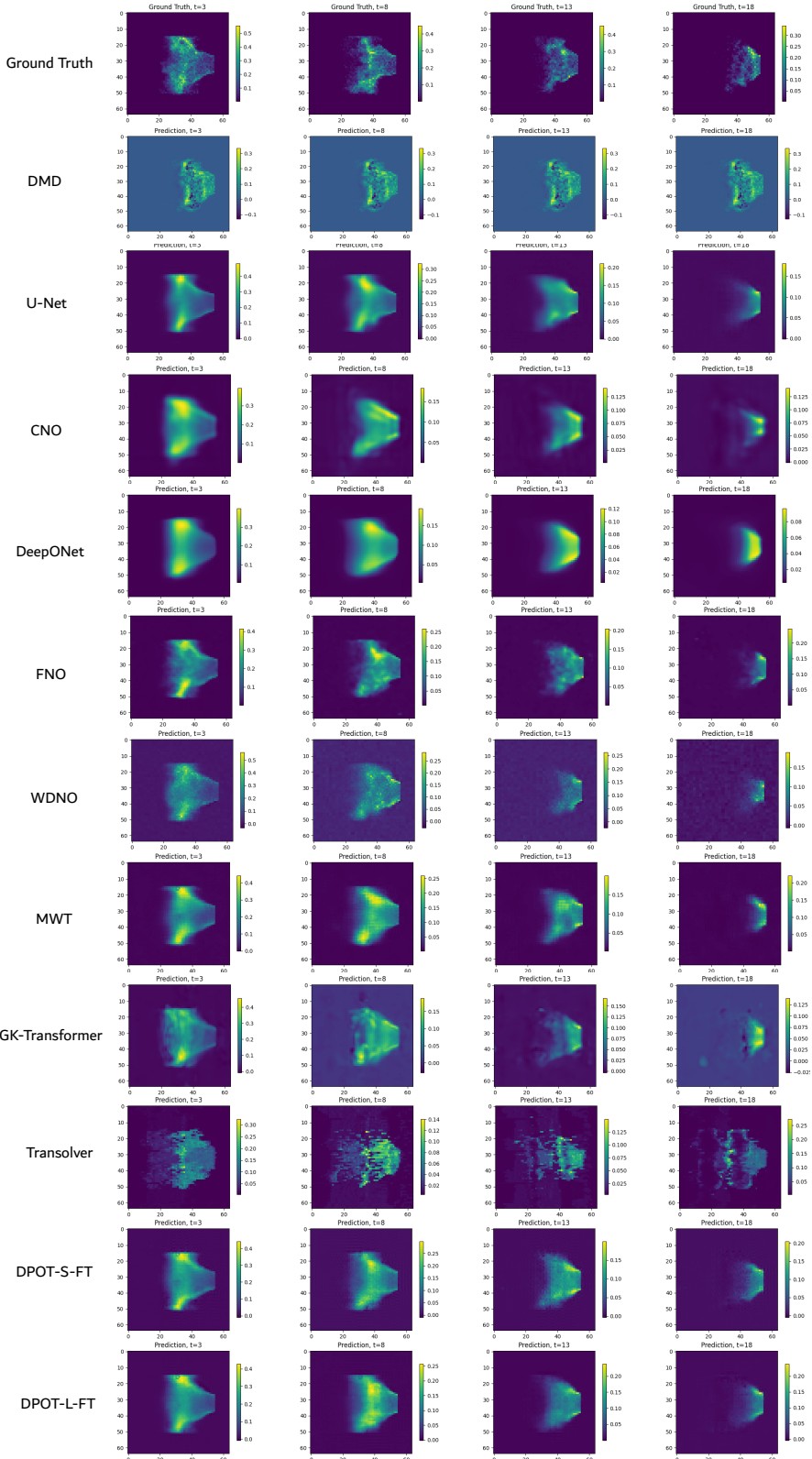

Figure 21: Visualization of results ($I$) on Combustion dataset.

