# OpenReview forum: "RealPDEBench: A Benchmark for Complex Physical Systems with Real-World Data"
_ICLR.cc/2026/Conference — ICLR 2026 Oral_

### Official Review · Reviewer_YXXo · 2025-10-29

**Soundness:** 3
**Presentation:** 4
**Contribution:** 4
**Rating:** 10
**Confidence:** 4

**Summary:**

This paper introduces a new dataset featuring paired simulation and experimental data for a variety of systems. The authors then benchmark a variety of state-of-the-art models on this data exploring the impact of simulated data on learning the real-world dynamics.

**Strengths:**

This type of dataset is so valuable to the community that I would strongly recommend acceptance even if the paper itself had major issues. Almost all work in this space today uses simulation data exclusively which is a space where learned models can at best offer speed improvements. This work and future datasets like it can offer a chance to explore in which regimes we may be able to achieve improved accuracy over numerical simulation and could great inform research directions over time.

That said, the paper itself is quite strong. Some particular reasons:
1. The presentation is excellent. It explains the problem very cleanly and describes prior work and the intended goal of this dataset.
2. The appendix is quite extensive providing data on the models and data generation process. In general, the division between main text and appendix content feels very well planned.
3. Experimental fluids data is extremely rare and valuable to the community for developing models that can operate independently of the standard numerical solver framework.

**Weaknesses:**

The benchmarking has a few issues, but I'm largely reading this as a dataset paper, so I'm not marking down for them. However, in the interest of improving the paper, I will still list them out here.

Major:
1. The major missing component of this submission is an evaluation of how accurately the numerical simulation models the real scenario. If this is possible from the data, providing this information would drastically increase the potential impact of the paper.
2. The model comparisons aren't very convincing due to the vastly different scales between them. It would improve the submission as a benchmarking effort to apply some form of normalization - FLOPs, parameter count, run time on fixed hardware - but I'm largely treating this as a dataset paper so not marking off for this.
3. Similarly, one-step prediction is a useful reference for how well the models do what they're trained for, but longer rollout evaluations are more reflective of real tasks. I'd add more details on longer autoregressive rollouts if possible.
4. Frequency domain error (in space) is generally going to be more informative when normalized. In 4.5 the text notes that a decrease of error in the high frequency is remarkable, but it's actually very much expected when comparing absolute metrics. Most fluid systems follow a polynomial decay law in spatial frequency, so the values being compared should be much smaller. It is actually concerning (for the baselines, not this submission) that this seems to be rare.

Minor notes
1. 3.1 prediction task - this is all discretized data, so it feels inaccurate to describe the task as a mapping between the continuous spaces.
2. (paragraph beginning at 155) It would make sense to also include a canonical validation/test set of simulated data to evaluate the level of overfitting to simulation data and the overall sim2real gap.
3. Table 4 is pretty hard to read. I don't know that this is the best way to demonstrate this data.

**Questions:**

1. Often one of the cited limitations of numerical simulation is the difficulty of simulating regimes (Reynolds/schmidt/ect) that occur commonly in real world settings. For CFD, the datasets currently contain mostly DNS data. Have you considered further comparisons between experimental data in regimes where DNS struggles and the more approximate simulations used to model them?
2. Do any of the models evaluated compare favorably to the numerical simulation for predicting the state of fields? Is the pairing sufficiently close that this is a question it is possible to ask with this data?

---

> ### Author Response · Authors · 2025-11-20
> **Official Response to Reviewer YXXo (1)**
>
> We sincerely appreciate your positive feedback, including comments such as valuable, excellent presentation, extensive appendix, and well planned paper. We are also grateful for the constructive suggestions you provided, which helped us further improve the quality of the paper. Below we present our responses to your comments, with the corresponding revisions highlighted in blue in the updated manuscript.
>
> >1. The major missing component of this submission is an evaluation of how accurately the numerical simulation models the real scenario. If this is possible from the data, providing this information would drastically increase the potential impact of the paper.
>
> A: This is indeed an interesting question. Directly applying many existing metrics to quantify the discrepancy between simulated and real-world data is challenging, primarily because some wave modes of both data affect statistics. Comparing the two requires care about some summary statistics. In fact, when generating the simulated data, we have adopted several established summary metrics from the fluid and combustion to characterize the differences between the two data. And we have now added them to the code repository (https://anonymous.4open.science/r/realbench) and the revised manuscript.
>
> Specifically, for four fluid datasets, we employ the **mean velocity profile** following previous numerical studies of fluid literatures [1-4]. We select different $(x/D, y/D)$ positions to calculate the differences between ground truth time-average velocity fields of real-world data, and simulated data, where $D$ is the diameter of the cylinder. For example, on the Cylinder data, we take $x/D=1, 2, 3, 4$ and 9 uniform $y/D$ values as the positions of probes, and measure the difference between simulated data and real-world data, which is visualized in Fig. 11 of the revised manuscript. From the figure, we can see that in $x/D$=1 and $y/D \in {(-0.5, 0.5)}$ region, $u$ will decrease, and this decrease will decay with the increase of $x/D$, which is consistent with previous studies. We also calculate the **mean velocity profile error (MVPE)**, the MAE between the mean velocity profile of simulated data and real-world data, as shown below. In the revised manuscript, the introduction of MVPE is added to Sec. 3.3.2 Physics-oriented Metrics, and this table is added to Appendix C.4 as Table 8.
>
> | Cylinder  | Controlled cylinder | FSI     | Foil    |
> |-----------|---------------------|---------|---------|
> | 0.08718   | 0.06985             | 0.11440 | 0.08653 |
>
> For the combustion data, we implement **frequency-domain analysis**, focusing on OH radical mole fraction dynamics. The experimental instantaneous OH chemiluminescence signal is provided by measurement, while the equivalent transient signal is extracted from the LES results by performing a spatial integration of the instantaneous OH concentration field. Both time series are detrended to remove the DC component, and transformed into the frequency domain using the Fast Fourier Transform (FFT), showing the dominant frequencies of the combustion instability. The results and figures are added to Appendix C.4 in the revised manuscript.
>
> [1] Kravchenko A G, Moin P. Numerical studies of flow over a circular cylinder at Re D= 3900[J]. Physics of fluids, 2000, 12(2): 403-417.
>
> [2] Ma X, Karamanos G S, Karniadakis G E. Dynamics and low-dimensionality of a turbulent near wake[J]. Journal of fluid mechanics, 2000, 410: 29-65.
>
> [3] Wissink J G, Rodi W. Numerical study of the near wake of a circular cylinder[J]. International journal of heat and fluid flow, 2008, 29(4): 1060-1070.
>
> [4] Neunaber I, Yadala S, Hearst R J. Induced periodicity in wakes[J]. Journal of Fluid Mechanics, 2025, 1022: R3.
>
> >2. The model comparisons aren't very convincing due to the vastly different scales between them. It would improve the submission as a benchmarking effort to apply some form of normalization - FLOPs, parameter count, run time on fixed hardware - but I'm largely treating this as a dataset paper so not marking off for this.
>
> A: Thanks for your suggestion. Because the baseline models include the foundation model DPOT and we perform hyperparameter tuning for all models, it is difficult to strictly align the parameter sizes across architectures. Also, the diffusion model-based method WDNO usually has longer run time due to the multiple-step sampling process. To mitigate the influence of scale and enable a fairer comparison, one feasible approach would be to evaluate all models under several different parameter budgets and plot the resulting performance curves, allowing us to assess each model’s capability based on its trend rather than a single parameter setting.

---

> ### Author Response · Authors · 2025-11-20
> **Official Response to Reviewer YXXo (2)**
>
> >3. Similarly, one-step prediction is a useful reference for how well the models do what they're trained for, but longer rollout evaluations are more reflective of real tasks. I'd add more details on longer autoregressive rollouts if possible.
>
> A: Thank you for the suggestion. In the revised manuscript, we now report the autoregressive results on the Cylinder dataset for both **5-round and 10-round rollouts**, corresponding to total prediction horizons of 100 and 200 steps, respectively. The results are also summarized in the table below and are added to Fig. 3(c) in the revised manuscript.
> |RMSE|FNO|WDNO|UNet|DeepONet|MWT|CNO|GK-Transformer|Transolver|DPOT-S-FT|DPOT-L-FT|DMD|
> |-|-|-|-|-|-|-|-|-|-|-|-|
> | 5 rounds  | 0.08352 | 0.07270 | 0.08849 | 0.09644| 0.07154 | 0.10907 | 0.10259 | 0.09421 | 0.07815 |0.07653|0.11710|
> | 10 rounds | 0.10457 | 0.08616 | 0.09641 | 0.10133| 0.08503 | 0.14075 | 0.10629 | 0.09904 | 0.08981|0.08833|0.13149|
>
> >4. Frequency domain error (in space) is generally going to be more informative when normalized. In 4.5 the text notes that a decrease of error in the high frequency is remarkable, but it's actually very much expected when comparing absolute metrics. Most fluid systems follow a polynomial decay law in spatial frequency, so the values being compared should be much smaller. It is actually concerning (for the baselines, not this submission) that this seems to be rare.
>
> A: Thank you for the insightful suggestion. In the table below, we present the normalized low-, mid-, and high-frequency fRMSE on Foil. These results are also presented in Sec. 4.5 of the revised version now. Results reveal that with the increase of the frequency, the normalized fRMSE gets larger, showing that it is harder for the deep learning models to capture high-frequency information.
>
> |  | FNO| WDNO | UNet    | DeepONet  | MWT | CNO | GK-Transformer | Transolver | DPOT-S-FT | DPOT-L-FT | DMD|
> |-|-|-|-|-|-|-|-|-|-|-|-|
> | Normalized low fRMSE  | 0.15657 | 0.12579 | 0.11282 | 0.31372   | 0.15063 | 0.15857 | 0.21014 | 0.29850| 0.13886   | 0.16730| 0.35305 |
> | Normalized mid fRMSE  | 0.30214 | 0.21934 | 0.20557 | 0.65962   | 0.32969 | 0.33139 | 0.41251| 0.54770 | 0.26997   | 0.24073| 0.96544 |
> | Normalized high fRMSE | 0.54577 | 0.40079 | 0.38713 | 0.88928   | 0.59697 | 0.54957 | 0.62780| 0.77847 | 0.48679   | 0.44469| 1.28496 |
>
> >5. 3.1 prediction task - this is all discretized data, so it feels inaccurate to describe the task as a mapping between the continuous spaces.
>
> A: Thank you for your advice. In the revised version, we have removed the word "infinite-dimensional" and pointed out that the states are discretized.
>
> >6. (paragraph beginning at 155) It would make sense to also include a canonical validation/test set of simulated data to evaluate the level of overfitting to simulation data and the overall sim2real gap.
>
> A: We agree with this suggestion. To increase flexibility for users, we now introduce **an additional argument "split_numerical" in the "RealDataset" module** in the code repository. This option controls whether the numerical data should be partitioned into training, validation, and test subsets. This design enables users to easily switch between different experimental settings depending on their research objectives. We also update the introduction of the "RealDataset" module in Appendix D.3.
>
> >7. Table 4 is pretty hard to read. I don't know that this is the best way to demonstrate this data.
>
> A: To improve readability, in Appendix B.2, we add Table 5 that reports the **RMSE, Relative L2, and fRMSE metrics for all models under 1-round, 2-round, and 3-round autoregressive evaluation**. This table enables a clearer comparison of how each model's performance evolves as the prediction horizon increases, allowing readers to better understand the temporal robustness and long-term stability of different approaches.

---

> > ### Author Response · Authors · 2025-11-20
> > **Official Response to Reviewer YXXo (3)**
> >
> > >8. Often one of the cited limitations of numerical simulation is the difficulty of simulating regimes (Reynolds/schmidt/ect) that occur commonly in real world settings. For CFD, the datasets currently contain mostly DNS data. Have you considered further comparisons between experimental data in regimes where DNS struggles and the more approximate simulations used to model them?
> >
> > A: Thanks for your question. Both the DNS and experimental measurements for high Reynolds numbers are challenging and interesting problems, and one of the key issues in fluid mechanics. This is the key gap in applying laboratory study (including numerical and experimental) to the real world. For example, the Reynolds number of the ultra-high speed in the air can reach $10^8$, which is extremely difficult for numerical computation and experimental measurement to capture multi-scale turbulence phenomena. Our current experimental equipment is unable to conduct such experiments. We hope to use this paper as an exploration of the application of fluid real-world data and simulated data in ML, and encourage other teams with available data to further improve the quality of the dataset. We are continuously open to this and welcome collaborators to join this project and contribute to the field of scientific ML.
> >
> > >9. Do any of the models evaluated compare favorably to the numerical simulation for predicting the state of fields? Is the pairing sufficiently close that this is a question it is possible to ask with this data?
> >
> > A: Thanks for the insightful question! As described in the answer to Weakness 1, we adopt **MVPE** to evaluate the discrepancy between model predictions and real-world measurements, as well as between simulated and real-world data. Specifically, we compute MVPE over a 200-step temporal window. Model predictions are obtained by running ten rounds of autoregression to predict 200 steps. For the simulated and real data, we separately compute the norm of the velocity field, select for each the time step at which the norm attains its minimum as the starting time point, and then extract the subsequent 200 time steps as the time window.
> >
> > The resulting MVPE for the simulated data is 0.05650. The table below reports the MVPE of each model under two training settings: Simulated training and Real-world finetuning. We observe that the MVPE of models pretrained on simulated data can be either larger or smaller than the MVPE of the simulated data itself. Smaller MVPE of models may be because the model's input is the initial state of real-world data, which has part of real data information, such as measurement noise distribution and basic modes. However, after finetuning on real-world data, all models achieve lower MVPE than simulated data, highlighting both the importance of real-world measurements and the potential of deep learning models. In the revised version, the discussions are added to Appendix C.4, and the visualizations of mean velocity profiles are provided in Fig. 5(b), 7, and 8.
> >
> > |                       | FNO     | WDNO    | UNet    | DeepONet | MWT     | CNO     | GK-Transformer | Transolver | DPOT-S-FT | DPOT-L-FT | DMD     |
> > |-----------------------|---------|---------|---------|----------|---------|---------|----------------|------------|-----------|-----------|---------|
> > | Simulated training    | 0.07418 | 0.05639 | 0.03137 | 0.05295  | 0.03547 | 0.06409 | 0.05431        | 0.08456    | 0.03378   | 0.03394   | 0.10668 |
> > | Real-world finetuning | 0.01317 | 0.02181 | 0.01405 | 0.01787  | 0.01821 | 0.03488 | 0.01485        | 0.02374    | 0.01363   | 0.01250   | 0.10668 |

---

### Official Review · Reviewer_1VNT · 2025-10-31

**Soundness:** 4
**Presentation:** 4
**Contribution:** 4
**Rating:** 10
**Confidence:** 4

**Summary:**

The paper introduces RealBench, which evaluates the generalizability of simulated-data trained ML models on real-world measurements. It is well written, and covers an obvious gap in the literature. The authors clearly put significant thought into solving this issue, and it appears that the paper is as close to reproducible as is possible. II see they even posted their code on anonymous github. I looked through their code and it has the same quality/usability as PDEBench, which has set the standard for this kind of release.

**Strengths:**

The paper is original, and tackles a difficult challenge of pairing real and simulated data. This a clear gap in the literature, where simulated data was the only solution before. This provides a unique insight into many of the claims made on various architectures aimed at training surrogates for PDEs. It is comprehensive, well written, and thoughtfully formulated (especially the figures). It appears to achieve the maximum possible level of reproducibility through the use of the anonymous github repo.

**Weaknesses:**

The only weakness is the obvious one - out of domain regimes are not covered. However, this is probably the biggest area of weakness for this area of study as a whole. The complexity of fluid dynamics makes that a separate challenge entirely (one I dont see being solved any time soon). Surrogates typically cover some precise range of reynolds numbers around a specific geometry. It is the nature of this domain.

**Questions:**

I don't have any questions.

---

> ### Author Response · Authors · 2025-11-20
> **Official Response to Reviewer 1VNT**
>
> Thanks very much for commending our paper as unique, tackling a difficult challenge, comprehensive, well written, and thoughtfully formulated. Below we respond to your concern. We highlight the revisions in the manuscript using blue text.
>
> >1. The only weakness is the obvious one - out of domain regimes are not covered. However, this is probably the biggest area of weakness for this area of study as a whole. The complexity of fluid dynamics makes that a separate challenge entirely (one I dont see being solved any time soon). Surrogates typically cover some precise range of reynolds numbers around a specific geometry. It is the nature of this domain.
>
> A: Thanks for the constructive suggestion! In fact, our code already supports the out-of-domain (OOD) evaluation functionality, and we have reserved a subset of OOD parameter regimes in the test set when constructing the dataset. Since we agree that this issue is important challenging for deep learning models, we expect that many users will find this functionality useful.
>
> Specifically, as described in Appendix D.3, the "RealDataset" module includes a parameter 'test_mode', which can be set to 'all', 'in_dist', 'out_dist', 'seen', or 'unseen' to select data in the test set corresponding to different parameter regimes. Here, 'in_dist' refers to parameters in the distribution of the training set, while 'out_dist' refers to the out-of-distribution parameters. And 'seen' refers to parameters that appear in the training set, while 'unseen' refers to parameters that do not. To highlight this functionality, we have additionally expanded the description of 'test_mode' and the OOD task in Appendix D.3 of the updated manuscript.

---

### Official Review · Reviewer_kvd1 · 2025-10-31

**Soundness:** 2
**Presentation:** 2
**Contribution:** 3
**Rating:** 6
**Confidence:** 3

**Summary:**

The paper describes a new physics-oriented benchmark and dataset that focuses on the gap between simulated and real experimental data.

The paper presents $5$ applications and provides metrics for evaluation, alongside classical and recent benchmark methods.

The description of the "TASK DEFINITION" should be improved and linked to the experiments, for example, in Table.1.

The dataset is not well described; a summary table is necessary (how many samples, what is the size of each sample, what is the total real time duration, step size, dimension of the problem 1d, 2d, 3d, ...).

The authors provide code to run the experiment, some sample data (real and simulated), but do not provide the script to generate the numerical data.

**Strengths:**

This dataset addresses the important aspect of bridging the gap between the simulated (numerical) and the physical system.

While is not possible to cover a large experimental setup, the authors provide $5$ taks with both numerical and physical data.

This work will therefore help in evaluating new models, even if may not cover all possible scenarios.

**Weaknesses:**

It is hard to say, but it is not possible to cover all possible physical experimental conditions. Nevertheless, the paper is a good contribution in the right direction.

The main point is that the numerical generation scripts are missing, therefore not possible to extend the data (at least numerical) to other scenarios.

On the experimental side, I am not able to judge if the information is sufficient.

I found the task description disconnected to the actual experiments. I would encourage the authors to improve that section.

**Questions:**

One possible source of difference between experimental and numerical experiments is the measurement noise, but i can assume there could be a larger difference, for example, if the state is not directly measurable (pressure is not available, but only velocity).

Could the author expand and position this paper in this context? What are the possible main and most critical differences between experiments and numerical simulations?

---

> ### Author Response · Authors · 2025-11-20
> **Official Response to Reviewer kvd1 (1)**
>
> Thank you for your recognition and constructive suggestions. Below we address the reviewer's concerns. The modifications have been highlighted in blue in the revised version.
>
> >1. The dataset is not well described; a summary table is necessary (how many samples, what is the size of each sample, what is the total real time duration, step size, dimension of the problem 1d, 2d, 3d, ...).
>
> A: In fact, Appendix C.1 of the original submission **already includes a summary table of the dataset**, and Sec. 3.2 refers readers to this appendix (l194). This table **contains the information you mentioned**, such as the number of samples, the size of each sample, the total real-time duration, and the problem dimensionality. It also includes additional details such as resolution and memory usage. Based on your suggestion, we have **added the step size** of the data in the revised manuscript.
>
> >2. The main point is that the numerical generation scripts are missing, therefore not possible to extend the data (at least numerical) to other scenarios.
>
> A: Thanks for your comment. The numerical generation of fluid data are the open-source CFD code developed by [1]. You can find the script in **https://github.com/WaterLily-jl/WaterLily.jl and https://github.com/weymouth/lily-pad**. We have added links in Appendix C.2 of the revised manuscript for future reference in expanding to other scenarios. We modified different boundary conditions and physical parameters based on the examples provided by this code to generate the simulation data we provided. We provide a numerical generation script of Foil data as an example in the anonymous repository (https://anonymous.4open.science/r/realbench/real_benchmark/numerical_scripts/ThreeD_NACA.jl). After following Waterlily's tutorial, the script can be run directly. We promise to **open-source all used numerical scripts** in the camera-ready version.
>
> As for the combustion data, the simulations are performed using the commercial software **STAR-CCM+ 2022.1** [2]. The computational framework utilizes a pressure-based, segregated solver for the coupled solution of the governing equations. Near-wall flow physics are captured using an all y+ wall treatment. For modeling turbulent combustion, the Eddy Dissipation Concept (EDC) is applied to account for finite-rate chemistry effects within the turbulent flame structure [3]. The combustion chemistry is described by a reduced chemical mechanism for ammonia-methane co-firing, comprising 38 species and 184 elementary reactions [4]. The stiff chemical kinetics are efficiently integrated using the CVODE solver. These descriptions are added to Appendix C.2 in the revised version for future extensions.
>
> [1] Weymouth, G D, etc. Boundary data immersion method for Cartesian-grid simulations of fluid-body interaction problems[J]. Journal of Computational Physics, 2011.
>
> [2] Siemens Digital Industries Software. Simcenter STAR-CCM+, version 2022.1, Siemens 2022.
>
> [3] J.-M. Lourier, etc. Scale adaptive simulation of a thermoacoustic instability in a partially premixed lean swirl combustor. Combustion and Flame, 183:343–357, 2017.
>
> [4] Jihao Sun, etc. Numerically study of ch4/nh3 combustion characteristics in an industrial gas turbine combustor based on a reduced mechanism. Fuel, 327:124897, 2022.
>
> >3. On the experimental side, I am not able to judge if the information is sufficient.
>
> A: To address your concerns, we have added **additional details about the experimental data collection process** in Appendix C.3 of the newly revised manuscript, where we describe the experimental setup and parameter specifications in detail. Specifically, we introduce parameter information of the circulating water tank (including the visualization in Fig. 9), details on parameter settings and attached devices for five datasets (including Fig. 10 for clear demonstration), detailed information on PIV measurements, and an introduction to post-processing software. In addition, we have included some examples of PIV raw data and calibration files in anonymous links:  https://drive.google.com/drive/u/0/folders/1ucRS-5u7eSnzLkqaIPtFpVNPGpr6yYug?dmr=1&ec=wgc-drive-hero-goto and clearly indicated the link to the post-processing software in Appendix C.3 of the revised manuscript for the fluid mechanics community to verify and reference.
>
> >4. I found the task description disconnected to the actual experiments. I would encourage the authors to improve that section.
>
> A: Actually, in the task description of the original submission, we explicitly define three tasks, real-world training, simulated training, and simulated pretraining with real-world finetuning (l162-164), and introduce them one by one (l168-171). To further address your concern, for clarity, we have **added the clarification** that "simulated pretraining with real-world finetuning is abbreviated as real-world finetuning in the Experiment section" in the revised manuscript.

---

> > ### Author Response · Authors · 2025-11-20
> > **Official Response to Reviewer kvd1 (2)**
> >
> > >5. One possible source of difference between experimental and numerical experiments is the measurement noise, but i can assume there could be a larger difference, for example, if the state is not directly measurable (pressure is not available, but only velocity).
> >
> > A: Thanks for your question. In fact, in the original submission, "the state is not directly measurable" corresponds to **unmeasured modalities** in the paper. In Introduction, we explain modalities as variables of the physical system and point out that modalities of simulated and real-world data are different. This is mentioned in multiple places, including but not limited to Fig. 1 and its caption (l22), the Introduction (l68), Sec. 3.1 Task Definition (l159), and Sec. 4.2 Gap of Real-world and Simulation. We have also designed training techniques, randomly masking unmeasured modalities, as introduced in Sec. 4.2. For clarification, we again explain the definition of modalities in Sec. 3.1 Task Definition (l159) in the revised manuscript.
> >
> > >6. Could the author expand and position this paper in this context? What are the possible main and most critical differences between experiments and numerical simulations?
> >
> > A: Thanks for your question. In the original manuscript, we have discussed the cost of measurement, noise, and states (like pressure) that cannot be directly measured, while simulation may have numerical errors from discretization and simplified physics. Based on your suggestion, we have added a **clearer and more detailed description of the sources of the differences in l155-158** of the revised manuscript. Real-world data avoid numerical errors and simplified physics but is costly, noisy, and often limited in observability. For example, the incoming flow cannot be strictly guaranteed to be uniform, and camera noise leads to measurement errors. Simulated data are relatively cheaper and offer broader modalities with dense parameter coverage, yet suffer from numerical errors that are often caused by closure modeling like Large Eddy Simulation (LES) and discretization like second-order convergence.
> >
> > We hope that the above have addressed your comments. If you have any further concerns, please feel free to reach out to us, and we will be happy to provide additional clarification.

---

### Official Review · Reviewer_6AF9 · 2025-11-04

**Soundness:** 2
**Presentation:** 2
**Contribution:** 2
**Rating:** 4
**Confidence:** 4

**Summary:**

This paper proposes a benchmark that pairs real measurements with matched numerical simulations across five scenarios (cylinder wake, controlled cylinder, FSI, foil, and combustion; governing equations span Navier-Stokes, coupled FSI, and reactive Navier-Stokes with species transport). The benchmark defines three training regimes (train on simulation, train on real, pretrain on simulation then finetune on real), includes a mix of pixel and physics flavored metrics, and evaluates a set of neural PDE baselines including a pretrained foundation model. The headline empirical messages are: there is a nontrivial gap between simulation and laboratory data; pretraining on simulation generally helps downstream on real; and the codebase makes it straightforward to add models or datasets.

I think this is timely and potentially useful. If we are serious about sim2real for scientific ML, we need carefully curated real data and a shared protocol. However, the current paper mixes benchmarking with a sim2real narrative in ways that are a bit loose, the experimental details are too thin for others to trust or extend the datasets, and some of the metrics and literature framing are not well aligned with fluid mechanics and combustion practice.

Am open to potentially increasing my score if the authors narrow the claims, remove the robotics sim2real framing argument and instead point to real sim2real problems in fluid dynamics, add significantly more context to place this in the existing fluid dynamics literature, and substantially improve the experimental documentation and physics grounded evaluation.

**Strengths:**

- The benchmark collects paired real and simulated trajectories for several nontrivial systems instead of yet another synthetic-only PDE suite. This is likely to be useful for the community.
- The split of training regimes (simulation only, real only, pretrain on simulation then finetune on real) is useful, and the pretraining result is consistent with what many of us have seen in practice.
- The code appears modular enough to add a new dataset or baseline without painful surgery, and using a single file format lowers friction for adoption.
- Including both data oriented metrics and physics oriented diagnostics is better than reporting only RMSE. The autoregressive evaluation option is also a good idea.
- The combustion scenario is ambitious and, if documented properly, could become a valuable stress test beyond the usual laminar toy problems.
- The baseline measurements for their benchmark are very extensive.

**Weaknesses:**

Major concerns:

- The documentation about the experiment is unacceptably thin in its current form. If the experimental data was created or modified from another source, you need to cite it. If someone else created the dataset for you, you need them to write documentation for it. The current documentation on experimental data generation (which, of course, can be included in the appendix) is simply unacceptable for publication, especially for a paper which is supposed to be about this very dataset.
- The paper motivates the sim2real gap by citing mostly work from robotics, which I found very strange, almost as if the authors are guessing there is a sim2real gap in fluids, without actually surveying the literature. Robotics has a much different sim2real gap than turbulence research does. In fluids, the sources of discrepancy, data acquisition, and noise models can be quite different. Please ground the narrative in fluids and combustion references. If you are addressing the sim2real gap in fluids, you must speak from the context of the fluids community, and discuss the ways in which the fluids community has quantified this gap.
- Please state clearly whether you will release raw data (e.g., the PIV frames), calibration files, and the full processing scripts, not just the final HDF5 arrays, so that it can be checked by others. If raw data cannot be released, say so and justify it. Benchmarks live or die by their data hygiene, and biases in a benchmark can leak into biases in the community's preferred models.
- Several figures quantify differences using frequency or Fourier errors over image-like arrays. This is admittedly a start, but it is not a physics grounded measure of mismatch between experiment and simulation, and certainly not something people in the fluids community would actually use as a robust measure of discrepancies between simulation and real data. In fact, for several real-world experimental problems, there is a difference here that is simply due to the nature of the real-world experiment, yet the simulation and experiment can actually have no discrepancy. This is because you would only care about some summary statistic, and not care about some wave mode that you know does not affect your statistic. Yet, your metric would completely miss this. I recommend reviewing and citing the fluids literature and how people measure discrepancies between simulations and real data. Note that this is a problem people have studied for literally decades in the fluids community. These additions would make your claims about a "gap" more convincing.
- Simulation contains modalities that are not observed in the lab, and the current strategy randomly masks channels and adds noise. That is a start, but it does not reflect the actual sensor physics. Please consider sensor specific degradations (camera noise models, optical blur, saturation, PIV algorithmic artifacts) and state explicitly which channels are used for training and which are hidden. It would also help to define tasks that force parity, for example training all models only on the modalities that the lab provides.
- The paper claims to be the first benchmark that integrates real-world measurements with paired numerical simulations across complex physical systems. Within fluids, this is far from true; as just one demonstration, "ERCOFTAC" has hosted combined experimental and numerical reference cases since 1995 across a wide range of flows. Please narrow the novelty claim to something that accurately represents existing datasets and benchmarks, and perhaps cite existing databases.


Additional comments and suggestions

- The update ratio metric is interesting, but it conflates pretraining data scale with optimization effects.
- Please make the train, validation, and test splits explicit at the parameter level so that generalization across Reynolds number, control frequency, mass ratio, or equivalence ratio is clear. I consider those to be most interesting axes. And if you already do this, show it more prominently.
- The autoregressive evaluation stops very early. If you want to make claims about stability, show longer horizons and add probe based diagnostics, not only field RMSE.
- The baselines are modern ML models, which is fine for a benchmark, but the story would be stronger if you add one or two domain baselines for each scenario (for example a simple reduced order model, or even a physics based filter) so readers have a calibration point.
- Throughout the paper there are small terminology issues. I would prefer "numerical error" over "computational error" (computational error sounds like a code bug). Be precise about whether errors arise from discretization, closure modeling, boundary conditions, or measurement.
- Not a criticism, but I think it would be better to rename the benchmark. "RealBench" seems far too broad and will clash with many domains. Something like "RealPDEBench" or "RealFlowBench" might be more appropriate.

**Questions:**

[Several questions discussed above]

---

> ### Author Response · Authors · 2025-11-20
> **Official Response to Reviewer 6AF9 (1)**
>
> Thank you for your recognition and the insightful suggestions. Here are our responses to your comments. The corresponding modifications have been highlighted in blue in the revised manuscript.
>
> >1. Documentation about experiments is unacceptably thin in its current form. If experimental data was created or modified from another source, you need to cite it. If someone else created the dataset for you, you need them to write documentation for it. The current documentation on experimental data generation (which, of course, can be included in the appendix) is simply unacceptable for publication, especially for a paper which is supposed to be about this very dataset.
>
> A: Thank you for the suggestion! First, all experimental data are collected through **our own experiments**, not sourced from existing datasets. Second, as our focus is on the field of **Scientific ML**, we currently emphasize describing the **processed datasets that are ready for direct use by deep learning models**, with relatively less attention given to the experimental data collection process. To make the description of datasets more comprehensive, based on your suggestion, we now include **detailed documentation on experimental data collection in Appendix C.3** in the revised manuscript, where we describe the experimental setup and parameter specifications in detail. Specifically, we introduce parameter information of the circulating water tank (with the visualization in Fig. 9), details on parameter settings and attached devices for five datasets (with Fig. 10 for clear demonstration), detailed information on PIV measurements, and an introduction to post-processing software.
>
> >2. The paper motivates the sim2real gap by citing mostly work from robotics, which I found very strange, almost as if the authors are guessing there is a sim2real gap in fluids, without actually surveying the literature. Robotics has a much different sim2real gap than turbulence research does. In fluids, the sources of discrepancy, data acquisition, and noise models can be quite different. Please ground the narrative in fluids and combustion references. If you are addressing the sim2real gap in fluids, you must speak from the context of the fluid community and discuss the ways in which the fluid community has quantified this gap.
>
> A: Thank you for pointing this out. We agree that the sim2real gap in robotics is very different from that in fluids and combustion. However, we would like to clarify that, except for this article [1], all the references in our original manuscript do not involve robotics, and we aim to discuss the problem of sim2real **from the perspective of the fluids and combustion communities**. We have cited some **references of validation in numerical studies** in the revised manuscript to testify the gap between sim and real in fluids and combustion. Additionally, [1] has now been **removed from the Introduction**, and we have **replaced it with references related to sim2real in fluids** [2-6] in the revised manuscript.
>
> [1] Wenshuai Zhao, etc. Sim-to-real transfer in deep reinforcement learning for robotics: a survey. 2020 IEEE Symposium Series on Computational Intelligence (SSCI), 2020.
>
> [2] Roache P J. Verification and validation in computational science and engineering[M]. Albuquerque, NM: Hermosa, 1998.
>
> [3] Oberkampf W L, etc. Verification and validation in computational fluid dynamics[J]. Progress in aerospace sciences, 2002.
>
> [4] Veynante D, etc. Turbulent combustion modeling[J]. Progress in energy and combustion science, 2002.
>
> [5] Hochgreb S. Mind the gap: Turbulent combustion model validation and future needs[J]. Proceedings of the Combustion Institute, 2019.
>
> [6] Kravchenko A G, etc. Numerical studies of flow over a circular cylinder at Re D= 3900[J]. Physics of fluids, 2000.
>
> >3. Please state whether you will release raw data (e.g., the PIV frames), calibration files, and full processing scripts, not just the final HDF5 arrays, so that others can check it. If raw data cannot be released, say so and justify it. Benchmarks live or die by their data hygiene, and biases in a benchmark can leak into biases in the community's preferred models.
>
> A: Thank you for the kind reminder! We are very willing to make all raw data, calibration files, and processing scripts publicly available. Currently, we provide some examples of raw data and calibration files during the peer review stage (https://drive.google.com/drive/u/0/folders/1ucRS-5u7eSnzLkqaIPtFpVNPGpr6yYug?dmr=1&ec=wgc-drive-hero-goto.), and more raw data will be made public together with the dataset in the camera-ready. We have now **added this statement in the Reproducibility Statement section**. Our PIV frames are processed using two software, MicroVec and PIVlab. PIVlab is an open-source software, but MicroVec is a commercial software purchased from a company. The names and links have been added in Appendix C.3 of the revised manuscript, and the processing process has been introduced.

---

> ### Author Response · Authors · 2025-11-20
> **Official Response to Reviewer 6AF9 (2)**
>
> >4. Several figures quantify differences using frequency or Fourier errors over image-like arrays. This is admittedly a start, but it is not a physics grounded measure of mismatch between experiment and simulation, and certainly not something people in the fluids community would actually use as a robust measure of discrepancies between simulation and real data. In fact, for several real-world experimental problems, there is a difference here that is simply due to the nature of the real-world experiment, yet the simulation and experiment can actually have no discrepancy. This is because you would only care about some summary statistic, and not care about some wave mode that you know does not affect your statistic. Yet, your metric would completely miss this. I recommend reviewing and citing the fluids literature and how people measure discrepancies between simulations and real data. Note that this is a problem people have studied for literally decades in the fluids community. These additions would make your claims about a "gap" more convincing.
>
> A: Thank you for the suggestion. We agree that summary statistics are very important. In fact, when generating the simulated data, we **already performed comparisons of summary statistics** between the simulated and real-world data, although we did not present these results explicitly in the original manuscript. We are happy to enrich the manuscript and provide these metrics for readers who are interested in comparing the simulated data.
>
> Specifically, for four fluid data, we employ the **mean velocity profile** following previous numerical studies of fluid literatures [1-4]. We select different $(x/D, y/D)$ positions to calculate the differences between ground-truth time-average velocity fields of real-world data, and simulated data, where $D$ is the diameter of the cylinder. For example, on the Cylinder data, we take $x/D=1, 2, 3, 4$ and 9 uniform $y/D$ values as the positions of probes, and measure the difference between simulated data and real-world data, which is visualized in Fig. 11 of the revised manuscript. From the figure, we can see that in $x/D$=1 and $y/D \in {(-0.5, 0.5)}$ region, $u$ will decrease, and this decrease will decay with the increase of $x/D$, which is consistent with previous studies. We also calculate the **mean velocity profile error (MVPE)**, the MAE between the mean velocity profile of simulated data and real-world data, between simulated and real-world data, as shown below. In the revised manuscript, the introduction of MVPE is added to Sec. 3.3.2 Physics-oriented Metrics, and this table is added to Appendix C.4 as Table 8. The relevant codes are updated in the anonymous repository (https://anonymous.4open.science/r/realbench) as well.
>
> | Cylinder  | Controlled cylinder | FSI     | Foil    |
> |-----------|---------------------|---------|---------|
> | 0.08718   | 0.06985             | 0.11440 | 0.08653 |
>
> As for the combustion data, we implement **frequency-domain analysis**, focusing on OH radical mole fraction dynamics. The experimental instantaneous OH chemiluminescence signal is provided by measurement, while the equivalent transient signal is extracted from the LES results by performing a spatial integration of the instantaneous OH concentration field. Both time series are detrended to remove the DC component, and transformed into the frequency domain using the Fast Fourier Transform (FFT), showing the dominant frequencies of the combustion instability. The results and figures are added to Appendix C.4 in the revised manuscript.
>
> [1] Kravchenko A G, Moin P. Numerical studies of flow over a circular cylinder at Re D= 3900[J]. Physics of fluids, 2000, 12(2): 403-417.
>
> [2] Ma X, Karamanos G S, Karniadakis G E. Dynamics and low-dimensionality of a turbulent near wake[J]. Journal of fluid mechanics, 2000, 410: 29-65.
>
> [3] Wissink J G, Rodi W. Numerical study of the near wake of a circular cylinder[J]. International journal of heat and fluid flow, 2008, 29(4): 1060-1070.
>
> [4] Neunaber I, Yadala S, Hearst R J. Induced periodicity in wakes[J]. Journal of Fluid Mechanics, 2025, 1022: R3.

---

> ### Author Response · Authors · 2025-11-20
> **Official Response to Reviewer 6AF9 (3)**
>
> >5. Simulation contains modalities that are not observed in the lab, and the current strategy randomly masks channels and adds noise. That is a start, but it does not reflect the actual sensor physics. Please consider sensor specific degradations (camera noise models, optical blur, saturation, PIV algorithmic artifacts) and state explicitly which channels are used for training and which are hidden. It would also help to define tasks that force parity, for example training all models only on the modalities that the lab provides.
>
> A: First, regarding the noise models, we are **using the most common type of noise**, which is Gaussian noise, a type of the camera noise models. Based on your suggestion, we have **added a plug-and-play feature in the code to allow the selection of different noise types** in the 'RealDataset' Module (Appendix C.3 and file 'real_benchmark/data/dataset.py'), including Gaussian noise, Poisson noise, and optical blur (2D Gaussian blur). Users can freely choose the type of noise during training.
>
> Second, in Sec. 4.1 of the original submission, we have explicitly mentioned that, when **using simulated data for training, we apply masking for modalities that are not measured in real-world experiments with a certain probability**. The different modalities included in the simulated and real-world datasets are shown in **Fig. 1**. In the revised manuscript, we provide a clear record of the modalities included in the simulated and real-world data for each dataset in **Table 7 of Appendix C.1**.
>
> To implement "training all models only on the modalities that the lab provides," users simply need to set **mask_prob=1**. In our experiments, we set a value smaller than 1 to **make fuller use of the information from all modalities** in simulated data.
>
> >6. The paper claims to be the first benchmark that integrates real-world measurements with paired numerical simulations across complex physical systems. Within fluids, this is far from true; as just one demonstration, "ERCOFTAC" has hosted combined experimental and numerical reference cases since 1995 across a wide range of flows. Please narrow the novelty claim to something that accurately represents existing datasets and benchmarks, and perhaps cite existing databases.
>
> A: First, we clarify that the scope is a **benchmark**, meaning that our work not only provides data but also designs experiments specifically for the datasets, provide a complete code platform that includes model training, testing, and other functionalities, and we implement and compare state-of-the-art machine learning models. Second, our dataset **pairs each real-world trajectory with a corresponding simulated trajectory**  (set the same numerical parameters to match the experimentally measured parameters), and each dataset contains **multiple operating conditions** to meet the requirements of scientific ML datasets.
>
> Therefore, the ERCOFTAC database, as mentioned, does not meet the definition of a benchmark, nor does it provide real-world measurements with paired simulated data. It only provides references where numerical solutions using experimental data are discussed, but does not offer data generated by numerical algorithms. We believe this highlights the contribution and effort of our work to the scientific ML community.
>
> However, to provide a more precise definition, in the revised version of our paper, we have changed the phrase "the first benchmark that integrates real-world measurements with paired numerical simulations across complex physical systems" to **"the first benchmark for scientific ML that..."**. Additionally, we have **cited** the ERCOFTAC database in the Related Work of the revised manuscript. If you have any other databases you recommend citing, please let us know.
>
> >7. The update ratio metric is interesting, but it conflates pretraining data scale with optimization effects.
>
> A: We point out that the distinction between Real-world Training and Real-world Finetuning lies in the model initialization. Apart from this difference, both approaches use **the same optimizer and other training settings**. The goal of this comparison is to observe the difference in convergence speed, thereby demonstrating how pretraining with simulated data can accelerate convergence.
>
> >8. Please make the train, validation, and test splits explicit at the parameter level so that generalization across Reynolds number, control frequency, mass ratio, or equivalence ratio is clear. I consider those to be most interesting axes. And if you already do this, show it more prominently.
>
> A: We actually split the train, validation, and test data at the parameter level. This statement has been added in Sec 4.1 of the revised manuscript.

---

> ### Author Response · Authors · 2025-11-20
> **Official Response to Reviewer 6AF9 (4)**
>
> >9. The autoregressive evaluation stops very early. If you want to make claims about stability, show longer horizons and add probe-based diagnostics, not only field RMSE.
>
> A: Thanks for the constructive suggestions. We report **5- and 10-round evaluation** results on Cylinder in the table below and in Fig. 3(c) of the revised manuscript. We note that 5 and 10 rounds correspond to 100 and 200 steps of output.
> |RMSE|FNO|WDNO|UNet|DeepONet|MWT|CNO|GK-Transformer|Transolver|DPOT-S-FT|DPOT-L-FT|DMD|
> |-|-|-|-|-|-|-|-|-|-|-|-|
> | 5 rounds  | 0.08352 | 0.07270 | 0.08849 | 0.09644| 0.07154 | 0.10907 | 0.10259 | 0.09421 | 0.07815 |0.07653|0.11710|
> | 10 rounds | 0.10457 | 0.08616 | 0.09641 | 0.10133| 0.08503 | 0.14075 | 0.10629 | 0.09904 | 0.08981|0.08833|0.13149|
>
> In addition, we have added the probe-based diagnostic "mean velocity profile error (MVPE)" under 10 round evaluation on Cylinder in the table below and in Fig. 5 (a) of the revised manuscript. We also visualize the mean velocity profiles of baselines in Fig. 5(b), Fig. 7 and Fig. 8.
> | | FNO| WDNO | UNet | DeepONet | MWT | CNO | GK-Transformer | Transolver | DPOT-S-FT | DPOT-L-FT | DMD |
> |-|-|-|-|-|-|-|-|-|-|-|-|
> | MVPE | 0.01317 | 0.02181 | 0.01405 | 0.01787| 0.01821 | 0.03488 | 0.01485| 0.02374 | 0.01363 | 0.01250| 0.10668 |
>
> >10. The baselines are modern ML models, which is fine for a benchmark, but the story would be stronger if you add one or two domain baselines for each scenario (for example a simple reduced order model, or even a physics based filter) so readers have a calibration point.
>
> A: Thanks for your suggestion! We have added the **Dynamic Mode Decomposition (DMD)** method as the baseline [1,2]. We have provided a detailed introduction in the Appendix F.1 of the revised manuscript and referred to it in Sec. 3.4. The experimental results are also added to the Table 1, Fig. 3, Fig. 4, Fig. 5, Fig. 6, and all appendix tables of the revised manuscript. We can see that the error of DMD is larger than that of deep learning models, which is a reasonable result because neural networks have a large number of parameters and the ability to fit complex nonlinear mappings from data. This also indicates that neural networks may bring new opportunities for the development of fluid mechanics, which is widely recognized in the fluid mechanics community [3,4]. However, it is worth noting that DMD is also a valuable method as it does not require a large amount of training data and training process to capture the basic modes of the flow field, which is important for analyzing fluid phenomena, while neural networks are black box models.
>
> [1] Kutz, J N, etc. Dynamic mode decomposition: data-driven modeling of complex systems[M]. SIAM, 2016.
>
> [2] Kou, J, & Zhang, W. Dynamic mode decomposition with exogenous input for data-driven modeling of unsteady flows[J]. Physics of fluids, 2019.
>
> [3] Brunton, S L, etc. Machine learning for fluid mechanics. Annual review of fluid mechanics[J], 2020.
>
> [4] Zhang, W, etc. A scientometric investigation of artificial intelligence for fluid mechanics: Emerging topics and active groups[J]. Progress in Aerospace Sciences, 2025.
>
> >11. Throughout the paper there are small terminology issues. I would prefer "numerical error" over "computational error" (computational error sounds like a code bug). Be precise about whether errors arise from discretization, closure modeling, boundary conditions, or measurement.
>
> A: Thanks for your suggestion. We have modified the "computational error" to the "numerical error" in the revised manuscript. As you understand, this error refers to the error of numerical methods. We consider that this error is due to the modeling using the implicit LES [1], as well as the discrete error caused by second-order convergence [2]. The measurement error of real-world data may be caused by uneven inflow (the incoming flow cannot be strictly guaranteed to be uniform) and camera noise. We have identified the source of the error in l155-158 in the revised manuscript.
>
> [1] Pitsch, H, etc. Large-eddy simulation of premixed turbulent combustion using a level-set approach[J]. Proceedings of the Combustion Institute, 2002.
>
> [2] Weymouth, G D, etc. Boundary data immersion method for Cartesian-grid simulations of fluid-body interaction problems[J]. Journal of Computational Physics, 2011.
>
> >12. Not a criticism, but I think it would be better to rename the benchmark. "RealBench" seems far too broad and will clash with many domains. Something like "RealPDEBench" or "RealFlowBench" might be more appropriate.
>
> A: Thank you for the good suggestion. We have renamed our benchmark from "RealBench" to "RealPDEBench", as shown throughout the paper. The title will be updated in the camera-ready version, because it cannot be modified in OpenReview now.
>
> We hope that the above have addressed your questions and resolved your concerns. If there is anything else you'd like to discuss, please feel free to reach out, and we will be glad to respond.

---

### Author Response · Authors · 2025-11-20
**General Response**

We thank the reviewers for the thorough reviews and constructive suggestions. We acknowledge the positive comments such as a good contribution in the right direction (Reviewer 1VNT, YXXo, kvd1), important and valuable (Reviewer kvd1, YXXo, 1VNT 6AF9), comprehensive (Reviewer 6AF9, 1VNT, YXXo), good codebase (Reviewer 6AF9, 1VNT), useful training and evaluation design (Reviewer 6AF9), well written (Reviewer 1VNT, YXXo), and thoughtfully formulated (Reviewer 1VNT, YXXo). We also believe that our proposed RealPDEBench would significantly contribute to the community.
Based on the reviewers' valuable feedback, we have conducted additional experiments and revised the manuscript, which hopefully resolve the reviewers' concerns. The major additional experiments and improvements are as follows:
1. We conduct longer rollout for autoregressive evaluation on Cylinder and report results in Fig. 3(c). Specifically, we take **5 and 10 rounds of autoregressive evaluation**, which corresponds to 100 and 200 steps of output. For more details, see responses to Reviewer 6AF9 and YXXo.
2. We provide **normalized Fourier Space Error** for better comparisons between different frequency bands. The results are updated in Fig. 6. More details can be found in responses to Reviewer YXXo.
3. We add **summary-statistic metrics** commonly used in previous numerical studies of fluid dynamics and combustion to Sec. 3.3.2, Fig.5, and Appendix C.4. For the fluid datasets, we introduce the mean velocity profile error, while for the combustion datasets, we introduce frequency-domain analysis. These metrics enable a statistically grounded comparison of discrepancies. Based on it, we also compare the error between simulated and real-world data, and the error between model predictions and real-world data, which further demonstrates the importance of real-world data for model training. See responses to Reviewer 6AF9 and YXXo for more details.
4. We add **details of data measurement** in Appendix C.3, including parameter information of the circulating water tank, details on parameter settings and attached devices for five datasets, detailed information on PIV measurements, and an introduction to post-processing software. We also add numerical generation scripts in Appendix C.2. This allows users to more easily extend the datasets and gain a deeper understanding of them. For details, see responses to Reviewer 6AF9 and kvd1.
5. We also **revise several descriptions in the manuscript** to improve clarity and readability. These updates include providing a clearer record of the step size and modalities in Table 7, explaining the sources of numerical error, and detailing how to perform out-of-domain testing. Please refer to responses to Reviewer 6AF9, kvd1 and 1VNT for details.
6. We add **new code functionalities** based on the reviewers' suggestions, including the option to select different noise types according to sensor physics and the ability to reserve a portion of the simulated data as a test set. For more details, see responses to Reviewer 6AF9 and YXXo.
7. We add the statement of releasing numerical scripts, raw data, calibration files, and processing software in the Reproducibility Statement. See responses to Reviewer 6AF9 and kvd1 for more details.

---

### Author Response · Authors · 2025-12-02
**Summary of the Discussion Period**

Dear Area Chair,

We sincerely appreciate the considerable time and effort you devote to re-evaluating the rebuttal and discussion of our manuscript, as well as to assessing the overall quality of our work. Here, we provide a concise summary of the rebuttal and discussion stages, so that you may clearly see that we have addressed nearly all reviewer concerns during the discussion, and they presented positive attitudes toward our discussions and the improvements of the revised manuscripts.

By establishing the first benchmark in Scientific ML for complex physics with paired real-world and simulated data, our work addresses a critical gap. It has been commended by reviewers for its important and comprehensive dataset, modular and clean code, and excellently written presentation. We are convinced it offers a foundational resource of lasting value to the Scientific ML community.

Regarding Reviewer 6AF9, we have addressed each of the concerns point by point. Although we have not yet received a follow-up feedback, the reviewer explicitly stated in the review that he/she is "open to potentially increasing score", which leads to a positive evaluation.

In response to Reviewer kvd1 with initially positive evaluations, we have diligently addressed all of the raised concerns. However, we have not received a response from this reviewer.

In addition, we extend our sincere appreciation to Reviewers 1VNT and YXXo for their positive assessment and perfect score of 10. In response to their constructive feedback, we have responded to all comments and incorporated corresponding revisions, which have further enhanced the overall quality of our paper.

In conclusion, we thank the reviewers for their input and have addressed all concerns, which has ultimately strengthened our paper. The specific positive comments and suggestions from the reviewers are outlined in our general response.

Kind regards,

Authors of ICLR Submission 2270

---

### Meta-Review · Area_Chair_p3zj · 2026-01-05

**Summary:**

Below is a summary of the major concerns raised from the four reviews:

1) Literature review: The paper should conduct a more thorough literature review to better position itself in the fluid community, particularly on the topic of measuring the sim-to-real gap in fluid experiments.

2) Data generation: The paper needs to provide more detailed documentation to better explain how it generated the data in the benchmark.

3) Sim-to-real gaps: reviewers raised multiple questions regarding this topic. Reviewers suggested more evaluations on how accurately the numerical simulation represents real-world scenarios, questioned the sensor noise models, and expressed concerns that the metrics (frequency or Fourier errors) are not able to capture summary statistics of fluids.

4) Model comparisons: Models should be normalized on certain metrics (e.g., FLOPs or parameter numbers) to ensure a fair comparison. Reviewers also suggested comparing with domain-specific methods, in addition to general ML models.

5) Autoregressive rollouts: Multiple reviewers suggested reporting longer rollouts for the autoregressive evaluation to better understand its performance in more practical settings.

**Reviewer Concerns:**

Overall, I think the rebuttal has properly addressed most of the questions.

1) The rebuttal has added and discussed more papers from the fluid community to address this concern. In particular, it distinguished itself from the fluid database (ERCOFTAC) mentioned by the reviewer. I think the response has largely resolved this concern, except that I am unsure whether the reviewer would consider the revised description “the first benchmark for scientific ML that …” acceptable or still think it is a bold claim.

2) The rebuttal clarified that all data were collected through their own experiments and added detailed descriptions in the appendix. I think this response has addressed the concern.

3) The rebuttal reported summary statistics (e.g., mean velocity profile) to demonstrate how accurately the numerical simulation captured real-world scenarios. I think the responses have largely addressed most of the sim-to-real questions. I am just a bit confused about their answer for the sensor noise models.

4) The rebuttal stated the difficulty of aligning the comparisons among different models and introduced a new baseline Dynamic Mode Decomposition (DMD). The new DMD experiment concludes that neural network models outperform DMD, suggesting new opportunities for fluid research. I think this new experiment has partially addressed the concern. Ensuring a fair comparison among different models remains an open issue, given the inclusion of some foundation models. Still, the rebuttal has made some fair points on this.

5) The rebuttal responded with additional statistics from 100-200 steps of output. I think the new statistics help address this concern.

**Reviewer Scores:**

Notably, the paper received two 10s before the rebuttal started. I think both reviewers would likely strongly support the paper, given that they did not raise any critical issues that the rebuttal did not address.

Another reviewer was mildly positive (6) before the rebuttal. I think they would maintain their positive score, given their relatively short statement about the paper’s weaknesses.

Finally, the negative reviewer (4) provided a high-quality review, with many thought-provoking comments on the paper's weaknesses. I think the rebuttal has partially addressed them as summarized above. I guess they would have maintained a borderline score and would have been unlikely to become more negative had they been able to participate in the discussion.

Overall, I think the four reviewers would very likely reach a consensus of accepting this paper.

---

### Decision · Program_Chairs · 2026-01-26

Accept (Oral)